# Bayesian Online Natural Gradient (BONG)

**Matt Jones**
University of Colorado
mcjones@colorado.edu

**Peter Chang**
MIT
gyuyoung@mit.edu

**Kevin Murphy**
Google DeepMind
kpmurphy@google.com

## Abstract

We propose a novel approach to sequential Bayesian inference based on variational Bayes (VB). The key insight is that, in the online setting, we do not need to add the KL term to regularize to the prior (which comes from the posterior at the previous timestep); instead we can optimize just the expected log-likelihood, performing a single step of natural gradient descent starting at the prior predictive. We prove this method recovers exact Bayesian inference if the model is conjugate. We also show how to compute an efficient deterministic approximation to the VB objective, as well as our simplified objective, when the variational distribution is Gaussian or a sub-family, including the case of a diagonal plus low-rank precision matrix. We show empirically that our method outperforms other online VB methods in the non-conjugate setting, such as online learning for neural networks, especially when controlling for computational costs.

## 1 Introduction

Bayesian methods for neural network (NN) training aim to minimize the Kullback-Leibler divergence between true and estimated posterior distributions. This is equivalent to minimizing the variational loss (or negative ELBO)

$$\mathcal{L}(\boldsymbol{\psi}) = -\mathbb{E}_{\boldsymbol{\theta} \sim q_{\boldsymbol{\psi}}}[\log p(\mathcal{D}|\boldsymbol{\theta})] + D_{\mathbb{KL}}(q_{\boldsymbol{\psi}}|p_0) \tag{1}$$

Here $\boldsymbol{\theta}$ are the network parameters, $\boldsymbol{\psi}$ are the variational parameters of the approximate posterior $q_{\boldsymbol{\psi}}(\boldsymbol{\theta})$, $\mathcal{D}$ is the training dataset, and $p_0(\boldsymbol{\theta})$ is the prior. The two terms in the variational loss correspond to data fit and regularization to the prior, the latter being analogous to a regularizer $r(\boldsymbol{\theta}) = -\log p_0(\boldsymbol{\theta})$ in traditional point estimation methods like SGD.

An important set of approaches learns the variational parameters by gradient descent on $\mathcal{L}(\boldsymbol{\psi})$ [Blundell et al., 2015]. More recently Khan and colleagues [Khan et al., 2018b, Khan and Rue, 2023, Shen et al., 2024] have proposed using the natural gradient $\mathbf{F}_{\boldsymbol{\psi}}^{-1} \nabla_{\boldsymbol{\psi}} \mathcal{L}(\boldsymbol{\psi})$ where $\mathbf{F}_{\boldsymbol{\psi}}$ is the Fisher information matrix of the variational family evaluated at $q_{\boldsymbol{\psi}}$. Natural gradient descent (NGD) is often more efficient than vanilla GD because it accounts for the intrinsic geometry of the variational family [Amari, 1998]. Khan and Rue [2023] call this approach the "Bayesian Learning Rule" or BLR. Using various choices for the variational distribution, generalized losses replacing negative log-likelihood, and other approximations, they reproduce many standard optimization methods such as Adam, and derive new ones.

We study Bayesian NN optimization in online learning, where the data are observed in sequence, $\mathcal{D}_t = \{(\boldsymbol{x}_k, \boldsymbol{y}_k)_{k=1}^t\}$, and the algorithm maintains an approximate posterior $q_{\boldsymbol{\psi}_t}(\boldsymbol{\theta}_t) \approx p(\boldsymbol{\theta}_t|\mathcal{D}_t)$, which it updates at each step. Fast updates (in terms of both computational speed and statistical efficiency) are critical for many online learning applications [Zhang et al., 2024]. To allow for nonstationarity in the datastream, we include a time index on $\boldsymbol{\theta}_t$, to represent that the parameters may change over time, as is standard for approaches based on state-space models and the extended Kalman filter (see e.g., [Sarkka and Svensson, 2023]). The belief state is updated recursively using

38th Conference on Neural Information Processing Systems (NeurIPS 2024).

the prior $q_{\boldsymbol{\psi}_{t|t-1}}$ derived from the previous time step so that the variational loss becomes

$$\mathcal{L}(\boldsymbol{\psi}_t) = -\mathbb{E}_{\boldsymbol{\theta}_t \sim q_{\boldsymbol{\psi}_t}}[\log p(\boldsymbol{y}_t|\boldsymbol{x}_t, \boldsymbol{\theta}_t)] + D_{\mathbb{KL}}\big(q_{\boldsymbol{\psi}_t}|q_{\boldsymbol{\psi}_{t|t-1}}\big) \tag{2}$$

One option for this online learning problem is to apply NGD on $\mathcal{L}(\boldsymbol{\psi}_t)$ at each time step, iterating until $\boldsymbol{\psi}_t$ converges before consuming the next observation. Our first contribution is a proposal for skipping this inner loop by (a) performing a single natural gradient step with unit learning rate and (b) omitting the $D_{\mathbb{KL}}$ term in Eq. (2) so that learning is based only on expected loglikelihood:

$$\boldsymbol{\psi}_t = \boldsymbol{\psi}_{t|t-1} + \mathbf{F}_{\boldsymbol{\psi}_{t|t-1}}^{-1} \nabla_{\boldsymbol{\psi}_{t|t-1}} \mathbb{E}_{\boldsymbol{\theta}_t \sim q_{\boldsymbol{\psi}_{t|t-1}}}[\log p(\boldsymbol{y}_t|\boldsymbol{x}_t, \boldsymbol{\theta}_t)] \tag{3}$$

These two modifications work together: instead of regularizing toward the prior explicitly using $D_{\mathbb{KL}}\big(q_{\boldsymbol{\psi}_t}|q_{\boldsymbol{\psi}_{t|t-1}}\big)$, we do so implicitly by using $\boldsymbol{\psi}_{t|t-1}$ as the starting point of our single natural gradient step. This may appear as a heuristic but we prove in Proposition 4.1 that it yields exact Bayesian inference when $q_{\boldsymbol{\psi}}$ and $p(\boldsymbol{y}|\boldsymbol{x}, \boldsymbol{\theta})$ are conjugate and $q_{\boldsymbol{\psi}}$ is an exponential family with natural parameter $\boldsymbol{\psi}$. Thus our proposed update can be viewed as a relaxation of the Bayesian update to the non-conjugate variational case. As is common in work on variational inference, we view the result for the conjugate case as a motivating foundation that ensures our method is exact in certain simple settings. The experiments reported in Section 5 and Appendix B complement the theory by showing our method also works well in more general settings. We call Eq. (3) the Bayesian online natural gradient (BONG).

Our second contribution concerns ways of computing the expectation in Eqs. (1) to (3). This is intractable for NNs, even for variational distributions that are easy to compute, since the likelihood takes the form $p(\boldsymbol{y}_t|\boldsymbol{x}_t, \boldsymbol{\theta}_t) = p(\boldsymbol{y}_t|f(\boldsymbol{x}_t, \boldsymbol{\theta}_t))$ with $f(\boldsymbol{x}_t, \boldsymbol{\theta}_t)$, representing the function computed by the network, is a complex, nonlinear function of $\boldsymbol{\theta}_t$. Many previous approaches have approximated the expected loglikelihood by sampling methods which add variance and computation time depending on the number of samples [Blundell et al., 2015, Shen et al., 2024]. We propose a deterministic, closed-form update that applies when the variational distribution is Gaussian (or a sub-family) and the likelihood is an exponential family with natural parameter $f(\boldsymbol{x}_t, \boldsymbol{\theta}_t)$ and mean parameter $h(\boldsymbol{x}_t, \boldsymbol{\theta}_t)$ (e.g., for classification, $f$ returns the vector of class logits, $h$ returns class probabilities, and $h = \text{softmax}(f)$). This update can be derived in two equivalent ways. First, we use a local linear approximation of the network $h(\boldsymbol{x}_t, \boldsymbol{\theta}_t) \approx \bar{h}_t(\boldsymbol{\theta}_t)$ [Immer et al., 2021a] and a Gaussian approximation of the likelihood $\mathcal{N}(\boldsymbol{y}_t|\bar{h}_t(\boldsymbol{\theta}_t), \mathbf{R}_t)$ [Ollivier, 2018, Tronarp et al., 2018]. Under these assumptions the expectation in Eq. (3) can be calculated analytically. Alternatively, we use a different linear approximation $f(\boldsymbol{x}_t, \boldsymbol{\theta}_t) \approx \bar{f}_t(\boldsymbol{\theta}_t)$ and a delta approximation $q_{\boldsymbol{\psi}_{t|t-1}}(\boldsymbol{\theta}_t) \approx \delta_{\boldsymbol{\mu}_{t|t-1}}(\boldsymbol{\theta}_t)$ where $\boldsymbol{\mu}_{t|t-1} = \mathbb{E}_{q_{\boldsymbol{\psi}_{t|t-1}}}[\boldsymbol{\theta}_t]$ is the prior mean, so that the expectation in Eq. (3) is replaced by a plugin prediction. The linear($h$)-Gaussian approximation is previously known but the linear($f$)-delta approximation is new, and we prove in Proposition 4.2 that they yield the same update, which we call linearized BONG, or BONG-LIN. Finally, we discuss different ways of approximating the Hessian of the objective, which is needed for NGD.

Our BONG framework unifies several existing methods for Bayesian online learning, and it offers new algorithms based on alternative variational families or parameterizations. We define a large space of methods by combining 4 different update rules with 4 different ways of computing the relevant expected gradients and Hessians and 3 different variational families (Gaussians with full, diagonal, and diagonal-plus-low rank precision matrices). We conduct experiments systematically testing how these factors affect performance. We find support for all three principles of our approach— NGD, implicit regularization to the prior, and linearization— in terms of both statistical and computational efficiency. Code for our experiments is available at https://github.com/petergchang/bong/.

## 2   Related work

Variational inference approximates the Bayesian posterior from within some suitable family in a way that bypasses the normalization term [Zellner, 1988, Jordan et al., 1999]. A common choice for the variational family is a Gaussian. For online learning, the exact update equations for Gaussian variational filtering are given by the RVGA method of [Lambert et al., 2021]. This update is implicit but can be approximated by an explicit RVGA update which we show arises as a special case of BONG. Most applications of Gaussian VI use a mean-field approximation defined by diagonal covariance, which scales linearly with model size. More expressive but still linear in the model size are methods

that express the covariance [Tomczak et al., 2020] or precision [Mishkin et al., 2018, Lambert et al., 2023, Chang et al., 2023] as a sum of diagonal and low rank matrices (DLR). In this paper, we consider variational families defined by full covariance, diagonal covariance, and DLR covariance.

For NNs and other complicated models, even the variational approximation can be intractable, so methods have been developed to approximately minimize the VI loss. Bayes by backprop (BBB) [Blundell et al., 2015] learns a variational distribution on NN weights by iterated GD on the VI loss of Eq. (1). They focus on mean-field Gaussian approximations but the approach also applies to other variational families. Here we adapt BBB to online learning to compare to our methods.

The Bayesian learning rule (BLR) replaces BBB's GD with NGD [Khan and Rue, 2023]. Several variants of BLR have been developed including VON and VOGN for a mean-field Gaussian prior [Khan et al., 2018b] and SLANG for a DLR Gaussian [Mishkin et al., 2018]. BLR has also been used to derive versions of many classic optimizers including SGD, RMSprop and Adam [Khan et al., 2018a, Khan and Rue, 2023, Lin et al., 2024, Shen et al., 2024]. Although BLR has been applied to online learning, we are particularly interested in Bayesian filtering including in nonstationary environments, where observations must be processed one at time and updates are based on the posterior from the previous step, often in conjunction with parameter dynamics. We therefore develop filtering versions of BLR to compare to BONG, some of which reduce to VON, VOGN and SLANG in the batch setting, while others are novel. We also note BLR is a mature theory including several clever tricks we have not yet incorporated into our framework.

Khan and Rue [2023] observe that conjugate updating is equivalent to one step of BLR with learning rate 1. This is similar to our Proposition 4.1 except that BLR retains the KL term in the variational loss. BLR and BONG agree in this case because the gradient of the KL is zero on BLR's first iteration: $\nabla_{\psi=\psi_{t|t-1}} D_{\mathbb{KL}}\big(q_\psi | q_{\psi_{t|t-1}}\big) = 0$. Therefore BONG can be seen as a special case of BLR with one update step per observation and learning rate 1. Our contribution is to recognize that doing a single update step allows the KL term to be dropped entirely, yielding a substantially simpler algorithm which our experiments show also performs better.

While BLR allows alternative losses in place of the NLL in Eq. (2), we can also replacing the KL term with other divergences [Knoblauch et al., 2022]. Our approach fits within that "generalized VB" framework in that it drops the divergence altogether. Our approach of implicitly regularizing to the prior using a single NGD step is also similar to the implicit MAP filter of [Bencomo et al., 2023] which performs truncated GD from the prior mode. The principal difference is they perform GD on model parameters ($\theta_t$) while we do NGD on the variational parameters ($\psi_t$). Thus BONG maintains a full prior and posterior while IMAP is more concerned with how the choice of optimizer can substitute for explicit tracking of covariance.

We show two other ways to derive the BONG update in Appendix D, one of which is to replace the expected NLL in Eq. (2) with a linear approximation and solve the resulting equation exactly. Several past works have taken this approach, arriving at updates similar to ours. Chérief-Abdellatif et al. [2019] study streaming variational Bayes and propose solving Eq. (2) with a linearized expected NLL. When the variational family is an exponential family their update becomes NGD [Khan and Lin, 2017] and matches the BONG update. Hoeven et al. [2018] show how mirror descent can be derived as a special case of Exponential Weights [Littlestone and Warmuth, 1994], which is closely related to Bayesian updating. The resulting algorithm is similar to BONG and follows from linearizing the NLL instead of expected NLL, with an additional delta assumption at the prior mean. Lyu and Tsang [2021] study relaxed block-box optimization where the objective is $\arg\min_{\psi} \mathbb{E}_{\boldsymbol{x} \sim q_\psi}[f(\boldsymbol{x})]$ for some target function $f$. They use a mirror descent formulation with linearized expected loss and KL regularizer and show the resulting update is NGD on expected loss, formally equivalent to our BONG update. From the perspective of this prior work, our contribution is to express the BONG update simply as NGD on the expected NLL, motivated by replacing the KL with implicit regularization, and to show how this yields a variety of known and novel algorithms for Bayesian filtering.

EKF applications to NNs apply Bayesian filtering using a local linear approximation of the network, leading to simple closed form updates [Singhal and Wu, 1989, Puskorius and Feldkamp, 1991]. The classic EKF assumes a Gaussian observation distribution but it has been extended to other exponential families (e.g. for classification) by matching the mean and covariance in what we call the conditional moments EKF (CM-EKF) [Ollivier, 2018, Tronarp et al., 2018]. Applying a KL projection to diagonal covariance yields the variational diagonal EKF (VD-EKF) [Chang et al., 2022].

Alternatively, projecting to diagonal plus low rank precision using SVD gives LO-FI [Chang et al., 2023]. We derive all these methods as special cases of BONG-LIN. Further developments in this direction include the method of [Titsias et al., 2024] which does Bayesian filtering on only the final weight layer, and WoLF [Duran-Martin et al., 2024] which achieves robustness to outliers through data-dependent weighting of the loglikelihood.

## 3 Background

We study online supervised learning where the agent receives input $\boldsymbol{x}_t \in \mathbb{R}^D$ and observation $\boldsymbol{y}_t \in \mathbb{R}^C$ on each time step, which it aims to model with a function $f_t(\boldsymbol{\theta}_t) = f(\boldsymbol{x}_t, \boldsymbol{\theta}_t)$ such as a NN with weights $\boldsymbol{\theta}_t \in \mathbb{R}^P$. The predictions for $\boldsymbol{y}_t$ are given by some observation distribution $p(\boldsymbol{y}_t | f_t(\boldsymbol{\theta}_t))$. For example, $f$ may compute the mean for regression or the class logits for classification.

We work in a Bayesian framework where the agent maintains an approximate posterior distribution over $\boldsymbol{\theta}_t$ after observing data $\mathcal{D}_t = \{(\boldsymbol{x}_k, \boldsymbol{y}_k)_{k=1}^t\}$. The filtering posterior $q_{\boldsymbol{\psi}_t}(\boldsymbol{\theta}_t) \approx p(\boldsymbol{\theta}_t | \mathcal{D}_t)$ is approximated within some parametric family indexed by the variational parameter $\boldsymbol{\psi}_t$. We allow for nonstationarity by assuming $\boldsymbol{\theta}$ changes over time according to some dynamic model $p(\boldsymbol{\theta}_t | \boldsymbol{\theta}_{t-1})$. By pushing the posterior from step $t-1$ through the dynamics we obtain a prior for step $t$ given by $q_{\boldsymbol{\psi}_{t|t-1}}(\boldsymbol{\theta}_t) \approx p(\boldsymbol{\theta}_t | \mathcal{D}_{t-1})$. For example suppose the variational posterior from the previous step is Gaussian, $q_{\boldsymbol{\psi}_{t-1}}(\boldsymbol{\theta}_{t-1}) = \mathcal{N}(\boldsymbol{\theta}_{t-1} | \boldsymbol{\mu}_{t-1}, \boldsymbol{\Sigma}_{t-1})$, and the dynamics model is an Ornstein-Uhlenbeck process, as proposed in prior work [Kurle et al., 2020, Titsias et al., 2024] to handle non-stationarity, i.e., the dynamics model has the form $\boldsymbol{\theta}_t \sim \mathcal{N}(\gamma_t \boldsymbol{\theta}_{t-1} + (1 - \gamma_t)\boldsymbol{\mu}_0, \mathbf{Q}_t)$, where $\mathbf{Q}_t = (1 - \gamma_t^2)\boldsymbol{\Sigma}_0$ is the covariance of the noise process, $0 \le \gamma_t \le 1$ is the degree of drift, and $p(\boldsymbol{\theta}_0) = \mathcal{N}(\boldsymbol{\mu}_0, \boldsymbol{\Sigma}_0)$ is the prior. In this case, the parameters of the prior predictive distribution are $\boldsymbol{\mu}_{t|t-1} = \gamma_t \boldsymbol{\mu}_{t-1} + (1 - \gamma_t)\boldsymbol{\mu}_0$ and $\boldsymbol{\Sigma}_{t|t-1} = \gamma_t^2 \boldsymbol{\Sigma}_{t-1} + \mathbf{Q}_t$. In general the predict step may require approximation to stay in the variational family (e.g., if the dynamics are nonlinear). In this paper, our focus is the update step from $\boldsymbol{\psi}_{t|t-1}$ to $\boldsymbol{\psi}_t$ upon observing $(\boldsymbol{x}_t, \boldsymbol{y}_t)$, so for simplicity we assume constant (static) parameters, i.e., $p(\boldsymbol{\theta}_t | \boldsymbol{\theta}_{-1}) = \delta(\boldsymbol{\theta}_t - \boldsymbol{\theta}_{t-1})$ (equivalently $\gamma_t = 1$), so $\boldsymbol{\psi}_{t|t-1} = \boldsymbol{\psi}_{t-1}$; however, our method can trivially handle non-stationary parameters.

Variational inference seeks an approximate posterior that minimizes the KL divergence from the exact Bayesian update from the prior. In the online setting this becomes

$$\boldsymbol{\psi}_t^* = \arg\min_{\boldsymbol{\psi}} D_{\mathbb{KL}}\big(q_{\boldsymbol{\psi}}(\boldsymbol{\theta}_t) | Z_t^{-1} q_{\boldsymbol{\psi}_{t|t-1}}(\boldsymbol{\theta}_t) \, p(\boldsymbol{y}_t | f_t(\boldsymbol{\theta}_t))\big) = \arg\min_{\boldsymbol{\psi}} \mathcal{L}_t(\boldsymbol{\psi}) \qquad (4)$$

where $\mathcal{L}_t$ is the online VI loss defined in Eq. (2), and the normalization term $Z_t$ (which depends on $\boldsymbol{x}_t$) drops out as an additive constant. Our goal is an efficient approximate solution to this variational optimization problem.

We will sometimes assume the variational posterior $q_{\boldsymbol{\psi}}$ is an exponential family distribution with natural parameter $\boldsymbol{\psi}$ so that $q_{\boldsymbol{\psi}_t}(\boldsymbol{\theta}_t) = \exp\big(\boldsymbol{\psi}_t^\mathsf{T} T(\boldsymbol{\theta}_t) - \Phi(\boldsymbol{\psi}_t)\big)$, with log-partition function $\Phi$ and sufficient statistics $T(\boldsymbol{\theta}_t)$. Assuming $\Phi$ is strictly convex (which holds in the cases we study) there is a bijection between $\boldsymbol{\psi}_t$ and the dual (or expectation) parameter $\boldsymbol{\rho}_t = \mathbb{E}_{\boldsymbol{\theta}_t \sim q_{\boldsymbol{\psi}_t}}[T(\boldsymbol{\theta}_t)]$. Classical thermodynamic identities imply that the Fisher information matrix has the form $\mathbf{F}_{\boldsymbol{\psi}_t} = \partial \boldsymbol{\rho}_t / \partial \boldsymbol{\psi}_t$. This has important implications for NGD on exponential families [Khan and Rue, 2023] because it implies that for any function $\ell$ defined on the variational parameter space the natural gradient wrt natural parameters $\boldsymbol{\psi}_t$ is the regular gradient wrt the dual parameters $\boldsymbol{\rho}_t$, i.e., $\mathbf{F}_{\boldsymbol{\psi}_t}^{-1} \nabla_{\boldsymbol{\psi}_t} \ell = \nabla_{\boldsymbol{\rho}_t} \ell$.

## 4 Methods

We propose to approximate the variational optimization problem in Eq. (4) using the BONG update in Eq. (3). When $q_{\boldsymbol{\psi}}$ is an exponential family, the fact that the natural gradient wrt the natural parameters $\boldsymbol{\psi}_t$ is the regular gradient wrt the dual parameters $\boldsymbol{\rho}_t$ implies an equivalent mirror descent form (see Appendix D for further analysis of BONG from the MD perspective):

$$\boldsymbol{\psi}_t = \boldsymbol{\psi}_{t|t-1} + \nabla_{\boldsymbol{\rho}_{t|t-1}} \mathbb{E}_{\boldsymbol{\theta}_t \sim q_{\boldsymbol{\psi}_{t|t-1}}}[\log p(\boldsymbol{y}_t | \boldsymbol{x}_t, \boldsymbol{\theta}_t)] \qquad (5)$$

This is NGD with unit learning rate on the variational loss in Eq. (2) but ignoring the $D_{\mathbb{KL}}\big(q_{\boldsymbol{\psi}} | q_{\boldsymbol{\psi}_{t|t-1}}\big)$ term. In this section we first prove this method is optimal when the model is conjugate and then describe extensions to more complex cases of practical interest.

## 4.1 Conjugate case

Our approach is motivated by the following result which states that BONG matches exact Bayesian inference when the variational distribution and the likelihood are conjugate exponential families:

**Proposition 4.1.** *Let the observation distribution (likelihood) be an exponential family with natural parameter $\boldsymbol{\theta}_t$ (where $T_l(\boldsymbol{y}_t) = \boldsymbol{y}_t$ is the sufficient statistics for the likelihood and $A(\boldsymbol{\theta}_t)$ is the log-partition function)*

$$p_t(\boldsymbol{y}_t|\boldsymbol{\theta}_t) = \exp\left(\boldsymbol{\theta}_t^\mathsf{T} \boldsymbol{y}_t - A(\boldsymbol{\theta}_t) - b(\boldsymbol{y}_t)\right) \tag{6}$$

*and let the prior be the conjugate exponential family*

$$q_{\boldsymbol{\psi}_{t|t-1}}(\boldsymbol{\theta}_t) = \exp\left(\boldsymbol{\psi}_{t|t-1}^\mathsf{T} T(\boldsymbol{\theta}_t) - \Phi(\boldsymbol{\psi}_{t|t-1})\right) \tag{7}$$

*with $T(\boldsymbol{\theta}_t) = [\boldsymbol{\theta}_t; -A(\boldsymbol{\theta}_t)]$. Then the exact Bayesian update agrees with Eq. (5).*

The proof is in Appendix C. Writing the natural parameters of the prior as $\boldsymbol{\psi}_{t|t-1} = [\boldsymbol{\chi}_{t|t-1}; \nu_{t|t-1}]$, we show the Bayesian update and BONG both yield $\boldsymbol{\chi}_t = \boldsymbol{\chi}_{t|t-1} + \boldsymbol{y}_t$ and $\nu_t = \nu_{t|t-1} + 1$. Intuitively, we are just accumulating a sum of the observed sufficient statistics, and a counter of the sample size (number of observations seen so far).

## 4.2 Variational case

In practical settings the conjugacy assumption of Proposition 4.1 will not be met, so Eqs. (3) and (5) will only approximate the Bayesian update. In this paper we restrict to Gaussian variational families. We refer to the unrestricted case as FC (full covariance), defined by the variational distribution

$$q_{\boldsymbol{\psi}_{t|t-1}}(\boldsymbol{\theta}_t) = \mathcal{N}\left(\boldsymbol{\theta}_t | \boldsymbol{\mu}_{t|t-1}, \boldsymbol{\Sigma}_{t|t-1}\right) \tag{8}$$

where $\boldsymbol{\Sigma}_{t|t-1}$ can be any positive semi-definite (PSD) matrix. The natural and dual parameters are $\boldsymbol{\psi} = (\boldsymbol{\Sigma}^{-1}\boldsymbol{\mu}, -\frac{1}{2}\mathrm{vec}(\boldsymbol{\Sigma}^{-1}))$ and $\boldsymbol{\rho} = (\boldsymbol{\mu}, \mathrm{vec}(\boldsymbol{\mu}\boldsymbol{\mu}^\mathsf{T} + \boldsymbol{\Sigma}))$. Appendix E.1.1 shows that Eq. (5) translated back to $(\boldsymbol{\mu}, \boldsymbol{\Sigma})$ gives the following BONG update for the FC case:

$$\boldsymbol{\mu}_t = \boldsymbol{\mu}_{t|t-1} + \boldsymbol{\Sigma}_t \underbrace{\mathbb{E}_{\boldsymbol{\theta}_t \sim q_{\boldsymbol{\psi}_{t|t-1}}}[\nabla_{\boldsymbol{\theta}_t} \log p(\boldsymbol{y}_t|f_t(\boldsymbol{\theta}_t))]}_{\boldsymbol{g}_t} \tag{9}$$

$$\boldsymbol{\Sigma}_t^{-1} = \boldsymbol{\Sigma}_{t|t-1}^{-1} - \underbrace{\mathbb{E}_{\boldsymbol{\theta}_t \sim q_{\boldsymbol{\psi}_{t|t-1}}}\left[\nabla_{\boldsymbol{\theta}_t}^2 \log p(\boldsymbol{y}_t|f_t(\boldsymbol{\theta}_t))\right]}_{\mathbf{G}_t} \tag{10}$$

which matches the explicit update in the RVGA method of [Lambert et al., 2021].

## 4.3 Monte Carlo approximation

The integrals over the prior $q_{\boldsymbol{\psi}_{t|t-1}}$ in Eqs. (9) and (10) are generally intractable and must be approximated. One option is to use Monte Carlo, in what we call BONG-MC. Given $M$ independent samples $\hat{\boldsymbol{\theta}}_t^{(m)} \sim q_{\boldsymbol{\psi}_{t|t-1}}$, we estimate the expected gradient $\boldsymbol{g}_t = \mathbb{E}_{\boldsymbol{\theta}_t \sim q_{\boldsymbol{\psi}_{t|t-1}}}[\nabla_{\boldsymbol{\theta}_t} \log p(\boldsymbol{y}_t|f_t(\boldsymbol{\theta}_t))]$ and expected Hessian $\mathbf{G}_t = \mathbb{E}_{\boldsymbol{\theta}_t \sim q_{\boldsymbol{\psi}_{t|t-1}}}\left[\nabla_{\boldsymbol{\theta}_t}^2 \log p(\boldsymbol{y}_t|f_t(\boldsymbol{\theta}_t))\right]$ as the empirical means

$$\boldsymbol{g}_t^{\text{MC}} = \frac{1}{M}\sum_{m=1}^M \hat{\boldsymbol{g}}_t^{(m)}, \qquad \hat{\boldsymbol{g}}_t^{(m)} = \nabla_{\boldsymbol{\theta}_t = \hat{\boldsymbol{\theta}}_t^{(m)}} \log p(\boldsymbol{y}_t|f_t(\boldsymbol{\theta}_t)) \tag{11}$$

$$\mathbf{G}_t^{\text{MC-HESS}} = \frac{1}{M}\sum_{m=1}^M \hat{\mathbf{G}}_t^{(m)}, \quad \hat{\mathbf{G}}_t^{(m)} = \nabla_{\boldsymbol{\theta}_t = \hat{\boldsymbol{\theta}}_t^{(m)}}^2 \log p(\boldsymbol{y}_t|f_t(\boldsymbol{\theta}_t)) \tag{12}$$

We use $\mathbf{G}^{\text{MC-HESS}}$ only for small models. Otherwise we use empirical Fisher (Section 4.5).

## 4.4 Linearized BONG

As an alternative to BONG-MC, we propose a linear approximation we call BONG-LIN that yields a deterministic and closed-form update. Assume the likelihood is an exponential family as in Proposition 4.1 but with natural parameter predicted by some function $f_t(\boldsymbol{\theta}_t) = f(\boldsymbol{x}_t, \boldsymbol{\theta}_t)$:

$$p(\boldsymbol{y}_t|\boldsymbol{x}_t, \boldsymbol{\theta}_t) = \exp\left(f_t(\boldsymbol{\theta}_t)^\mathsf{T} \boldsymbol{y}_t - A(f_t(\boldsymbol{\theta}_t)) - b(\boldsymbol{y}_t)\right) \tag{13}$$

We also define the dual (moment) parameter of the likelihood as $h_t(\boldsymbol{\theta}_t) = \mathbb{E}\left[\boldsymbol{y}_t | f_t(\boldsymbol{\theta}_t)\right]$. In a NN, $f_t$ and $h_t$ are related by the final response layer. For example in classification $f_t$ and $h_t$ give the class logits and probabilities, with $h_t(\boldsymbol{\theta}_t) = \text{softmax}(f_t(\boldsymbol{\theta}_t))$, with $\boldsymbol{y}_t$ being the one-hot encoding.

We now define two methods for approximating the expected gradient $\boldsymbol{g}_t$ and expected Hessian $\mathbf{G}_t$, based on linearizing the predictive model at the prior mean $\boldsymbol{\mu}_{t|t-1}$ in terms of either $f_t(\boldsymbol{\theta}_t)$ or $h_t(\boldsymbol{\theta}_t)$, and then prove their equivalence.

The **linear($h$)-Gaussian** approximation [Ollivier, 2018, Tronarp et al., 2018] linearizes $h_t(\boldsymbol{\theta}_t)$

$$\bar{h}_t(\boldsymbol{\theta}_t) = \hat{\boldsymbol{y}}_t + \mathbf{H}_t(\boldsymbol{\theta}_t - \boldsymbol{\mu}_{t|t-1}) \tag{14}$$

$$\hat{\boldsymbol{y}}_t = h_t(\boldsymbol{\mu}_{t|t-1}) \tag{15}$$

$$\mathbf{H}_t = \frac{\partial h_t}{\partial \boldsymbol{\theta}_t}\bigg|_{\boldsymbol{\theta}_t = \boldsymbol{\mu}_{t|t-1}} \tag{16}$$

and approximates the likelihood by a Gaussian with variance based at $\boldsymbol{\mu}_{t|t-1}$

$$\bar{p}_t^{\text{LG}}(\boldsymbol{y}_t|\boldsymbol{\theta}_t) = \mathcal{N}(\boldsymbol{y}_t|\bar{h}_t(\boldsymbol{\theta}_t), \mathbf{R}_t), \quad \mathbf{R}_t = \mathbb{V}\left[\boldsymbol{y}_t|\boldsymbol{\theta}_t = \boldsymbol{\mu}_{t|t-1}\right] \tag{17}$$

The **linear($f$)-delta approximation** linearizes $f_t(\boldsymbol{\theta}_t)$ and maintains the original exponential family likelihood distribution in Eq. (13)

$$\bar{f}_t(\boldsymbol{\theta}_t) = f_t(\boldsymbol{\mu}_{t|t-1}) + \mathbf{F}_t(\boldsymbol{\theta}_t - \boldsymbol{\mu}_{t|t-1}) \tag{18}$$

$$\mathbf{F}_t = \frac{\partial f_t}{\partial \boldsymbol{\theta}_t}\bigg|_{\boldsymbol{\theta}_t = \boldsymbol{\mu}_{t|t-1}} \tag{19}$$

$$\bar{p}_t^{\text{LD}}(\boldsymbol{y}_t|\boldsymbol{\theta}_t) \propto \exp\left(\bar{f}_t(\boldsymbol{\theta}_t)^\intercal \boldsymbol{y}_t - A(\bar{f}_t(\boldsymbol{\theta}_t)) - b(\boldsymbol{y}_t)\right) \tag{20}$$

It also uses a plug-in approximation that replaces $q_{\boldsymbol{\psi}_{t|t-1}}(\boldsymbol{\theta}_t)$ with a point mass $\delta_{\boldsymbol{\mu}_{t|t-1}}(\boldsymbol{\theta}_t)$ so that the expected gradient and Hessian are approximated by their values at the prior mean, i.e., $\nabla_{\boldsymbol{\theta}_t = \boldsymbol{\mu}_{t|t-1}} \log \bar{p}_t^{\text{LD}}(\boldsymbol{y}_t|\boldsymbol{\theta}_t)$ and $\nabla^2_{\boldsymbol{\theta}_t = \boldsymbol{\mu}_{t|t-1}} \log \bar{p}_t^{\text{LD}}(\boldsymbol{y}_t|\boldsymbol{\theta}_t)$, rather than being sampled.

**Proposition 4.2.** *Under a Gaussian variational distribution, the linear($h$)-Gaussian and linear($f$)-delta approximations yield the same values for the expected gradient and Hessian*

$$\boldsymbol{g}_t^{\text{LIN}} = \mathbf{H}_t^\intercal \mathbf{R}_t^{-1}(\boldsymbol{y}_t - \hat{\boldsymbol{y}}_t) \tag{21}$$

$$\mathbf{G}_t^{\text{LIN-HESS}} = -\mathbf{H}_t^\intercal \mathbf{R}_t^{-1} \mathbf{H}_t \tag{22}$$

See Appendix C for the proof. The main idea for the $\boldsymbol{g}_t^{\text{LIN}}$ part is that the linear-Gaussian assumptions make the gradient linear in $\boldsymbol{\theta}_t$ so the expected gradient equals the gradient at the mean. The main idea for the $\mathbf{G}_t^{\text{LIN-HESS}}$ part is that eliminating the Hessian of the NN requires different linearizing assumptions for the Gaussian and delta approximations, and the remaining nonlinear terms (from the log-likelihood in Eq. (13)) agree because of the property of exponential families that the Hessian of the log-partition $A$ equals the conditional variance $\mathbf{R}_t$.

Applying Proposition 4.2 to Eqs. (9) and (10) gives the BONG-LIN update for a FC Gaussian prior

$$\boldsymbol{\mu}_t = \boldsymbol{\mu}_{t|t-1} + \mathbf{K}_t(\boldsymbol{y}_t - \hat{\boldsymbol{y}}_t) \tag{23}$$

$$\boldsymbol{\Sigma}_t = \boldsymbol{\Sigma}_{t|t-1} - \mathbf{K}_t \mathbf{H}_t \boldsymbol{\Sigma}_{t|t-1} \tag{24}$$

$$\mathbf{K}_t = \boldsymbol{\Sigma}_{t|t-1} \mathbf{H}_t^\intercal \left(\mathbf{R}_t + \mathbf{H}_t \boldsymbol{\Sigma}_{t|t-1} \mathbf{H}_t^\intercal\right)^{-1} \tag{25}$$

where $\mathbf{K}_t$ is the Kalman gain matrix (see Appendix E.1.2). This matches the CM-EKF [Tronarp et al., 2018, Ollivier, 2018].

## 4.5 Empirical Fisher

The methods in Sections 4.3 and 4.4 require explicitly computing the Hessian of the loss (MC-HESS) or the Jacobian of the network (LIN-HESS). These are too expensive for large models or high-dimensional observations. Instead we can use an empirical Fisher approximation that replaces the Hessian with the outer product of the gradient (see e.g, [Martens, 2020]).

| Name | Eqs. |
|---|---|
| MC-HESS | (11), (12) |
| LIN-HESS | (21), (22) |
| MC-EF | (11), (26) |
| LIN-EF | (28), (29) |

Table 1: The 4 Hessian approximations.

For the MC-EF variant, we make the following approximation:

$$\mathbf{G}_t^{\text{MC-EF}} = -\frac{1}{M} \hat{\mathbf{G}}_t^{(1:M)} \hat{\mathbf{G}}_t^{(1:M)\mathsf{T}} \tag{26}$$

where $\hat{\mathbf{G}}_t^{(1:M)} = [\hat{\boldsymbol{g}}_t^{(1)}, \ldots, \hat{\boldsymbol{g}}_t^{(M)}]$ is the $P \times M$ matrix of gradients from the MC samples.

We can also consider a similar approach for the LIN-EF variant that is Jacobian-free and sampling-free. Note that if $\hat{\boldsymbol{y}}_t$ were the true value of $\mathbb{E}[\boldsymbol{y}_t | \boldsymbol{x}_t]$ (i.e., if the model were correct) then we would have $\mathbb{E}[(\boldsymbol{y}_t - \hat{\boldsymbol{y}}_t)(\boldsymbol{y}_t - \hat{\boldsymbol{y}}_t)^\mathsf{T}] = \mathbf{R}_t$, implying $\mathbb{E}[\boldsymbol{g}_t^{\text{LIN}} (\boldsymbol{g}_t^{\text{LIN}})^\mathsf{T}] = -\mathbf{G}_t^{\text{LIN-HESS}}$. This suggests using

$$\boldsymbol{g}_t^{\text{LIN-EF}} = \nabla_{\boldsymbol{\theta}_t = \boldsymbol{\mu}_{t|t-1}} \left[ -\tfrac{1}{2} \left(\boldsymbol{y}_t - h_t(\boldsymbol{\theta}_t)\right)^\mathsf{T} \mathbf{R}_t^{-1}(\boldsymbol{y}_t - h_t(\boldsymbol{\theta}_t)) \right] \tag{27}$$

$$= \left( \frac{\partial h_t(\boldsymbol{\theta}_t)}{\partial \boldsymbol{\theta}_t} \right)^\mathsf{T}_{\boldsymbol{\theta}_t = \boldsymbol{\mu}_{t|t-1}} \mathbf{R}_t^{-1}(\boldsymbol{y}_t - h_t(\boldsymbol{\mu}_{t|t-1})) = \boldsymbol{g}_t^{\text{LIN}} \tag{28}$$

$$\mathbf{G}_t^{\text{LIN-EF}} = -\boldsymbol{g}_t^{\text{LIN}} (\boldsymbol{g}_t^{\text{LIN}})^\mathsf{T} \tag{29}$$

where Eq. (29) is the EF approximation to Eq. (22).

A more accurate EF approximation is possible by sampling virtual observations $\tilde{\boldsymbol{y}}_t$ from $p(\cdot | f_t(\hat{\boldsymbol{\theta}}_t^{(m)}))$ or $p(\cdot | f_t(\boldsymbol{\mu}_{t|t-1}))$ and using them for the gradients in Eq. (26) or Eq. (29) (respectively) [Martens, 2020, Kunstner et al., 2020]. However, in our experiments we use the actual observations $\boldsymbol{y}_t$ which is faster and follows previous work (e.g., [Khan et al., 2018b]).

### 4.6 Update rules

In addition to the four ways of approximating the expected Hessian (summarized in Table 1), we also consider four variants of BONG, based on what kind of loss we optimize and what kind of update we perform, as we describe below. See Table 2 for a summary.

| Name | Loss | Update |
|------|------|--------|
| BONG | $\mathbb{E}[\text{NLL}]$ | NGD($I=1$) |
| BOG | $\mathbb{E}[\text{NLL}]$ | GD($I=1$) |
| BLR | VI | NGD($I \geq 1$) |
| BBB | VI | GD($I \geq 1$) |

Table 2: The 4 update algorithms.

BONG (Bayesian online natural gradient) performs one step of NGD on the expected log-likelihood. We set learning rate to $\alpha_t = 1$ since this is optimal for conjugate models. The update (for an exponential variational family) is as in Eq. (5):

$$\boldsymbol{\psi}_t = \boldsymbol{\psi}_{t|t-1} + \nabla_{\boldsymbol{\rho}_{t|t-1}} \mathbb{E}_{\boldsymbol{\theta}_t \sim q_{\boldsymbol{\psi}_{t|t-1}}} [\log p(\boldsymbol{y}_t | \boldsymbol{x}_t, \boldsymbol{\theta}_t)] \tag{30}$$

BOG (Bayesian online gradient) performs one step of GD (instead of NGD) on the expected log-likelihood. We include a learning rate $\alpha$ because GD does not have the scale-invariance of NGD:

$$\boldsymbol{\psi}_t = \boldsymbol{\psi}_t + \alpha_t \nabla_{\boldsymbol{\psi}_t} \mathbb{E}_{\boldsymbol{\theta}_t \sim q_{\boldsymbol{\psi}_t}} [\log p(\boldsymbol{y}_t | f_t(\boldsymbol{\theta}_t))] \tag{31}$$

BLR (Bayesian learning rule, [Khan and Rue, 2023]) uses NGD (like BONG) but optimizes the VI loss using multiple iterations, instead of optimizing the expected NLL with a single step. When modified to the online setting, BLR starts an inner loop at each time step with $\boldsymbol{\psi}_{t,0} = \boldsymbol{\psi}_{t|t-1}$ and iterates

$$\boldsymbol{\psi}_{t,i} = \boldsymbol{\psi}_{t,i-1} + \alpha_t \mathbf{F}_{\boldsymbol{\psi}_{t,i-1}} \nabla_{\boldsymbol{\psi}_{t,i-1}} \left( \mathbb{E}_{\boldsymbol{\theta}_t \sim q_{\boldsymbol{\psi}_{t,i-1}}} [\log p(\boldsymbol{y}_t | f_t(\boldsymbol{\theta}_t))] - D_{\mathbb{KL}}\left( q_{\boldsymbol{\psi}_{t,i-1}} | q_{\boldsymbol{\psi}_{t|t-1}} \right) \right) \tag{32}$$

For an exponential variational family this can be written in mirror descent form

$$\boldsymbol{\psi}_{t,i} = \boldsymbol{\psi}_{t,i-1} + \alpha_t \nabla_{\boldsymbol{\rho}_{t,i-1}} \left( \mathbb{E}_{\boldsymbol{\theta}_t \sim q_{\boldsymbol{\psi}_{t,i-1}}} [\log p(\boldsymbol{y}_t | f_t(\boldsymbol{\theta}_t))] - D_{\mathbb{KL}}\left( q_{\boldsymbol{\psi}t,i-1} | q_{\boldsymbol{\psi}_{t|t-1}} \right) \right) \tag{33}$$

BBB (Bayes By Backprop, [Blundell et al., 2015]) is like BLR but uses GD instead of NGD. When adapted to online learning, it starts each time step at $\boldsymbol{\psi}_{t,0} = \boldsymbol{\psi}_{t|t-1}$ and iterates with GD:

$$\boldsymbol{\psi}_{t,i} = \boldsymbol{\psi}_{t,i-1} + \alpha_t \nabla_{\boldsymbol{\psi}_{t,i-1}} \left( \mathbb{E}_{\boldsymbol{\theta}_t \sim q_{\boldsymbol{\psi}_{t,i-1}}} [\log p(\boldsymbol{y}_t | f_t(\boldsymbol{\theta}_t))] - D_{\mathbb{KL}}\left( q_{\boldsymbol{\psi}_{t,i-1}} | q_{\boldsymbol{\psi}_{t|t-1}} \right) \right) \tag{34}$$

| Update | Hessian | Variational Family | | |
| --- | --- | --- | --- | --- |
| | | Full | Diag | DLR |
| BONG | MC-EF | $O(MP^2)$ [RVGA] | $O(MP)$ | $O((R+M)^2P)$ |
| BLR | MC-EF | $O(IP^3)$ | $O(IMP)$ [VON] | $O(I(R+M)^2P)$ [SLANG] |
| BOG | MC-EF | $O(P^3)$ | $O(MP)$ | $O(RMP)$ |
| BBB | MC-EF | $O(IP^3)$ | $O(IMP)$ [BBB] | $O(IR(R+M)P)$ |
| BONG | LIN-HESS | $O(CP^2)$ [CM-EKF] | $O(C^2P)$ [VD-EKF] | $O((R+C)^2P)$ [LO-FI] |
| BLR | LIN-HESS | $O(IP^3)$ | $O(IC^2P)$ | $O(I(2R+C)^2P)$ |
| BOG | LIN-HESS | $O(P^3)$ | $O(C^2P)$ | $O(C(C+R)P)$ |
| BBB | LIN-HESS | $O(IP^3)$ | $O(IC^2P)$ | $O(I(C+R)RP)$ |
| BONG | LIN-EF | $O(P^2)$ | $O(P)$ | $O(R^2P)$ |
| BLR | LIN-EF | $O(IP^3)$ | $O(IP)$ | $O(IR^2P)$ |
| BOG | LIN-EF | $O(P^3)$ | $O(P)$ | $O(RP)$ |
| BBB | LIN-EF | $O(IP^3)$ | $O(IP)$ | $O(IR^2P)$ |

Table 3: Time complexity of the algorithms. $P$: params, $C$: observation dim, $M$: MC samples, $I$: iterations, $R$: DLR rank. We assume $P \gg \{I, C, R, M\}$ so display only the terms of leading order in $P$. Time complexities for MC-HESS algorithms (not shown) are always at least as great as for the corresponding MC-EF. Full (full covariance) and Diag (diagonal covariance) columns indicate natural parameters; corresponding algorithms using moment parameters have the same complexities except BOG-FC_MOM which is $O(MP^2)$ for MC-EF, $O(CP^2)$ for LIN-HESS, and $O(P^2)$ for LIN-EF. [Names] correspond to the following existing methods (or variants thereof) in the literature: RVGA: [Lambert et al., 2021] (explicit update version); VON: [Khan et al., 2018b] (modified for online); SLANG: [Mishkin et al., 2018] (modified for online); BBB: [Blundell et al., 2015] (modified for online and uses moment parameters); CM-EKF: [Ollivier, 2018, Tronarp et al., 2018]; VD-EKF: [Chang et al., 2022]; LO-FI: [Chang et al., 2023].

## 4.7 Variational families and their parameterizations

We investigate five variational families for the posterior distribution: (1) FC Gaussian using natural parameters $\boldsymbol{\psi} = (\boldsymbol{\Sigma}^{-1}\boldsymbol{\mu}, -\frac{1}{2}\boldsymbol{\Sigma}^{-1})$, (2) FC Gaussian using central moment parameters $\boldsymbol{\psi} = (\boldsymbol{\mu}, \boldsymbol{\Sigma})$, (3) diagonal Gaussian using natural parameters $\boldsymbol{\psi} = (\boldsymbol{\sigma}^{-2}\boldsymbol{\mu}, -\frac{1}{2}\boldsymbol{\sigma}^{-2})$ (using elementwise exponents and products), (4) diagonal Gaussian using central moment parameters $\boldsymbol{\psi} = (\boldsymbol{\mu}, \boldsymbol{\sigma}^2)$, and (5) DLR Gaussian with parameters $\boldsymbol{\psi} = (\boldsymbol{\mu}, \boldsymbol{\Upsilon}, \mathbf{W})$ and precision $\boldsymbol{\Sigma}^{-1} = \boldsymbol{\Upsilon} + \mathbf{W}\mathbf{W}^\intercal$ where $\boldsymbol{\Upsilon} \in \mathbb{R}^{P \times P}$ is diagonal and $\mathbf{W} \in \mathbb{R}^{P \times R}$ with $R \ll P$. The moment parameterizations are included to test the importance of using natural parameters per Proposition 4.1. The diagonal family allows learning of large models because it scales linearly in the model size $P$. DLR also scales linearly but is more expressive than diagonal, maintaining some of the correlation information between parameters that is lost in the mean field (diagonal) approximation [Lambert et al., 2023, Mishkin et al., 2018, Chang et al., 2023].

Optimizing the BONG objective wrt $(\boldsymbol{\mu}, \boldsymbol{\Upsilon}, \mathbf{W})$ using NGD methods is challenging because the Fisher information matrix in this parameterization cannot be efficiently inverted. Instead we first derive the update wrt the FC natural parameters (leveraging the fact that the prior $\boldsymbol{\Sigma}_{t|t-1}^{-1}$ is DLR to make this efficient), and then use SVD to project the posterior precision back to low-rank form, following our prior LO-FI work [Chang et al., 2023]. However, if we omit the Fisher preconditioning matrix and use GD as in BOG and BBB, we can directly optimize the objective wrt $(\boldsymbol{\mu}, \boldsymbol{\Upsilon}, \mathbf{W})$ (see Appendix E.5).

## 4.8 Overall space of methods

Crossing the four algorithms in Table 2, the four methods of approximating the Hessian in Table 1, and the five variational families yields 80 algorithms. Table 3 shows the 36 based on the three tractable Hessian approximations and the three variational families that use natural parameters. Update equations for all the algorithms are derived in Appendix E. Pseudocode is given in Appendix A.

# 5 Experiments

This section presents our primary experimental results. These are based on MNIST ($D = 784$, $N_{\text{train}} = 60$k, $N_{\text{test}} = 10$k, $C = 10$ classes) [LeCun et al., 2010]. See Appendix B for more details on these experiments, and more results on MNIST and other datasets. We focus on training on a prefix of the first $T = 2000$ examples from each dataset, since our main interest is in online learning from potentially nonstationary distributions, where rapid adaptation of a model in response to a small number of new data points is critical.

Our primary evaluation objective is the negative log predictive density (NLPD) of the test set as a function of the number of training points observed so far.[1] It is defined as $\text{NLPD}_t = -\frac{1}{N_{\text{test}}} \sum_{i \in \mathcal{D}^{\text{test}}} \log \left[ \int p(\boldsymbol{y}_i | f(\boldsymbol{x}_i, \boldsymbol{\theta}_t)) q_{\boldsymbol{\psi}_t}(\boldsymbol{\theta}_t) d\boldsymbol{\theta}_t \right]$. We approximate this integral in two main ways: (1) using Monte Carlo sampling[2], or (2) using a plugin approximation, where we replace the posterior $q_{\boldsymbol{\psi}_t}(\boldsymbol{\theta}_t)$ with a delta function centered at the mean, $\delta(\boldsymbol{\theta}_t - \boldsymbol{\mu}_t)$.

For methods that require a learning rate (i.e., all methods except BONG), we optimize it wrt mid-way or final performance on a holdout validation set, using Bayesian optimization on NLL. All methods require specifying the prior belief state, $p(\boldsymbol{\theta}_0) = \mathcal{N}(\boldsymbol{\mu}_0, \boldsymbol{\Sigma}_0 = \sigma_0^2 \mathbf{I})$. We optimize over $\sigma_0^2$ and sample $\boldsymbol{\mu}_0$ from a standard NN initializer. As Hessian approximations, we use MC-EF with $M = 100$ samples, as well as the deterministic approximations LIN-HESS and LIN-EF. (In the appendix, we also study MC-HESS but find that it works very poorly, even with $M = 1000$.)

In Fig. 1 we compare the 4 main algorithms using DLR family with rank $R = 10$. We apply these to a CNN with two convolutional layers (each with 16 features and a (5,5) kernel), followed by two linear layers (one with 64 features and the final one with 10 features), for a total of 57,722 parameters. Shaded regions in these and all other plots indicate $\pm 1$ SE, based on 5 independent trials randomly varying in the prior mean $\boldsymbol{\mu}_0$, data ordering, and MC sampling. From this figure (and additional results in Appendix B) we conclude

- Linearization helps: LIN-HESS and LIN-EF both outperform the MC variants.
- NGD helps: BONG outperforms BOG.
- Implicit regularization helps: BONG outperforms BLR.
- The LIN-HESS approximation outperforms LIN-EF, at least for BONG.
- BBB generally does poorly.
- The BONG posterior predictive (using LIN-HESS) is slightly better calibrated than BOG, and both are much better than BLR and BBB, at least for small sample sizes, as shown in Fig. 7.
- The plugin posterior predictive is similar to the Lin-MC predictive (see Footnote 2), and both are generally much better than the simple MC predictive, as shown in Fig. 5.

In Fig. 2 we compare BONG using different variational families, and conclude

- DLR-10 outperforms DLR-1, which is similar to diagonal (except when using BONG-LIN-EF, where DLR-1 is worse than diagonal). Also, we find (in results not reported here) that rank 5–10 often gives results as good as FC, but is much cheaper.
- Both natural and moment parameterizations for the diagonal representation perform comparably to each other, although with LIN-EF, the moment parameterization can be numerically unstable.[3]

Finally, in Fig. 3 we report the runtimes from these experiments and conclude

- One-step methods (BONG and BOG) are faster than iterative methods (BLR and BBB), as expected.
- Linearized methods (LIN-HESS and LIN-EF) are faster than MC methods (MC-EF).

---

[1]We assume the training and test sets are drawn from the same static distribution. Alternatively, if there is only one stream of data coming from a potential notstationary source, we can use the prequential or one-step-ahead log predictive density Gama et al. [2013]. We leave studying the non-stationary case to future work.

[2]That is, we compute $S = 100$ posterior samples $\boldsymbol{\theta}_t^s \sim p(\boldsymbol{\theta}_t | \mathcal{D}_{1:t}^{\text{train}})$ and then use $p(\boldsymbol{y}|\boldsymbol{x}) \approx \frac{1}{S} \sum_{s=1}^{S} p(\boldsymbol{y}|\boldsymbol{x}, \boldsymbol{\theta}_t^s)$. For efficiently sampling from a DLR Gaussian we follow Mishkin et al. [2018]. Following Immer et al. [2021b], we find it better to linearize the model (i.e., replacing $h(\boldsymbol{x}_i, \boldsymbol{\theta}_t)$ with $\bar{h}(\boldsymbol{x}_i, \boldsymbol{\theta}_t)$ defined in Eq. (14)) before pushing posterior samples through. We call this the Lin-MC approximation.

[3]This is likely due to the fact that in the moment update in Eq. (249), we add $\text{diag}(\mathbf{G}_t)$ to the variance, whereas in the natural update in Eq. (213), we subtract $\text{diag}(\mathbf{G}_t)$ from the precision; the latter is more stable since $\mathbf{G}_t$ is negative semi-definite for all 4 Hessian approximations.

- EF methods (LIN-EF) are a bit faster than methods that compute the Hessian exactly (LIN-HESS), especially for diagonal family. (This speedup is larger when the output dimensionality $C$ is big.)

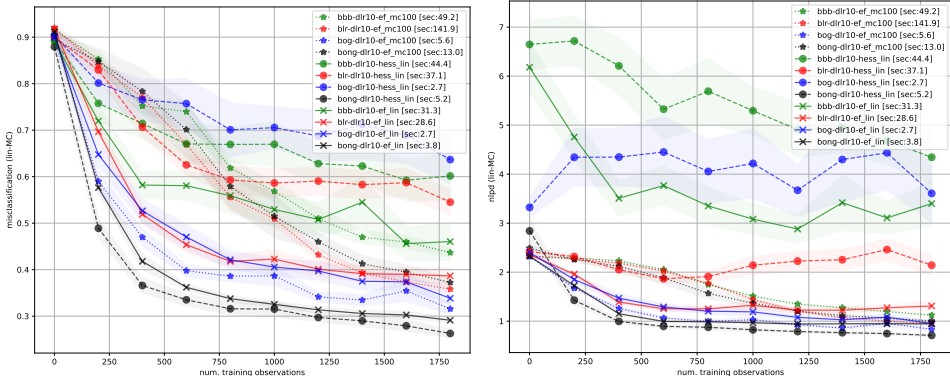

Figure 1: Performance on MNIST using Lin-MC posterior predictive, where the posterior is computed using BONG, BOG, BBB and BLR and the 3 tractable Hessian approximations with DLR-10 variational family.

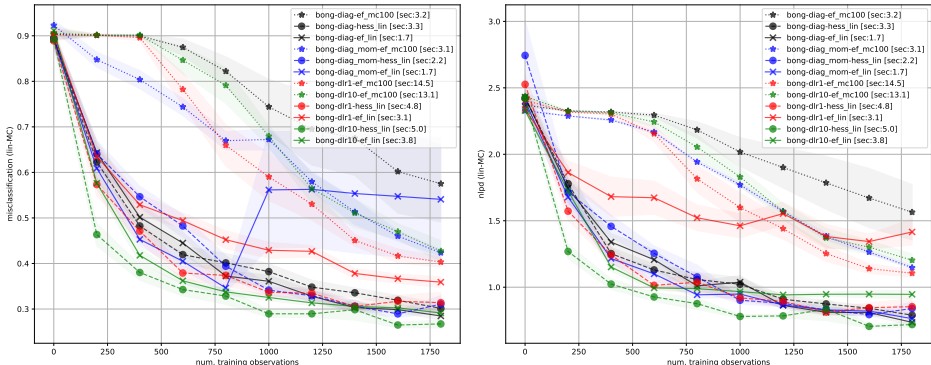

Figure 2: Performance on MNIST using Lin-MC posterior predictive, where the posterior is computed using BONG with different variational families, namely diagonal (natural and moment), DLR-1, DLR-10.

## 6 Conclusions, limitations and future work

Our experiment results show benefits of BONG's three main principles: NGD, implicit regularization to the prior, and linearization. The clear winner across datasets and variational families is BONG-LIN-HESS, which embodies all three principles. BLR-LIN-HESS nearly matches its performance but is much slower. Several of the best-performing algorithms are previously known (notably CM-EKF and LO-FI) but we explain these results within a systematic theory that also offers new methods (including BLR-LIN-HESS).

BONG is motivated by Proposition 4.1 which applies only in the idealized setting of a conjugate prior. Nevertheless we find it performs well in non-conjugate settings. On the other hand our experiments are based on relatively small models and datasets. It will be important to test how our methods scale up, especially using the promising DLR representation.

## Acknowledgments and Disclosure of Funding

Thanks to Gerardo Duràn-Martìn and Alex Shestopaloff for helpful input. MJ was supported by NSF grant 2020-906.

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

# A Abstract pseudocode

Algorithms 1 to 7 give pseudocode for applying the methods we study. For the predict step, we assume the dynamics model has the form $\boldsymbol{\theta}_t \sim \mathcal{N}(\mathbf{F}_t\boldsymbol{\theta}_{t-1} + \boldsymbol{b}_t, \mathbf{Q}_t)$. The update step in Algorithm 3 calls one of four grad functions (Algorithms 4 to 7) that estimate the expected gradient and Hessian using either MC or linearization combined with either the direct Hessian (or Jacobian and observation covariance) or EF. The update step also calls an inner step function that implements BONG, BLR, BOG or BBB on some variational family corresponding to the encoding of $\boldsymbol{\psi}$ (not shown). In practice the grad-fn and inner-step-fn are not as cleanly separated because the full matrix $\bar{\mathbf{G}}_{t,i}$ is not passed between them except when the variational family is FC. When the family is diagonal, grad-fn only needs to return $\mathrm{diag}(\bar{\mathbf{G}}_{t,i})$. When grad-fn uses EF, it only needs to return $\hat{\mathbf{G}}_{t,i}^{(1:M)}$ (grad-MC-EF) or $\boldsymbol{g}_{t,i}^{\mathrm{LIN}}$ (grad-LIN-EF) and inner-step-fn will implicitly use the outer product of this output as $\bar{\mathbf{G}}_{t,1}$. Finally, note the expressions for $\boldsymbol{g}_{t,i}^{\mathrm{LIN}}$ in Algorithms 6 and 7 are equivalent ways of computing the same quantity, as explained after Eq. (28).

---

**for** $t = 1 : \infty$ **do**
  $\quad \boldsymbol{\psi}_{t|t-1} = \mathrm{predict}(\boldsymbol{\psi}_{t-1})$
  $\quad \boldsymbol{\psi}_t = \mathrm{update}(\boldsymbol{\psi}_{t|t-1}, \boldsymbol{x}_t, \boldsymbol{y}_t)$
**end**

**Algorithm 1:** Main loop.

---

def $\mathrm{predict}(\boldsymbol{\psi}_{t-1} = (\boldsymbol{\mu}_{t-1}, \boldsymbol{\Sigma}_{t-1}))$:
$\boldsymbol{\mu}_{t|t-1} = \mathbf{F}_t\boldsymbol{\mu}_{t-1} + \boldsymbol{b}_t$
$\boldsymbol{\Sigma}_{t|t-1} = \mathbf{F}_t\boldsymbol{\Sigma}_{t-1}\mathbf{F}_t^\mathsf{T} + \mathbf{Q}_t$
Return $\boldsymbol{\psi}_{t|t-1} = (\boldsymbol{\mu}_{t|t-1}, \boldsymbol{\Sigma}_{t|t-1})$

**Algorithm 2:** Predict step.

---

def $\mathrm{update}(\boldsymbol{\psi}_{t|t-1}, \boldsymbol{x}_t, \boldsymbol{y}_t, f(), A(), \text{grad-fn}, \text{inner-step-fn}, \alpha, I, M)$:
$\boldsymbol{\psi}_{t,0} = \boldsymbol{\psi}_{t|t-1}$
$f_t(\boldsymbol{\theta}_t) = f(\boldsymbol{x}_t, \boldsymbol{\theta}_t)$
$h_t(\boldsymbol{\theta}_t) = E[\boldsymbol{y}_t|\boldsymbol{x}_t, \boldsymbol{\theta}_t] = \nabla_{\boldsymbol{\eta}=f_t(\boldsymbol{\theta}_t)}A(\boldsymbol{\eta})$
$\mathbf{V}_t(\boldsymbol{\theta}_t) = \mathrm{Cov}[\boldsymbol{y}_t|\boldsymbol{x}_t, \boldsymbol{\theta}_t] = \nabla^2_{\boldsymbol{\eta}=f_t(\boldsymbol{\theta}_t)}A(\boldsymbol{\eta})$
$\ell_t(\boldsymbol{\theta}_t) = \log p(\boldsymbol{y}_t|\boldsymbol{x}_t, \boldsymbol{\theta}_t) = \log p(\boldsymbol{y}_t|\boldsymbol{f}_t(\boldsymbol{\theta}_t))$
**for** $i = 1 : I$ **do**
  $\quad (\bar{\boldsymbol{g}}_{t,i}, \bar{\mathbf{G}}_{t,i}) = \text{grad-fn}(\boldsymbol{\psi}_{t,i-1}, \ell_t, h_t, \mathbf{V}_t, M)$
  $\quad \boldsymbol{\psi}_{t,i} = \text{inner-step-fn}(\boldsymbol{\psi}_{t|t-1}, \boldsymbol{\psi}_{t,i-1}, \bar{\boldsymbol{g}}_{t,i}, \bar{\mathbf{G}}_{t,i}, \alpha)$
**end**
Return $\boldsymbol{\psi}_{t,I}$

**Algorithm 3:** Update step. The inner-step-fn is BONG, BLR, BOG or BBB (not shown).

---

# B Additional experiment results

In this section, we give a more thorough set of experimental results.

## B.1 Running time measures

The running times of the methods for the experiments in Figs. 1 and 2, where we fit a CNN to MNIST, are shown in Fig. 3.

The running times of the methods for the FC and DLR case, where we fit an MLP to a synthetic regression dataset, are shown in Fig. 4. The slower speed of BLR (even with $I = 1$) relative to BONG is at least partly attributable to the fact that BLR must compute the SVD of a larger matrix (see Appendices E.5.3 and E.5.4).

def grad-MC-HESS($\boldsymbol{\psi}_{t,i-1}, \ell_t, h_t = [], \mathbf{V}_t = [], M$):
**for** $m = 1 : M$ **do**

$\quad \hat{\boldsymbol{\theta}}_{t,i}^{(m)} \sim q_{\boldsymbol{\psi}_{t,i-1}}(\boldsymbol{\theta})$

$\quad \hat{\boldsymbol{g}}_{t,i}^{(m)} = \nabla_{\boldsymbol{\theta}_t = \hat{\boldsymbol{\theta}}_{t,i}^{(m)}} \ell_t(\boldsymbol{\theta}_t)$

$\quad \hat{\mathbf{G}}_{t,i}^{(m)} = \nabla_{\boldsymbol{\theta}_t = \hat{\boldsymbol{\theta}}_{t,i}^{(m)}}^2 \ell_t(\boldsymbol{\theta}_t)$

**end**

$\boldsymbol{g}_{t,i}^{\text{MC}} = \frac{1}{M} \sum_{m=1}^{M} \hat{\boldsymbol{g}}_{t,i}^{(m)}$

$\mathbf{G}_{t,i}^{\text{MC-HESS}} = \frac{1}{M} \sum_{m=1}^{M} \hat{\mathbf{G}}_{t,i}^{(m)}$

Return ($\boldsymbol{g}_{t,i}^{\text{MC}}, \mathbf{G}_{t,i}^{\text{MC-HESS}}$)

**Algorithm 4:** MC gradient/Hessian estimator

def grad-MC-EF($\boldsymbol{\psi}_{t,i-1}, \ell_t, h_t = [], \mathbf{V}_t = [], M$):
**for** $m = 1 : M$ **do**

$\quad \hat{\boldsymbol{\theta}}_{t,i}^{(m)} \sim q_{\boldsymbol{\psi}_{t,i-1}}(\boldsymbol{\theta})$

$\quad \hat{\boldsymbol{g}}_{t,i}^{(m)} = \nabla_{\boldsymbol{\theta}_t = \hat{\boldsymbol{\theta}}_{t,i}^{(m)}} \ell_t(\boldsymbol{\theta}_t)$

**end**

$\boldsymbol{g}_{t,i}^{\text{MC}} = \frac{1}{M} \sum_{m=1}^{M} \hat{\boldsymbol{g}}_{t,i}^{(m)}$

$\hat{\mathbf{G}}_{t,i}^{(1:M)} = [\hat{\boldsymbol{g}}_{t,i}^{(1)}, \ldots, \hat{\boldsymbol{g}}_{t,i}^{(M)}]$

$\mathbf{G}_{t,i}^{\text{MC-EF}} = -\frac{1}{M} \hat{\mathbf{G}}_{t,i}^{(1:M)} \hat{\mathbf{G}}_{t,i}^{(1:M)\mathsf{T}}$

Return ($\boldsymbol{g}_{t,i}^{\text{MC}}, \mathbf{G}_{t,i}^{\text{MC-EF}}$)

**Algorithm 5:** MC gradient/Hessian estimator with Empirical Fisher

def grad-LIN-HESS($\boldsymbol{\psi}_{t,i-1}, \ell_t = [], h_t, \mathbf{V}_t, M = []$):
$\boldsymbol{\mu}_{t,i-1} = E[\boldsymbol{\theta}_t | \boldsymbol{\psi}_{t,i-1}]$
$\hat{\boldsymbol{y}}_{t,i} = h_t(\boldsymbol{\mu}_{t,i-1})$
$\mathbf{H}_{t,i} = \frac{\partial h_t}{\partial \boldsymbol{\theta}_t}|_{\boldsymbol{\theta} = \boldsymbol{\mu}_{t,i-1}}$
$\mathbf{R}_{t,i} = \mathbf{V}_t(\boldsymbol{\mu}_{t,i-1})$
$\boldsymbol{g}_{t,i}^{\text{LIN}} = \mathbf{H}_{t,i}^{\mathsf{T}} \mathbf{R}_{t,i}^{-1} (\boldsymbol{y}_t - \hat{\boldsymbol{y}}_{t,i})$
$\mathbf{G}_{t,i}^{\text{LIN-HESS}} = -\mathbf{H}_{t,i}^{\mathsf{T}} \mathbf{R}_{t,i}^{-1} \mathbf{H}_{t,i}$
Return ($\boldsymbol{g}_{t,i}^{\text{LIN}}, \mathbf{G}_{t,i}^{\text{LIN-HESS}}$)

**Algorithm 6:** Linearized gradient/Hessian estimator

def grad-LIN-EF($\boldsymbol{\psi}_{t,i-1}, \ell_t = [], h_t, \mathbf{V}_t, M = []$):
$\boldsymbol{\mu}_{t,i-1} = E[\boldsymbol{\theta}_t | \boldsymbol{\psi}_{t,i-1}]$
$\mathbf{R}_{t,i} = \mathbf{V}_t(\boldsymbol{\mu}_{t,i-1})$
$\boldsymbol{g}_{t,i}^{\text{LIN}} = \nabla_{\boldsymbol{\theta}_t = \boldsymbol{\mu}_{t,i-1}} \left[ -\frac{1}{2} (\boldsymbol{y}_t - h_t(\boldsymbol{\theta}_t))^{\mathsf{T}} \mathbf{R}_{t,i}^{-1} (\boldsymbol{y}_t - h_t(\boldsymbol{\theta}_t)) \right]$
$\mathbf{G}_{t,i}^{\text{LIN-EF}} = -\boldsymbol{g}_{t,i}^{\text{LIN}} (\boldsymbol{g}_{t,i}^{\text{LIN}})^{\mathsf{T}}$
Return ($\boldsymbol{g}_{t,i}^{\text{LIN}}, \mathbf{G}_{t,i}^{\text{LIN-EF}}$)

**Algorithm 7:** Linearized gradient/Hessian estimator with empirical Fisher

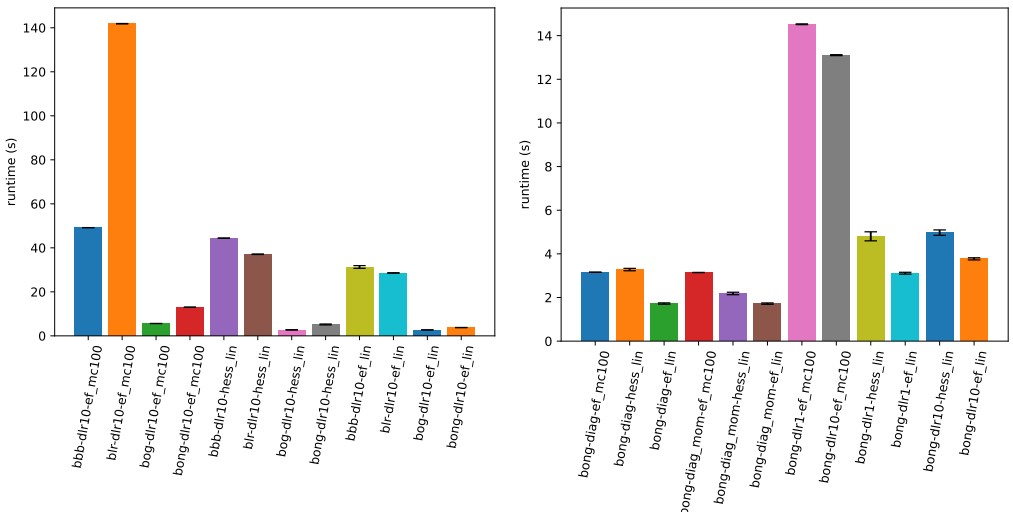

Figure 3: Runtimes for methods on MNIST. Left: Corresponding to Fig. 1 using different algorithms on DLR-10 family. Right: Corresponding to Fig. 2, using BONG on different variational families.

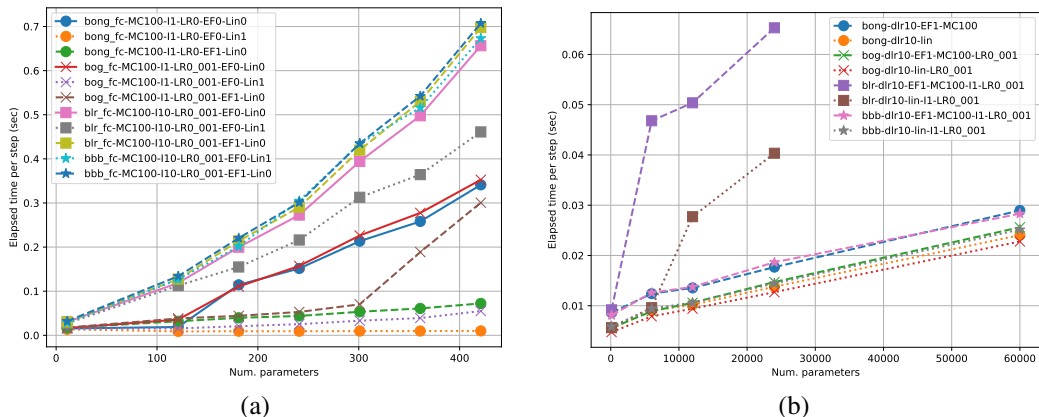

(a)                                                    (b)

Figure 4: Running time (seconds) vs number of parameters $P$ (size of state space) on a synthetic regression problem. For BBB and BLR, we show results using $I = 1$ and $I = 10$ iterations per step. Hessian approximations are denoted as follows: EF0-Lin0 = MC-Hess, EF1-Lin0 = EF-Hess, EF0-Lin1 = Lin-Hess. (a) Full Covariance representation. (b) DLR representation. The BLR plot is truncated due to out of memory problem.

## B.2  Detailed results for CNN on MNIST

Here we report further metrics for the experiments in Figs. 1 and 2. We show 3 approximations to the NLPD: plugin, MC, and Linearized MC.[4] For each of these approximations to the posterior predictive, we also measure the corresponding misclassification rate based on picking the most probable predicted class. Results are shown in Figs. 5 and 6. We see that the plugin and lin-MC approximations are similar, and both are generally much better than standard MC.

Finally, in Fig. 7 and Fig. 8, we report the test-set expected calibration error (ECE) at time steps [250, 500, 1,000, 2,000], computed using 20 bins. Note that among the LIN-HESS variants, BONG-DLR-10

---

[4]Lin-MC is defined in Footnote 2. The motivation for this approximation (from Immer et al. [2021b]) is the following: If we push posterior samples through a nonlinear predictive model, the results can be poor if the samples are far from the mean, but if we linearize the predictive model, extrapolations away from the mean are more sensible. This is true even if the posterior was not explicitly computed using a linear approximation.

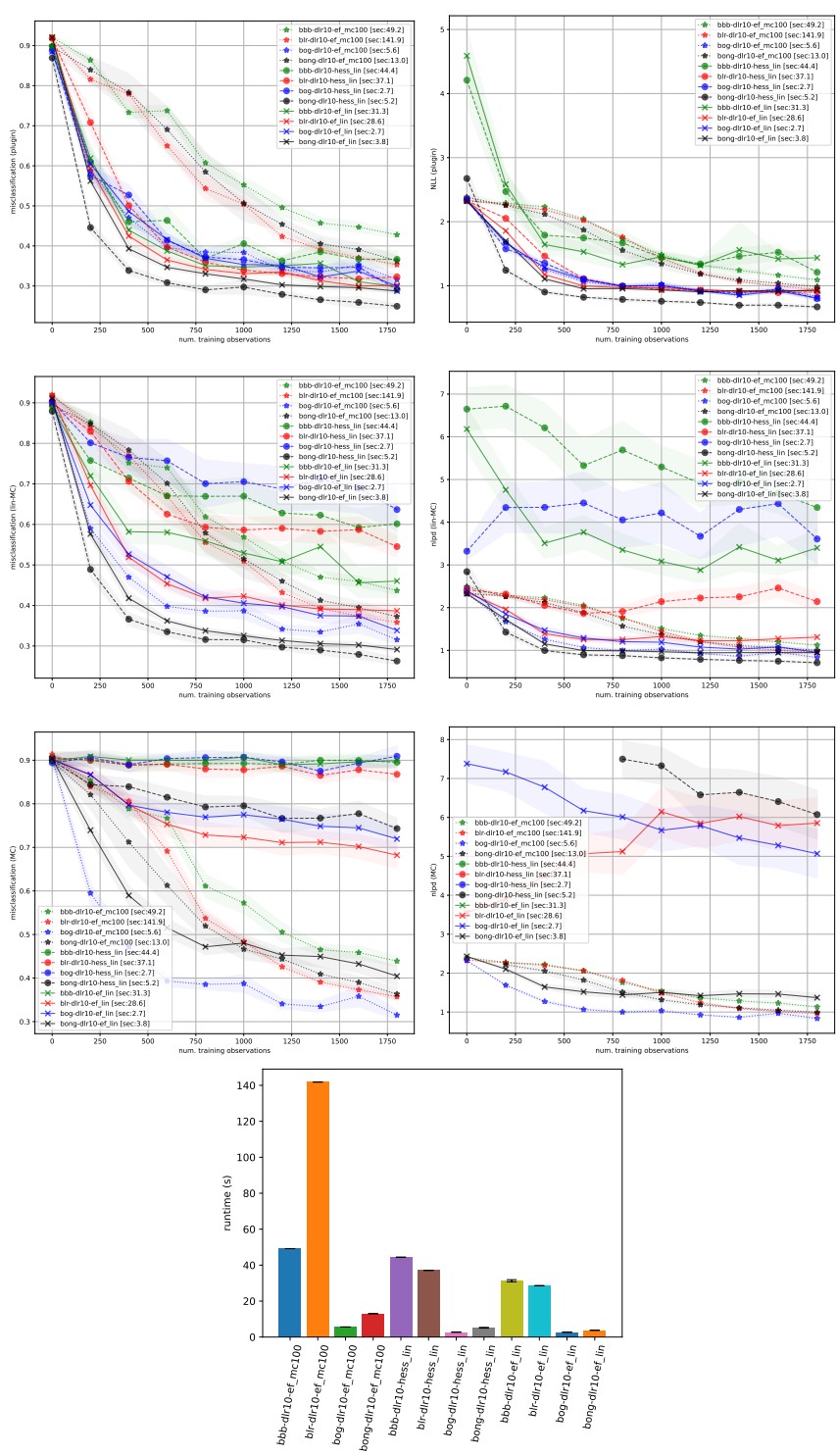

Figure 5: MNIST results for methods using DLR family. Left column shows misclassification rate, right column showns NLL. First row uses plugin approximation to the posterior predictive, second row uses linearized MC approximation, and third row uses standard MC approximation.

method is the most well-calibrated (in addition to exhibiting the strongest plugin and linearized-MC NLPD results), when compared to other DLR-10 methods as well as other BONG variants.

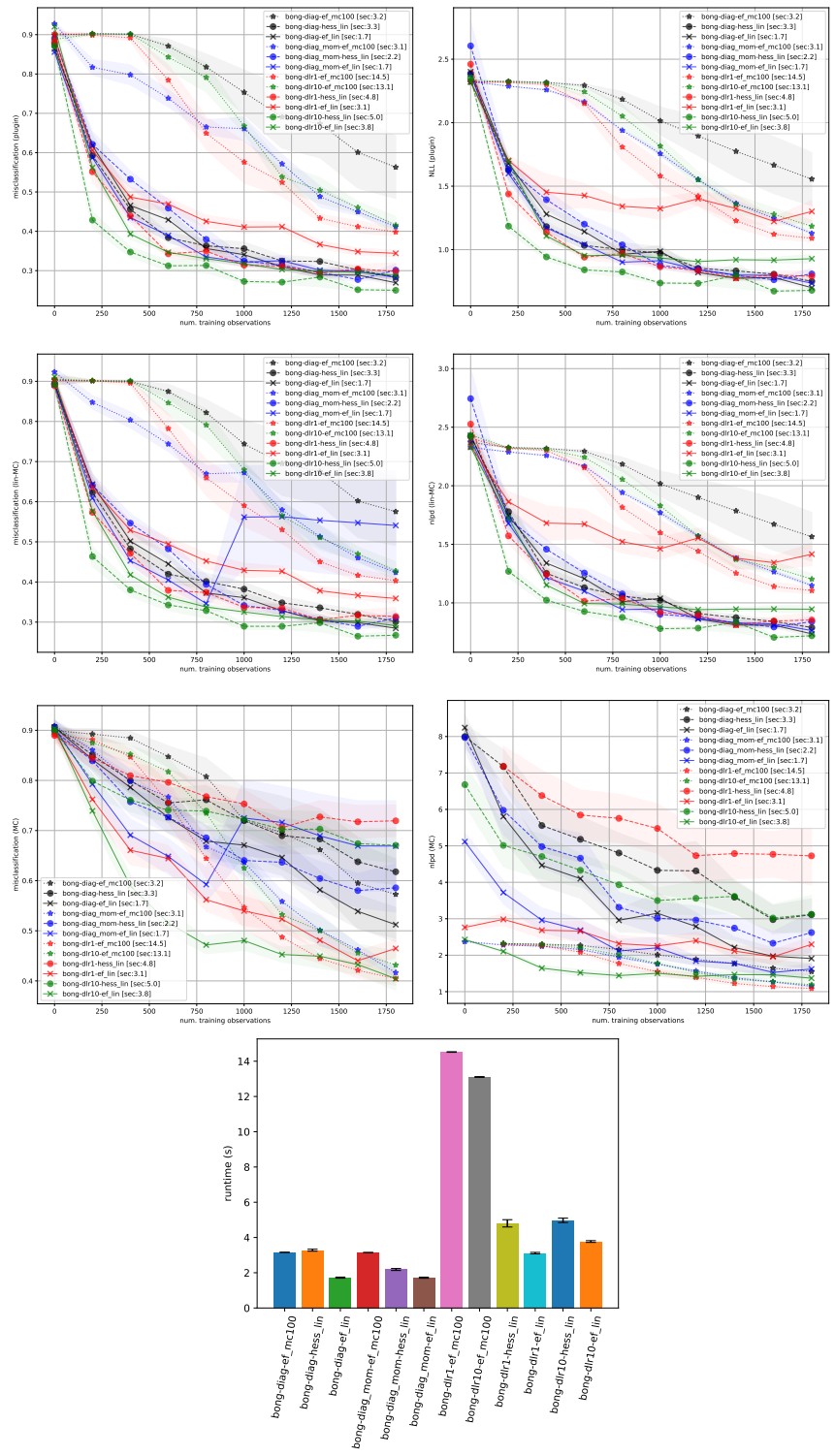

Figure 6: MNIST results for BONG variants. Left column shows misclassification rate, right column shows NLL. First row uses plugin approximation to the posterior predictive, second row uses linearized MC approximation, and third row uses standard MC approximation.

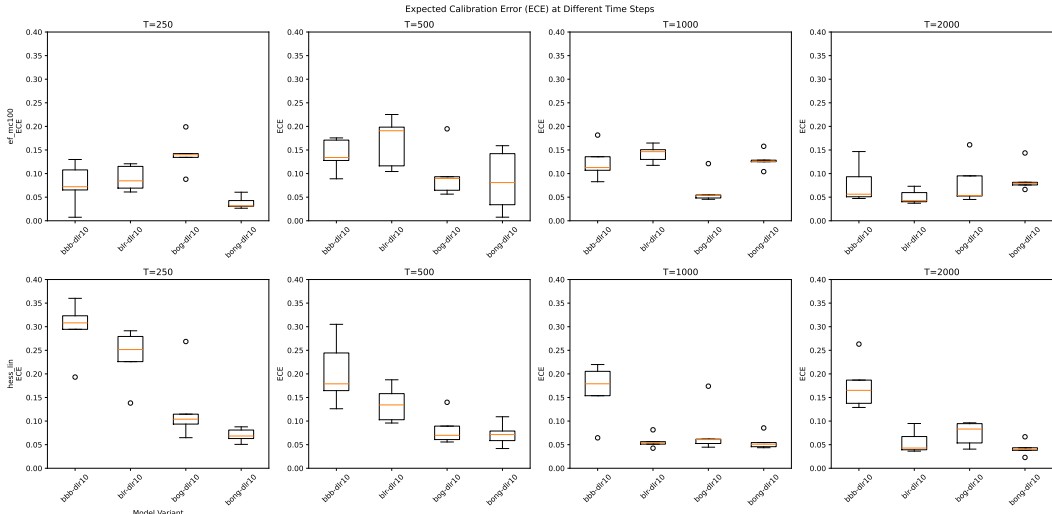

Figure 7: MNIST expected calibration error (ECE) results at selected timesteps for methods using DLR family.

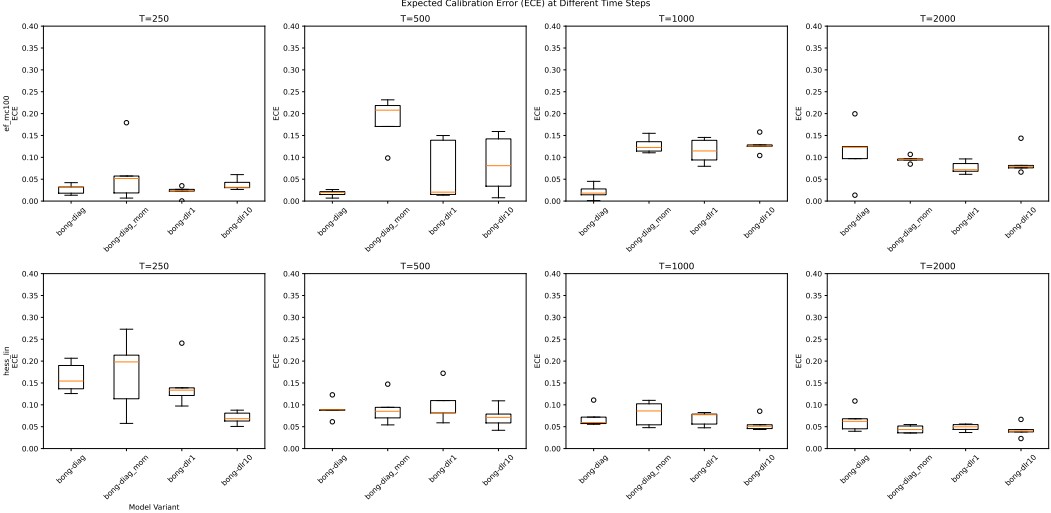

Figure 8: MNIST expected calibration error (ECE) results at selected timesteps for BONG variants. We see that BONG and BOG are both well calibrated, as least when combined with LIN-HESS, with BONG have a slight edge, especially for small sample sizes, where there is more posterior uncertainty.

### B.3 SARCOS dataset

In addition to MNIST, we report experiments on the SARCOS regression dataset ($D = 22$, $N_{\text{train}} = 44{,}484$, $N_{\text{test}} = 4449$, $C = 1$). This dataset derives from an inverse dynamics problem for a seven degrees-of-freedom SARCOS anthropomorphic robot arm. The task is to map from a 21-dimensional input space (7 joint positions, 7 joint velocities, 7 joint accelerations) to the corresponding 7 joint torques. Following Rasmussen and Williams [2006], we pick a single target output dimension, so $C = 1$. The data is from https://gaussianprocess.org/gpml/data/.

We use a small MLP of size 21-20-20-1, so there are $P = 881$ parameters. For optimizing learning rates for SARCOS, we use grid search on NLPD-PI. We fix the variance of the prior belief state

to $\sigma_0^2 = 1.0$, which represents a mild degree of regularization.[5] We fix the observation variance to $R_t = 0.1\hat{R}$, where $\hat{R} = \text{Var}(y_{1:T})$ is the maximum likelihood estimate based on the training sequence; we can think of this as a simple empirical Bayes approximation, and the factor of 0.1 accounts for the fact that the variance of the residuals from the learned model will be smaller than from the unconditional baseline. We focus on DLR approximation of rank 10. This gives similar results to full covariance, but is much faster to compute. We also focus on the plugin approximation to NLPD, since the MC approximation gives much worse results (not shown).

### B.3.1 Comparison of BONG, BLR, BBB and BOG

In Fig. 9 we show the results of using the LIN-HESS approximation. For 1 iteration per step, we see that BONG and BLR are indistinguishable in performance, and BBB and BOG are also indistinguishable, but much worse. For 10 iterations per step, we see that BBB improves significantly, and approaches BONG and BLR. However, BLR and BBB are now about 10 times slower. (In practice, the slowdown is less than 10, due to constant factors of the implementation.) (Note that BONG and BOG always use a single iteration, so their performance does not change.)

In Fig. 10 we show the results of using the MC-EF approximation with $\text{MC} = 100$ samples. The trends are similar to the LIN-HESS case. In particular, for $I = 1$, BONG and BLR are similar, with BONG having a slight edge; and for $I = 10$, BBB catches up with both BONG and BLR, with BOG always in last place. Finally, we see that the performance of MC-EF is slightly worse than LIN-HESS when $I = 1$, but catches up with $I = 10$. However, in larger scale experiments, we usually find that LIN-HESS is significantly better than MC-EF, even with $I = 10$.

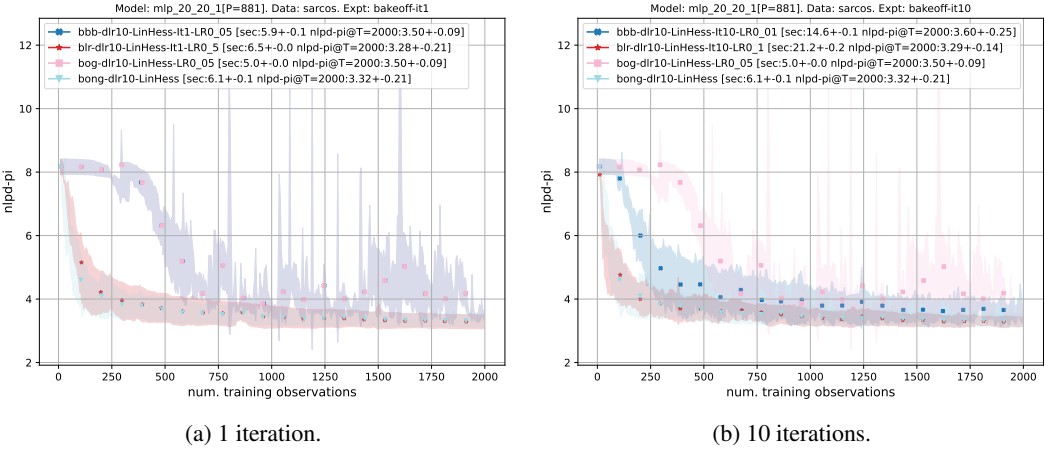

(a) 1 iteration.

(b) 10 iterations.

Figure 9: Predictive performance on SARCOS using MLP 21-20-20-1 with DLR rank 10. Error bars represent $\pm 1$ standard deviation computed from 3 random trials, randomizing over data order and initial state $\boldsymbol{\mu}_0$. (a) We show all 4 algorithms combined with LIN-HESS approximation and $I = 1$. (b) Same as (a) but with $I = 10$.

### B.3.2 Learning rate sensitivity

In Fig. 11 we show the test set performance for BLR (with LIN-HESS approximation) for 5 different learning rates (namely $5 \times 10^{-3}$, $1 \times 10^{-2}$, $5 \times 10^{-2}$, $1 \times 10^{-1}$, and $5 \times 10^{-1}$).

When using 1 iteration per step, the best learning rate is $\alpha = 0.5$, which is also the value chosen based on validation set performance. With this value, BLR matches BONG. For other learning rates, BLR performance is much worse. When using 10 iterations per step, there are several learning rates all of which give performance as good as BONG.

---

[5]This value was based on a small amount of trial and error. Using a smaller value of $\sigma_0$ results in underfitting relative to a linear least squares baseline, and using a much larger value results in unstable posterior covariances, causing the NLPD-MC samples to result in NaNs after a few hundred steps.

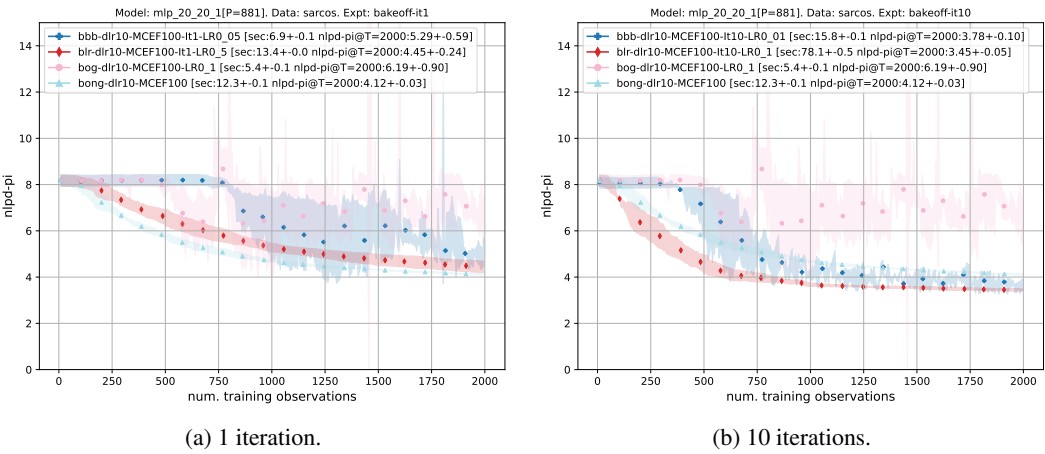

(a) 1 iteration.

(b) 10 iterations.

Figure 10: Same as Fig. 9 except we use MC-EF approximation with MC = 100.

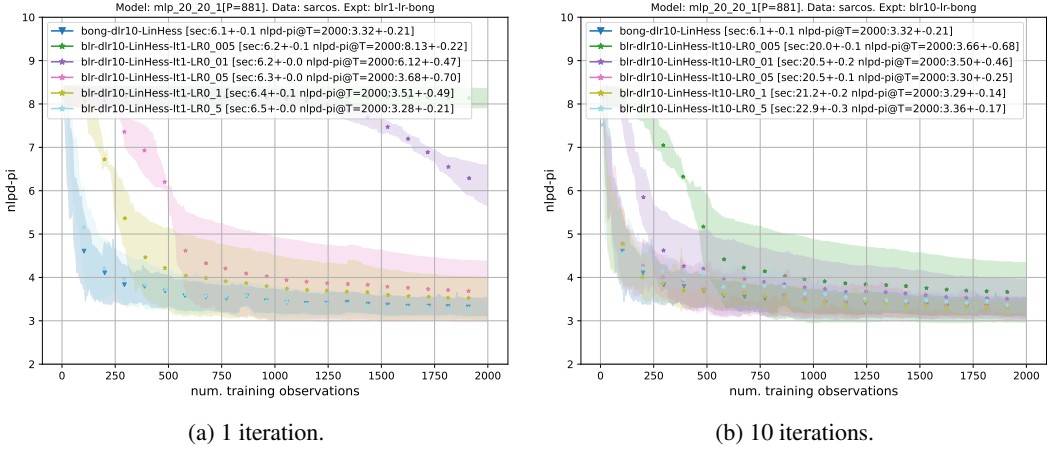

(a) 1 iteration.

(b) 10 iterations.

Figure 11: Same setup as Fig. 9, except now we plot performance for BLR for 5 different learning rates. We also show BONG as a baseline, which uses a fixed learning rate step size of 1.0.

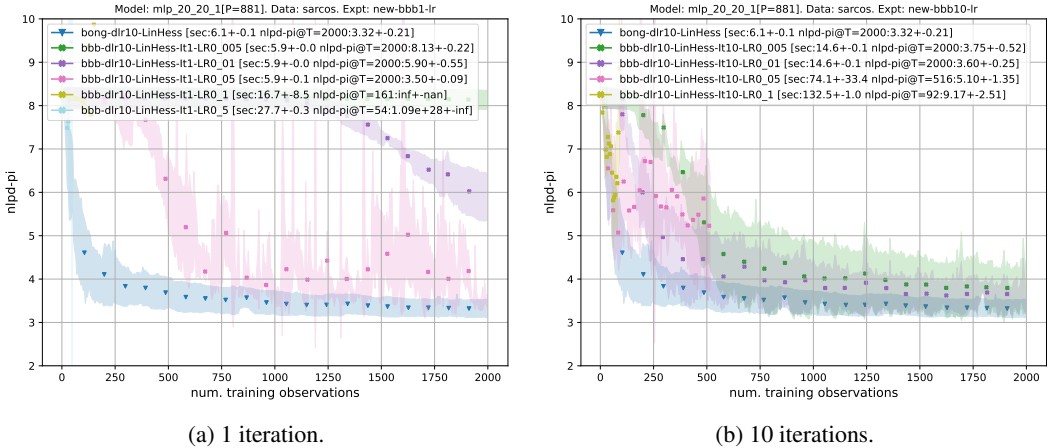

(a) 1 iteration.

(b) 10 iterations.

Figure 12: Same setup as Fig. 9, except now we plot performance for BBB for 5 different learning rates. We also show BONG as a baseline.

In Fig. 12, we show the analogous plot for BBB. When using 1 iteration per step, all learning rates result in poor performance, with many resulting in NaNs. When using 10 iterations per step, there are some learning rates that enable BBB to get close to (but still not match) the performance of BONG.

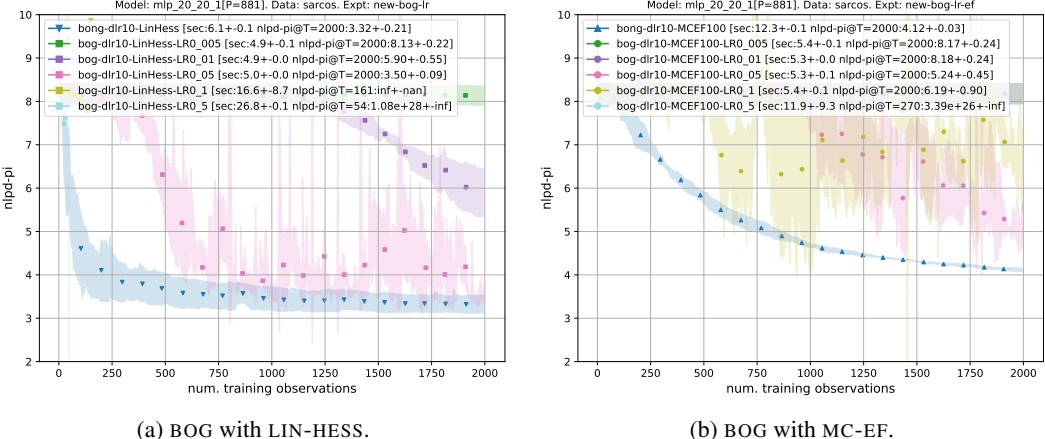

(a) BOG with LIN-HESS.  (b) BOG with MC-EF.

Figure 13: We plot performance for BOG for 5 different learning rates. We also show BONG as a baseline. (a) LIN-HESS approximation. (b) MC-EF approximation.

Finally, in Fig. 13a, we show the analogous plot for BOG with LIN-HESS, and in Fig. 13b with MC-EF, where results are much worse.

Overall we conclude that all the methods (except BONG) are quite sensitive to the learning rate. In our experiments, we pick a value based on performance on a validation set, but in the truly online setting, where there is just a single data stream, picking an optimal learning rate is difficult, which is an additional advantage of BONG.

## C  Proof of Propositions 4.1 and 4.2

*Proposition 4.1.* To ease notation we write the natural parameters of the prior as $\boldsymbol{\psi}_{t|t-1} = [\boldsymbol{\chi}_{t|t-1}; \nu_{t|t-1}]$, which can be interpreted as the prior sufficient statistics and prior sample size. Note that $\boldsymbol{x}_t$ can be omitted as a constant. Based on the prior $q_{\boldsymbol{\psi}_{t|t-1}}(\boldsymbol{\theta}_t)$ the exact posterior is

$$p(\boldsymbol{\theta}_t|\mathcal{D}_t) \propto q_{\boldsymbol{\psi}_{t|t-1}}(\boldsymbol{\theta}_t)\, p_t(\boldsymbol{y}_t|\boldsymbol{\theta}_t) \tag{35}$$

$$\propto \exp\left(\boldsymbol{\chi}_{t|t-1}^\mathsf{T}\boldsymbol{\theta}_t - \nu_{t|t-1}A(\boldsymbol{\theta}_t)\right)\exp\left(\boldsymbol{\theta}_t^\mathsf{T}\boldsymbol{y}_t - A(\boldsymbol{\theta}_t)\right) \tag{36}$$

$$\propto q_{\boldsymbol{\psi}_t}(\boldsymbol{\theta}_t) \tag{37}$$

$$\boldsymbol{\psi}_t = \left[\begin{array}{c} \boldsymbol{\chi}_{t|t-1} + \boldsymbol{y}_t \\ \nu_{t|t-1} + 1 \end{array}\right] \tag{38}$$

For BONG, we first note the dual parameter is given by

$$\boldsymbol{\rho}_{t|t-1} = \mathbb{E}_{\boldsymbol{\theta}_t \sim q_{\boldsymbol{\psi}_{t|t-1}}}[T(\boldsymbol{\theta}_t)] \tag{39}$$

$$= \mathbb{E}_{\boldsymbol{\theta}_t \sim q_{\boldsymbol{\psi}_{t|t-1}}}\left[\begin{array}{c} \boldsymbol{\theta}_t \\ -A(\boldsymbol{\theta}_t) \end{array}\right] \tag{40}$$

Therefore the natural gradient in Eq. (5) is

$$\nabla_{\boldsymbol{\rho}_{t|t-1}}\mathbb{E}_{\boldsymbol{\theta}_t \sim q_{\boldsymbol{\psi}_{t|t-1}}}[\log p\left(\boldsymbol{y}_t|\boldsymbol{\theta}_t\right)] = \nabla_{\boldsymbol{\rho}_{t|t-1}}\mathbb{E}_{\boldsymbol{\theta}_t \sim q_{\boldsymbol{\psi}_{t|t-1}}}\left[\boldsymbol{\theta}_t^\top\boldsymbol{y}_t - A\left(\boldsymbol{\theta}_t\right)\right] \tag{41}$$

$$= \nabla_{\boldsymbol{\rho}_{t|t-1}}\boldsymbol{\rho}_{t|t-1}^\mathsf{T}\left[\begin{array}{c} \boldsymbol{y}_t \\ 1 \end{array}\right] \tag{42}$$

$$= \left[\begin{array}{c} \boldsymbol{y}_t \\ 1 \end{array}\right] \tag{43}$$

Therefore the BONG update yields $\boldsymbol{\psi}_t = [\boldsymbol{\chi}_{t|t-1} + \boldsymbol{y}_t; \nu_{t|t-1} + 1]$ in agreement with Eq. (38).  $\square$

*Proposition 4.2.* The intuition behind this proof is as follows. For the mean update in Eq. (9) $\nabla_{\boldsymbol{\theta}_t}\ell_t$ is linear in $\boldsymbol{\theta}_t$ so the expectation equals the value at the mean. For the covariance update in Eq. (10)

$\nabla^2_{\boldsymbol{\theta}_t} \ell_t$ is independent of $\boldsymbol{\theta}_t$ so we can drop the expectation operator. The tricky part is why we need different linearizations for the Hessians to agree. It has to do with making the Hessian of the NN disappear (as in GGN). In the Gaussian approximation this happens when the predicted mean ($\hat{\boldsymbol{y}}_t = f_t(\boldsymbol{\theta}_t)$) is linear in $\boldsymbol{\theta}_t$. In the plugin approximation it happens when the outcome-dependent part of the loglikelihood ($h_t(\boldsymbol{\theta}_t)^\intercal \boldsymbol{y}_t$) is linear in $\boldsymbol{\theta}_t$. In the latter case the only nonlinear term remaining in the log-likelihood is the log-partition $A$, and the two methods end up agreeing because of the property that the Hessian of the log-partition equals the conditional variance $\mathbf{R}_t$.

Formally, under the linear($h$)-Gaussian approximation in Eqs. (14) and (17) the expected gradient and Hessian can be calculated directly:

$$\mathbb{E}_{\boldsymbol{\theta}_t \sim q_{\boldsymbol{\psi}_{t|t-1}}} \left[ \nabla_{\boldsymbol{\theta}_t} \log \bar{p}_t^{\mathrm{LG}}(\boldsymbol{y}_t | \boldsymbol{\theta}_t)) \right] = \mathbb{E}_{\boldsymbol{\theta}_t \sim q_{\boldsymbol{\psi}_{t|t-1}}} \left[ \mathbf{H}_t^\intercal \mathbf{R}_t^{-1}(\boldsymbol{y}_t - \hat{\boldsymbol{y}}_t - \mathbf{H}_t(\boldsymbol{\theta}_t - \boldsymbol{\mu}_{t|t-1})) \right] \quad (44)$$

$$= \mathbf{H}_t^\intercal \mathbf{R}_t^{-1}(\boldsymbol{y}_t - \hat{\boldsymbol{y}}_t) \quad (45)$$

$$\mathbb{E}_{\boldsymbol{\theta}_t \sim q_{\boldsymbol{\psi}_{t|t-1}}} \left[ \nabla^2_{\boldsymbol{\theta}_t} \log \bar{p}_t^{\mathrm{LG}}(\boldsymbol{y}_t | \boldsymbol{\theta}_t) \right] = \mathbb{E}_{\boldsymbol{\theta}_t \sim q_{\boldsymbol{\psi}_{t|t-1}}} \left[ -\mathbf{H}_t^\intercal \mathbf{R}_t^{-1} \mathbf{H}_t \right] \quad (46)$$

$$= -\mathbf{H}_t^\intercal \mathbf{R}_t^{-1} \mathbf{H}_t \quad (47)$$

For the linear($f$)-delta approximation we use the properties of exponential families that (1) the gradient of the log-partition $A$ with respect to the natural parameter $f_t(\boldsymbol{\theta}_t)$ equals the expectation parameter $h_t(\boldsymbol{\theta}_t)$, (2) the Hessian of the log-partition with respect to the natural parameter equals the conditional variance, and consequently (3) the Jacobian of the expectation parameter with respect to the natural parameter equals the conditional variance $\mathbf{R}_t$:

$$\nabla_{\boldsymbol{\eta} = f_t(\boldsymbol{\mu}_{t|t-1})} A(\boldsymbol{\eta}) = h_t(\boldsymbol{\mu}_{t|t-1}) = \hat{\boldsymbol{y}} \quad (48)$$

$$\nabla^2_{\boldsymbol{\eta} = f_t(\boldsymbol{\mu}_{t|t-1})} A(\boldsymbol{\eta}) = \mathbb{V}\left[ \boldsymbol{y}_t | \boldsymbol{\theta}_t = \boldsymbol{\mu}_{t|t-1} \right] = \mathbf{R}_t \quad (49)$$

$$\frac{\partial h_t(\boldsymbol{\theta}_t)}{\partial f_t(\boldsymbol{\theta}_t)}_{|\boldsymbol{\theta}_t = \boldsymbol{\mu}_{t|t-1}} = \mathbf{R}_t \quad (50)$$

The last of these implies $\mathbf{F}_t = \mathbf{R}_t^{-1} \mathbf{H}_t$. Therefore the expected gradient and Hessian can be calculated as

$$\mathbb{E}_{\boldsymbol{\theta}_t \sim \delta_{\boldsymbol{\mu}_{t|t-1}}} \left[ \nabla_{\boldsymbol{\theta}_t} \log \bar{p}_t^{\mathrm{LD}}(\boldsymbol{y}_t | \boldsymbol{\theta}_t) \right] = \nabla_{\boldsymbol{\theta}_t = \boldsymbol{\mu}_{t|t-1}} \log \bar{p}_t^{\mathrm{LD}}(\boldsymbol{y}_t | \boldsymbol{\theta}_t) \quad (51)$$

$$= \mathbf{F}_t^\intercal \boldsymbol{y}_t - \mathbf{F}_t^\intercal \nabla_{\boldsymbol{\eta} = f_t(\boldsymbol{\mu}_{t|t-1})} A(\boldsymbol{\eta}) \quad (52)$$

$$= \mathbf{F}_t^\intercal (\boldsymbol{y}_t - \hat{\boldsymbol{y}}_t) \quad (53)$$

$$= \mathbf{H}_t^\intercal \mathbf{R}_t^{-1}(\boldsymbol{y}_t - \hat{\boldsymbol{y}}_t) \quad (54)$$

$$\mathbb{E}_{\boldsymbol{\theta}_t \sim \delta_{\boldsymbol{\mu}_{t|t-1}}} \left[ \nabla^2_{\boldsymbol{\theta}_t} \log \bar{p}_t^{\mathrm{LD}}(\boldsymbol{y}_t | \boldsymbol{\theta}_t) \right] = \nabla^2_{\boldsymbol{\theta}_t = \boldsymbol{\mu}_{t|t-1}} \log \bar{p}_t^{\mathrm{LD}}(\boldsymbol{y}_t | \boldsymbol{\theta}_t) \quad (55)$$

$$= -\mathbf{F}_t^\intercal \left( \nabla^2_{\boldsymbol{\eta} = f_t(\boldsymbol{\mu}_{t|t-1})} A(\boldsymbol{\eta}) \right) \mathbf{F}_t \quad (56)$$

$$= -\mathbf{F}_t^\intercal \mathbf{R}_t \mathbf{F}_t \quad (57)$$

$$= -\mathbf{H}_t^\intercal \mathbf{R}_t^{-1} \mathbf{H}_t \quad (58)$$

$\square$

# D  Mirror descent formulation

In this section we give a more detailed derivation of BONG as mirror descent and use this to give two alternative interpretations of how BONG approximates exact VB: (1) by approximating the expected NLL as linear in the expectation parameter $\boldsymbol{\rho}$, or (2) by replacing an implicit update with an explicit one.

Assume the variational family is an exponential one as introduced at the end of Section 3, with natural and dual parameters $\boldsymbol{\psi}$ and $\boldsymbol{\rho}$, sufficient statistics $T(\boldsymbol{\theta})$, and log-partition $\Phi(\boldsymbol{\psi})$:

$$q_{\boldsymbol{\psi}}(\boldsymbol{\theta}) = \exp\left( \boldsymbol{\psi}^\intercal T(\boldsymbol{\theta}) - \Phi(\boldsymbol{\psi}) \right) \quad (59)$$

$$\boldsymbol{\rho} = \mathbb{E}_{\boldsymbol{\theta} \sim q_{\boldsymbol{\psi}}}[T(\boldsymbol{\theta})] \quad (60)$$

We first review how NGD on an exponential family is a special case of mirror descent [Khan and Rue, 2023, Martens, 2020]. The mirror map is the gradient of the log-partition, which satisfies the thermodynamic identity

$$\boldsymbol{\rho} = \nabla\Phi(\boldsymbol{\psi}) \tag{61}$$

This is a bijection when $\Phi$ is convex (which includes the cases we study), so we can implicitly treat $\boldsymbol{\psi}$ and $\boldsymbol{\rho}$ as functions of each other. Given a loss function $L(\boldsymbol{\psi})$, MD iteratively solves the local optimization problem

$$\boldsymbol{\psi}_{i+1} = \arg\min_{\boldsymbol{\psi}} \langle \nabla_{\boldsymbol{\rho}_i} L(\boldsymbol{\psi}_i), \boldsymbol{\rho} \rangle + \frac{1}{\alpha}\mathbb{D}_\Phi(\boldsymbol{\psi}_i, \boldsymbol{\psi}) \tag{62}$$

The first term is a linear (in $\boldsymbol{\rho}$) approximation of $L$ about the previous iteration $\boldsymbol{\psi}_i$ and the second term is the Bregman divergence

$$\mathbb{D}_\Phi(\boldsymbol{\psi}_i, \boldsymbol{\psi}_{i+1}) = \Phi(\boldsymbol{\psi}_i) - \Phi(\boldsymbol{\psi}_{i+1}) - (\boldsymbol{\psi}_i - \boldsymbol{\psi}_{i+1})^\mathsf{T} \boldsymbol{\rho}_{i+1} \tag{63}$$

The Bregman divergence acts as a regularizer toward $\boldsymbol{\psi}_i$ and captures the intrinsic geometry of the parameter space because of its equivalence with the (reverse) KL divergence

$$D_{\mathbb{KL}}\left(q_{\boldsymbol{\psi}_{i+1}}|q_{\boldsymbol{\psi}_i}\right) = \mathbb{E}_{\boldsymbol{\theta}\sim q_{\boldsymbol{\psi}_{i+1}}}[(\boldsymbol{\psi}_{i+1} - \boldsymbol{\psi}_i)^\mathsf{T} T(\boldsymbol{\theta}) + \Phi(\boldsymbol{\psi}_i) - \Phi(\boldsymbol{\psi}_{i+1})] \tag{64}$$

$$= \mathbb{D}_\Phi(\boldsymbol{\psi}_i, \boldsymbol{\psi}_{i+1}) \tag{65}$$

Importantly, this recursive regularizer is not part of the loss and serves only to define an iterated algorithm that converges to a local minimum of $L$. Solving Eq. (62) by differentiating by $\boldsymbol{\rho}$ yields the MD update

$$\boldsymbol{\psi}_{i+1} = \boldsymbol{\psi}_i - \alpha\nabla_{\boldsymbol{\rho}_i} L(\boldsymbol{\psi}_i) \tag{66}$$

Because the Fisher matrix for an exponential family is $\mathbf{F}_{\boldsymbol{\psi}} = \partial\boldsymbol{\rho}/\partial\boldsymbol{\psi}$, this is equivalent to NGD with respect to $\boldsymbol{\psi}$. Khan and Rue [2023] offer this as a derivation of the BLR, when $L(\boldsymbol{\psi})$ is taken to be the variational loss from Eq. (1).

By applying this analysis to the online setting, our approach can be seen to follow from two insights. First, the online variational loss in Eq. (2) already includes KL divergence from the previous step, so we do not need the artificial regularizer in Eq. (62). That is, if we start from the online variational problem in Eq. (4) and define $L_t(\boldsymbol{\psi})$ as the expected NLL,

$$L_t(\boldsymbol{\psi}) = -\mathbb{E}_{\boldsymbol{\theta}_t \sim q_{\boldsymbol{\psi}}}[\log p(\boldsymbol{y}_t|f_t(\boldsymbol{\theta}_t))] \tag{67}$$

then replacing $L_t(\boldsymbol{\psi})$ with a linear approximation based at $\boldsymbol{\psi}_{t|t-1}$ and applying Eq. (65) leads to

$$\boldsymbol{\psi}_t = \arg\min_{\boldsymbol{\psi}} \langle \nabla_{\boldsymbol{\rho}_{t|t-1}} L_t(\boldsymbol{\psi}_{t|t-1}), \boldsymbol{\rho} \rangle + \mathbb{D}_\Phi(\boldsymbol{\psi}_{t|t-1}, \boldsymbol{\psi}) \tag{68}$$

By comparing to Eq. (62) we see this defines an MD algorithm with unit learning rate that works in a single step rather than by iterating. Paralleling the derivation of Eq. (66) from Eq. (62) we get

$$\boldsymbol{\psi}_t = \boldsymbol{\psi}_{t|t-1} - \nabla_{\boldsymbol{\rho}_{t|t-1}} L_t(\boldsymbol{\psi}_{t|t-1}) \tag{69}$$

which matches the BONG update in Eq. (5). Thus BONG can be seen as an approximate solution of the online variational problem in Eq. (4) based on linearizing the expected NLL wrt $\boldsymbol{\rho}$. (Note this is different from the assumption underlying BONG-LIN that $f_t(\boldsymbol{\theta}_t)$ or $h_t(\boldsymbol{\theta}_t)$ is linear in $\boldsymbol{\theta}_t$.)

Second, in the conjugate case, this linearity assumption is true: $L_t$ is linear in $\boldsymbol{\rho}$ (see proof of Proposition 4.1). Therefore 68 is equivalent to solving Eq. (4) exactly:

$$\boldsymbol{\psi}_t = \arg\min_{\boldsymbol{\psi}} L_t(\boldsymbol{\psi}) + \mathbb{D}_\Phi(\boldsymbol{\psi}_{t|t-1}, \boldsymbol{\psi}) \tag{70}$$

This recapitulates Proposition 4.1 that BONG is Bayes optimal in the conjugate case. In general the exact solution to Eq. (70) is

$$\boldsymbol{\psi}_t = \boldsymbol{\psi}_{t|t-1} - \nabla_{\boldsymbol{\rho}_t} L_t(\boldsymbol{\psi}_t) \tag{71}$$

This is an implicit update because the gradient is evaluated at the (unknown) posterior, whereas Eq. (69) is an explicit update because it evaluates the gradient at the prior. (In the Gaussian case these can be shown to match the implicit and explicit RVGA updates of [Lambert et al., 2021].) Therefore BONG can be also interpreted as an approximation of exact VB that replaces the implicit update, Eq. (71), with an explicit update, Eq. (69).

# E  Derivations

This section derives the update equations for all 80 algorithms we investigate (Table 3 plus the MC-HESS and LIN-EF variants). In Appendix E.6 we also translate the BLR algorithms from our online setting back to the batch setting used in Khan and Rue [2023].

For an exponential variational family with natural parameters $\boldsymbol{\psi}$ and dual parameters $\boldsymbol{\rho}$, we can derive all 16 methods (BONG, BLR, BOG, BBB under all four Hessian approximations) from four quantities:

$$\nabla_{\boldsymbol{\rho}_{t,i-1}} \mathbb{E}_{\boldsymbol{\theta}_t \sim q_{\boldsymbol{\psi}_{t,i-1}}} [\log p\left(\boldsymbol{y}_t | f_t\left(\boldsymbol{\theta}_t\right)\right)] \tag{72}$$

$$\nabla_{\boldsymbol{\rho}_{t,i-1}} D_{\mathbb{KL}} \left(q_{\boldsymbol{\psi}_{t,i-1}} | q_{\boldsymbol{\psi}_{t|t-1}}\right) \tag{73}$$

$$\nabla_{\boldsymbol{\psi}_{t,i-1}} \mathbb{E}_{\boldsymbol{\theta}_t \sim q_{\boldsymbol{\psi}_{t,i-1}}} [\log p\left(\boldsymbol{y}_t | f_t\left(\boldsymbol{\theta}_t\right)\right)] \tag{74}$$

$$\nabla_{\boldsymbol{\psi}_{t,i-1}} D_{\mathbb{KL}} \left(q_{\boldsymbol{\psi}_{t,i-1}} | q_{\boldsymbol{\psi}_{t|t-1}}\right) \tag{75}$$

The NGD methods (BONG and BLR) use gradients with respect to $\boldsymbol{\rho}_{t,t-i}$ while the GD methods (BOG and BBB) use gradients with respect to $\boldsymbol{\psi}_{t,i-1}$. For BONG and BOG the $D_{\mathbb{KL}}$ term is not relevant, and there is no inner loop so $\boldsymbol{\psi}_{t,i-1} = \boldsymbol{\psi}_{t|t-1}$ and $\boldsymbol{g}_{t,i} = \boldsymbol{g}_t$, $\mathbf{G}_{t,i} = \mathbf{G}_t$.

When $\boldsymbol{\psi}$ is not the natural parameter of an exponential family we must explicitly compute the inverse-Fisher preconditioner for the NGD methods. Therefore the updates can be derived from these three quantities:

$$\mathbf{F}_{\boldsymbol{\psi}_{t,i-1}} \tag{76}$$

$$\nabla_{\boldsymbol{\psi}_{t,i-1}} \mathbb{E}_{\boldsymbol{\theta}_t \sim q_{\boldsymbol{\psi}_{t,i-1}}} [\log p\left(\boldsymbol{y}_t | f_t\left(\boldsymbol{\theta}_t\right)\right)] \tag{77}$$

$$\nabla_{\boldsymbol{\psi}_{t,i-1}} D_{\mathbb{KL}} \left(q_{\boldsymbol{\psi}_{t,i-1}} | q_{\boldsymbol{\psi}_{t|t-1}}\right) \tag{78}$$

We will frequently use Bonnet's and Price's theorems [Bonnet, 1964, Price, 1958]

$$\nabla_{\boldsymbol{\mu}_{t,i-1}} \mathbb{E}_{\mathcal{N}(\boldsymbol{\mu}_{t,i-1}, \boldsymbol{\Sigma}_{t,i-1})} [\log p\left(\boldsymbol{y}_t | f_t\left(\boldsymbol{\theta}_t\right)\right)] = \mathbb{E}_{\mathcal{N}(\boldsymbol{\mu}_{t,i-1}, \boldsymbol{\Sigma}_{t,i-1})} [\nabla_{\boldsymbol{\theta}_t} \log p\left(\boldsymbol{y}_t | f_t\left(\boldsymbol{\theta}_t\right)\right)] \tag{79}$$

$$= \boldsymbol{g}_{t,i} \tag{80}$$

$$\nabla_{\boldsymbol{\Sigma}_{t,i-1}} \mathbb{E}_{\mathcal{N}(\boldsymbol{\mu}_{t,i-1}, \boldsymbol{\Sigma}_{t,i-1})} [\log p\left(\boldsymbol{y}_t | f_t\left(\boldsymbol{\theta}_t\right)\right)] = \tfrac{1}{2} \mathbb{E}_{\mathcal{N}(\boldsymbol{\mu}_{t,i-1}, \boldsymbol{\Sigma}_{t,i-1})} [\nabla^2_{\boldsymbol{\theta}_t} \log p\left(\boldsymbol{y}_t | f_t\left(\boldsymbol{\theta}_t\right)\right)] \tag{81}$$

$$= \tfrac{1}{2} \mathbf{G}_{t,i} \tag{82}$$

For diagonal Gaussians with covariance $\text{Diag}\left(\boldsymbol{\sigma}^2\right)$, Price's theorem also implies[6]

$$\nabla_{\boldsymbol{\sigma}^2_{t,i-1}} \mathbb{E}_{\boldsymbol{\theta}_t \sim \mathcal{N}(\boldsymbol{\mu}_{t,i-1}, \boldsymbol{\Sigma}_{t,i-1})} [\log p\left(\boldsymbol{y}_t | f_t\left(\boldsymbol{\theta}_t\right)\right)] = \tfrac{1}{2} \text{diag}\left(\mathbf{G}_{t,i}\right) \tag{83}$$

Update equations for MC-HESS, MC-EF and LIN-EF methods are displayed together in the subsections that follow, because for the most part they differ only in the choice of $\mathbf{G}_t^{\text{MC-HESS}}$, $\mathbf{G}_t^{\text{MC-EF}}$ or $\mathbf{G}_t^{\text{LIN-EF}}$ to approximate $\mathbf{G}_t$ and $\boldsymbol{g}_t^{\text{MC}}$ or $\boldsymbol{g}_t^{\text{LIN}}$ to approximate $\boldsymbol{g}_t$. We note cases where decomposing $\mathbf{G}_t^{\text{MC-EF}} = -\frac{1}{M} \hat{\mathbf{G}}_t^{(1:M)} \hat{\mathbf{G}}_t^{(1:M)\mathsf{T}}$ or $\mathbf{G}_t^{\text{LIN-EF}} = -\boldsymbol{g}_t^{\text{LIN}} (\boldsymbol{g}_t^{\text{LIN}})^\mathsf{T}$ allows a more efficient update.

We derive updates for BONG-LIN-HESS and BOG-LIN-HESS from the corresponding BONG-MC-HESS and BOG-MC-HESS updates using Proposition 4.2 which entails substituting

$$\boldsymbol{g}_t \rightarrow \mathbf{H}_t^\mathsf{T} \mathbf{R}_t^{-1} (\boldsymbol{y}_t - \hat{\boldsymbol{y}}_t) \tag{84}$$

$$\mathbf{G}_t \rightarrow -\mathbf{H}_t^\mathsf{T} \mathbf{R}_t^{-1} \mathbf{H}_t \tag{85}$$

For the algorithms with inner loops (BLR and BBB) we adapt the notation of Section 4.4 as follows:

$$\boldsymbol{y}_{t,i} = h_t\left(\boldsymbol{\mu}_{t,i-1}\right) \tag{86}$$

$$\mathbf{H}_{t,i} = \frac{\partial h_t}{\partial \boldsymbol{\theta}_t}_{|\boldsymbol{\theta} = \boldsymbol{\mu}_{t,i-1}} \tag{87}$$

$$\mathbf{R}_{t,i} = \mathbb{V}\left[\boldsymbol{y}_t | \boldsymbol{\theta}_t = \boldsymbol{\mu}_{t,i-1}\right] \tag{88}$$

$$\boldsymbol{g}_{t,i} = \mathbf{H}_{t,i}^\mathsf{T} \mathbf{R}_{t,i}^{-1} \left(\boldsymbol{y}_t - \hat{\boldsymbol{y}}_{t,i}\right) \tag{89}$$

$$\mathbf{G}_{t,i} = -\mathbf{H}_{t,i}^\mathsf{T} \mathbf{R}_{t,i}^{-1} \mathbf{H}_{t,i} \tag{90}$$

---

[6]We use $\text{diag}(\mathbf{A})$ to denote the vector of diagonal elements of matrix $\mathbf{A}$ and $\text{Diag}(\boldsymbol{v})$ to denote the matrix whose diagonal entries are $\boldsymbol{v}$ and off-diagonal entries are 0.

This corresponds to basing the linear($f$)-Gaussian and linear($h$)-delta approximations at $\boldsymbol{\mu}_{t,i-1}$ instead of $\boldsymbol{\mu}_{t|t-1}$. Thus the updates for BLR-LIN-HESS and BBB-LIN-HESS are obtained by substituting

$$\boldsymbol{g}_{t,i} \rightarrow \mathbf{H}_{t,i}^{\mathsf{T}} \mathbf{R}_{t,i}^{-1} (\boldsymbol{y}_t - \hat{\boldsymbol{y}}_{t,i}) \tag{91}$$

$$\mathbf{G}_{t,i} \rightarrow -\mathbf{H}_{t,i}^{\mathsf{T}} \mathbf{R}_{t,i}^{-1} \mathbf{H}_{t,i} \tag{92}$$

### E.1 Full covariance Gaussian, natural parameters

The natural and dual parameters for a general Gaussian are given by

$$\boldsymbol{\psi}_{t,i-1}^{(1)} = \boldsymbol{\Sigma}_{t,i-1}^{-1} \boldsymbol{\mu}_{t,i-1} \qquad \boldsymbol{\rho}_{t,i-1}^{(1)} = \boldsymbol{\mu}_{t,i-1} \tag{93}$$

$$\boldsymbol{\psi}_{t,i-1}^{(2)} = -\tfrac{1}{2} \boldsymbol{\Sigma}_{t,i-1}^{-1} \qquad \boldsymbol{\rho}_{t,i-1}^{(2)} = \boldsymbol{\mu}_{t,i-1} \boldsymbol{\mu}_{t,i-1}^{\mathsf{T}} + \boldsymbol{\Sigma}_{t,i-1} \tag{94}$$

Inverting these relationships gives

$$\boldsymbol{\mu}_{t,i-1} = -\tfrac{1}{2} \boldsymbol{\psi}_{t,i-1}^{(2)-1} \boldsymbol{\psi}_{t,i-1}^{(1)} = \boldsymbol{\rho}_{t,i-1}^{(1)} \tag{95}$$

$$\boldsymbol{\Sigma}_{t,i-1} = -\tfrac{1}{2} \boldsymbol{\psi}_{t,i-1}^{(2)-1} \qquad = \boldsymbol{\rho}_{t,i-1}^{(2)} - \boldsymbol{\rho}_{t,i-1}^{(1)} \boldsymbol{\rho}_{t,i-1}^{(1)\mathsf{T}} \tag{96}$$

The KL divergence in the VI loss is

$$D_{\mathbb{KL}}\big(q_{\boldsymbol{\psi}_{t,i-1}}|q_{\boldsymbol{\psi}_{t|t-1}}\big) = \tfrac{1}{2}\big(\boldsymbol{\mu}_{t,i-1} - \boldsymbol{\mu}_{t|t-1}\big)^{\mathsf{T}} \boldsymbol{\Sigma}_{t|t-1}^{-1}\big(\boldsymbol{\mu}_{t,i-1} - \boldsymbol{\mu}_{t|t-1}\big)$$

$$+ \tfrac{1}{2} Tr\left(\boldsymbol{\Sigma}_{t|t-1}^{-1} \boldsymbol{\Sigma}_{t,i-1}\right) - \tfrac{1}{2}\log|\boldsymbol{\Sigma}_{t,i-1}|) + \text{const} \tag{97}$$

with gradients

$$\nabla_{\boldsymbol{\mu}_{t,i-1}} D_{\mathbb{KL}}\big(q_{\boldsymbol{\psi}_{t,i-1}}|q_{\boldsymbol{\psi}_{t|t-1}}\big) = \boldsymbol{\Sigma}_{t|t-1}^{-1}\big(\boldsymbol{\mu}_{t,i-1} - \boldsymbol{\mu}_{t|t-1}\big) \tag{98}$$

$$\nabla_{\boldsymbol{\Sigma}_{t,i-1}} D_{\mathbb{KL}}\big(q_{\boldsymbol{\psi}_{t,i-1}}|q_{\boldsymbol{\psi}_{t|t-1}}\big) = \tfrac{1}{2}\left(\boldsymbol{\Sigma}_{t|t-1}^{-1} - \boldsymbol{\Sigma}_{t,i-1}^{-1}\right) \tag{99}$$

Following Appendix C of [Khan et al., 2018a], for any scalar function $\ell$ the chain rule gives

$$\nabla_{\boldsymbol{\rho}_{t,i-1}^{(1)}} \ell = \frac{\partial \boldsymbol{\mu}_{t,i-1}}{\partial \boldsymbol{\rho}_{t,i-1}^{(1)}} \nabla_{\boldsymbol{\mu}_{t,i-1}} \ell + \frac{\partial \boldsymbol{\Sigma}_{t,i-1}}{\partial \boldsymbol{\rho}_{t,i-1}^{(1)}} \nabla_{\boldsymbol{\Sigma}_{t,i-1}} \ell \tag{100}$$

$$= \nabla_{\boldsymbol{\mu}_{t,i-1}} \ell - 2\left(\nabla_{\boldsymbol{\Sigma}_{t,i-1}} \ell\right) \boldsymbol{\mu}_{t,i-1} \tag{101}$$

$$\nabla_{\boldsymbol{\rho}_{t,i-1}^{(2)}} \ell = \frac{\partial \boldsymbol{\mu}_{t,i-1}}{\partial \boldsymbol{\rho}_{t,i-1}^{(2)}} \nabla_{\boldsymbol{\mu}_{t,i-1}} \ell + \frac{\partial \boldsymbol{\Sigma}_{t,i-1}}{\partial \boldsymbol{\rho}_{t,i-1}^{(2)}} \nabla_{\boldsymbol{\Sigma}_{t,i-1}} \ell \tag{102}$$

$$= \nabla_{\boldsymbol{\Sigma}_{t,i-1}} \ell \tag{103}$$

Therefore

$$\nabla_{\boldsymbol{\rho}_{t,i-1}^{(1)}} \mathbb{E}_{\boldsymbol{\theta}_t \sim q_{\boldsymbol{\psi}_{t,i-1}}}[\log p\left(\boldsymbol{y}_t|f_t\left(\boldsymbol{\theta}_t\right)\right)] = \boldsymbol{g}_{t,i} - \mathbf{G}_{t,i}\boldsymbol{\mu}_{t,i-1} \tag{104}$$

$$\nabla_{\boldsymbol{\rho}_{t,i-1}^{(2)}} \mathbb{E}_{\boldsymbol{\theta}_t \sim q_{\boldsymbol{\psi}_{t,i-1}}}[\log p\left(\boldsymbol{y}_t|f_t\left(\boldsymbol{\theta}_t\right)\right)] = \tfrac{1}{2}\mathbf{G}_{t,i} \tag{105}$$

and

$$\nabla_{\boldsymbol{\rho}_{t,i-1}^{(1)}} D_{\mathbb{KL}}\big(q_{\boldsymbol{\psi}_{t,i-1}}|q_{\boldsymbol{\psi}_{t|t-1}}\big) = \boldsymbol{\Sigma}_{t,i-1}^{-1}\boldsymbol{\mu}_{t,i-1} - \boldsymbol{\Sigma}_{t|t-1}^{-1}\boldsymbol{\mu}_{t|t-1} \tag{106}$$

$$\nabla_{\boldsymbol{\rho}_{t,i-1}^{(2)}} D_{\mathbb{KL}}\big(q_{\boldsymbol{\psi}_{t,i-1}}|q_{\boldsymbol{\psi}_{t|t-1}}\big) = \tfrac{1}{2}\left(\boldsymbol{\Sigma}_{t|t-1}^{-1} - \boldsymbol{\Sigma}_{t,i-1}^{-1}\right) \tag{107}$$

Following the same approach for $\boldsymbol{\psi}$ gives

$$\nabla_{\boldsymbol{\psi}_{t,i-1}^{(1)}} \ell = \frac{\partial \boldsymbol{\mu}_{t,i-1}}{\partial \boldsymbol{\psi}_{t,i-1}^{(1)}} \nabla_{\boldsymbol{\mu}_{t,i-1}} \ell + \frac{\partial \boldsymbol{\Sigma}_{t,i-1}}{\partial \boldsymbol{\psi}_{t,i-1}^{(1)}} \nabla_{\boldsymbol{\Sigma}_{t,i-1}} \ell \tag{108}$$

$$= -\tfrac{1}{2} \boldsymbol{\psi}_{t,i-1}^{(2)-1} \nabla_{\boldsymbol{\mu}_{t,i-1}} \ell \tag{109}$$

$$= \boldsymbol{\Sigma}_{t,i-1} \nabla_{\boldsymbol{\mu}_{t,i-1}} \ell \tag{110}$$

$$\nabla_{\boldsymbol{\psi}_{t,i-1}^{(2)}} \ell = \frac{\partial \boldsymbol{\mu}_{t,i-1}}{\partial \boldsymbol{\psi}_{t,i-1}^{(2)}} \nabla_{\boldsymbol{\mu}_{t,i-1}} \ell + \frac{\partial \boldsymbol{\Sigma}_{t,i-1}}{\partial \boldsymbol{\psi}_{t,i-1}^{(2)}} \nabla_{\boldsymbol{\Sigma}_{t,i-1}} \ell \tag{111}$$

$$= \tfrac{1}{2} \boldsymbol{\psi}_{t,i-1}^{(2)-1}\left(\nabla_{\boldsymbol{\mu}_{t,i-1}} \ell\right) \boldsymbol{\psi}_{t,i-1}^{(1)\mathsf{T}} \boldsymbol{\psi}_{t,i-1}^{(2)-1} + \tfrac{1}{2} \boldsymbol{\psi}_{t,i-1}^{(2)-1}\left(\nabla_{\boldsymbol{\Sigma}_{t,i-1}} \ell\right) \boldsymbol{\psi}_{t,i-1}^{(2)-1} \tag{112}$$

$$= 2\boldsymbol{\Sigma}_{t,i-1}\left(\nabla_{\boldsymbol{\mu}_{t,i-1}} \ell\right) \boldsymbol{\mu}_{t,i-1}^{\mathsf{T}} + 2\boldsymbol{\Sigma}_{t,i-1}\left(\nabla_{\boldsymbol{\Sigma}_{t,i-1}} \ell\right) \boldsymbol{\Sigma}_{t,i-1} \tag{113}$$

Therefore

$$\nabla_{\boldsymbol{\psi}_{t,i-1}^{(1)}} \mathbb{E}_{\boldsymbol{\theta}_t \sim q_{\boldsymbol{\psi}_{t,i-1}}} \left[ \log p \left( \boldsymbol{y}_t | f_t \left( \boldsymbol{\theta}_t \right) \right) \right] = \boldsymbol{\Sigma}_{t,i-1} \boldsymbol{g}_{t,i} \tag{114}$$

$$\nabla_{\boldsymbol{\psi}_{t,i-1}^{(2)}} \mathbb{E}_{\boldsymbol{\theta}_t \sim q_{\boldsymbol{\psi}_{t,i-1}}} \left[ \log p \left( \boldsymbol{y}_t | f_t \left( \boldsymbol{\theta}_t \right) \right) \right] = 2\boldsymbol{\Sigma}_{t,i-1} \boldsymbol{g}_{t,i} \boldsymbol{\mu}_{t,i-1}^{\mathsf{T}} + \boldsymbol{\Sigma}_{t,i-1} \mathbf{G}_{t,i} \boldsymbol{\Sigma}_{t,i-1} \tag{115}$$

and

$$\nabla_{\boldsymbol{\psi}_{t,i-1}^{(1)}} D_{\mathbb{KL}} \left( q_{\boldsymbol{\psi}_{t,i-1}} | q_{\boldsymbol{\psi}_{t|t-1}} \right) = \boldsymbol{\Sigma}_{t,i-1} \boldsymbol{\Sigma}_{t|t-1}^{-1} \left( \boldsymbol{\mu}_{t,i-1} - \boldsymbol{\mu}_{t|t-1} \right) \tag{116}$$

$$\nabla_{\boldsymbol{\psi}_{t,i-1}^{(2)}} D_{\mathbb{KL}} \left( q_{\boldsymbol{\psi}_{t,i-1}} | q_{\boldsymbol{\psi}_{t|t-1}} \right) = 2\boldsymbol{\Sigma}_{t,i-1} \boldsymbol{\Sigma}_{t|t-1}^{-1} \left( \boldsymbol{\mu}_{t,i-1} - \boldsymbol{\mu}_{t|t-1} \right) \boldsymbol{\mu}_{t,i-1}^{\mathsf{T}}$$
$$+ \boldsymbol{\Sigma}_{t,i-1} \left( \boldsymbol{\Sigma}_{t|t-1}^{-1} - \boldsymbol{\Sigma}_{t,i-1}^{-1} \right) \boldsymbol{\Sigma}_{t,i-1} \tag{117}$$

### E.1.1 BONG FC (explicit RVGA)

Substituting Eqs. (104) and (105) into Eq. (5) gives

$$\boldsymbol{\psi}_t^{(1)} = \boldsymbol{\psi}_{t|t-1}^{(1)} + \boldsymbol{g}_t - \mathbf{G}_t \boldsymbol{\mu}_{t|t-1} \tag{118}$$

$$\boldsymbol{\psi}_t^{(2)} = \boldsymbol{\psi}_{t|t-1}^{(2)} + \tfrac{1}{2} \mathbf{G}_t \tag{119}$$

Translating to $(\boldsymbol{\mu}_t, \boldsymbol{\Sigma}_t)$ gives the BONG-FC update

$$\boldsymbol{\mu}_t = \boldsymbol{\mu}_{t|t-1} + \boldsymbol{\Sigma}_t \boldsymbol{g}_t \tag{120}$$

$$\boldsymbol{\Sigma}_t^{-1} = \boldsymbol{\Sigma}_{t|t-1}^{-1} - \mathbf{G}_t \tag{121}$$

This is equivalent to the explicit update form of RVGA [Lambert et al., 2021]. Using $\mathbf{G}_t^{\text{MC-HESS}}$ this update takes $O(P^3)$ because of the matrix inversion. Using $\mathbf{G}_t^{\text{MC-EF}}$ and the Woodbury matrix identity we can write the update in a form that takes $O(MP^2 + M^3)$:

$$\boldsymbol{\mu}_t = \boldsymbol{\mu}_{t|t-1} + \mathbf{K}_t \mathbf{1}_M \tag{122}$$

$$\boldsymbol{\Sigma}_t = \boldsymbol{\Sigma}_{t|t-1} - \mathbf{K}_t \hat{\mathbf{G}}_t^{(1:M)\mathsf{T}} \boldsymbol{\Sigma}_{t|t-1} \tag{123}$$

$$\mathbf{K}_t = \boldsymbol{\Sigma}_{t|t-1} \hat{\mathbf{G}}_t^{(1:M)} \left( M\mathbf{I}_M + \hat{\mathbf{G}}_t^{(1:M)\mathsf{T}} \boldsymbol{\Sigma}_{t|t-1} \hat{\mathbf{G}}_t^{(1:M)} \right)^{-1} \tag{124}$$

Likewise using $\mathbf{G}_t^{\text{LIN-EF}}$ takes $O(P^2)$:

$$\boldsymbol{\mu}_t = \boldsymbol{\mu}_{t|t-1} + \mathbf{K}_t \tag{125}$$

$$\boldsymbol{\Sigma}_t = \boldsymbol{\Sigma}_{t|t-1} - \mathbf{K}_t \left( g_t^{\text{LIN}} \right)^{\mathsf{T}} \boldsymbol{\Sigma}_{t|t-1} \tag{126}$$

$$\mathbf{K}_t = \frac{\boldsymbol{\Sigma}_{t|t-1} g_t^{\text{LIN}}}{1 + \left( g_t^{\text{LIN}} \right)^{\mathsf{T}} \boldsymbol{\Sigma}_{t|t-1} g_t^{\text{LIN}}} \tag{127}$$

### E.1.2 BONG-LIN-HESS FC (CM-EKF)

Applying Proposition 4.2 to Eqs. (120) and (121) gives the BONG-LIN-HESS-FC update

$$\boldsymbol{\mu}_t = \boldsymbol{\mu}_{t|t-1} + \boldsymbol{\Sigma}_t \mathbf{H}_t^{\mathsf{T}} \mathbf{R}_t^{-1} \left( \boldsymbol{y}_t - \hat{\boldsymbol{y}}_t \right) \tag{128}$$

$$\boldsymbol{\Sigma}_t^{-1} = \boldsymbol{\Sigma}_{t|t-1}^{-1} + \mathbf{H}_t^{\mathsf{T}} \mathbf{R}_t^{-1} \mathbf{H}_t \tag{129}$$

This is equivalent to CM-EKF [Tronarp et al., 2018, Ollivier, 2018] and can be rewritten using the Woodbury identity in a form that takes $O(CP^2 + C^3)$:

$$\boldsymbol{\mu}_t = \boldsymbol{\mu}_{t|t-1} + \mathbf{K}_t (\boldsymbol{y}_t - \hat{\boldsymbol{y}}_t) \tag{130}$$

$$\boldsymbol{\Sigma}_t = \boldsymbol{\Sigma}_{t|t-1} - \mathbf{K}_t \mathbf{H}_t \boldsymbol{\Sigma}_{t|t-1} \tag{131}$$

$$\mathbf{K}_t = \boldsymbol{\Sigma}_{t|t-1} \mathbf{H}_t^{\mathsf{T}} \left( \mathbf{R}_t + \mathbf{H}_t \boldsymbol{\Sigma}_{t|t-1} \mathbf{H}_t^{\mathsf{T}} \right)^{-1} \tag{132}$$

### E.1.3  BLR FC

Substituting Eqs. (104) to (107) into Eq. (33) gives

$$\boldsymbol{\psi}_{t,i}^{(1)} = \boldsymbol{\psi}_{t,i-1}^{(1)} + \alpha \left( \boldsymbol{g}_{t,i} - \mathbf{G}_{t,i} \boldsymbol{\mu}_{t,i-1} - \boldsymbol{\Sigma}_{t,i-1}^{-1} \boldsymbol{\mu}_{t,i-1} + \boldsymbol{\Sigma}_{t|t-1}^{-1} \boldsymbol{\mu}_{t|t-1} \right) \tag{133}$$

$$\boldsymbol{\psi}_{t,i}^{(2)} = \boldsymbol{\psi}_{t,i-1}^{(2)} + \frac{\alpha}{2} \left( \mathbf{G}_{t,i} + \boldsymbol{\Sigma}_{t,i-1}^{-1} - \boldsymbol{\Sigma}_{t|t-1}^{-1} \right) \tag{134}$$

Translating to $(\boldsymbol{\mu}_{t,i}, \boldsymbol{\Sigma}_{t,i})$ gives the BLR-FC update

$$\boldsymbol{\mu}_{t,i} = \boldsymbol{\mu}_{t,i-1} + \alpha \boldsymbol{\Sigma}_{t,i} \boldsymbol{\Sigma}_{t|t-1}^{-1} \left( \boldsymbol{\mu}_{t|t-1} - \boldsymbol{\mu}_{t,i-1} \right) + \alpha \boldsymbol{\Sigma}_{t,i} \boldsymbol{g}_{t,i} \tag{135}$$

$$\boldsymbol{\Sigma}_{t,i}^{-1} = (1 - \alpha) \boldsymbol{\Sigma}_{t,i-1}^{-1} + \alpha \boldsymbol{\Sigma}_{t|t-1}^{-1} - \alpha \mathbf{G}_{t,i} \tag{136}$$

This update takes $O(P^3)$ per iteration because of the matrix inversion. In Appendix E.1.1 we were able to use the Woodbury identity to exploit the low rank of $\mathbf{G}_t^{\text{MC-EF}}$ and $\mathbf{G}_t^{\text{LIN-EF}}$ and obtain BONG updates with complexity quadratic in $P$. This does not appear possible with Eq. (136) because of the extra precision term on the RHS (applying Woodbury would require inverting $(1-\alpha)\boldsymbol{\Sigma}_{t,i-1}^{-1} + \alpha\boldsymbol{\Sigma}_{t|t-1}^{-1}$). Therefore unlike BONG-FC, BLR-FC requires time cubic in the model size, for reasons that can be traced back to the KL term in Eq. (33).

### E.1.4  BLR-LIN-HESS FC

Applying Proposition 4.2 to Eqs. (135) and (136) gives the BLR-LIN-HESS-FC update

$$\boldsymbol{\mu}_{t,i} = \boldsymbol{\mu}_{t,i-1} + \alpha \boldsymbol{\Sigma}_{t,i} \boldsymbol{\Sigma}_{t|t-1}^{-1} \left( \boldsymbol{\mu}_{t|t-1} - \boldsymbol{\mu}_{t,i-1} \right) + \alpha \boldsymbol{\Sigma}_{t,i} \mathbf{H}_{t,i}^{\mathsf{T}} \mathbf{R}_{t,i}^{-1} \left( \boldsymbol{y}_t - \hat{\boldsymbol{y}}_{t,i} \right) \tag{137}$$

$$\boldsymbol{\Sigma}_{t,i}^{-1} = (1 - \alpha) \boldsymbol{\Sigma}_{t,i-1}^{-1} + \alpha \boldsymbol{\Sigma}_{t|t-1}^{-1} + \alpha \mathbf{H}_{t,i}^{\mathsf{T}} \mathbf{R}_{t,i}^{-1} \mathbf{H}_{t,i} \tag{138}$$

This update takes $O(P^3)$ per iteration because of the matrix inversion.

### E.1.5  BOG FC

Substituting Eqs. (114) and (115) into Eq. (31) gives

$$\boldsymbol{\psi}_t^{(1)} = \boldsymbol{\psi}_{t|t-1}^{(1)} + \alpha \boldsymbol{\Sigma}_{t|t-1} \boldsymbol{g}_t \tag{139}$$

$$\boldsymbol{\psi}_t^{(2)} = \boldsymbol{\psi}_{t|t-1}^{(2)} + 2\alpha \boldsymbol{\Sigma}_{t|t-1} \boldsymbol{g}_t \boldsymbol{\mu}_{t|t-1}^{\mathsf{T}} + \alpha \boldsymbol{\Sigma}_{t|t-1} \mathbf{G}_t \boldsymbol{\Sigma}_{t|t-1} \tag{140}$$

Translating to $(\boldsymbol{\mu}_t, \boldsymbol{\Sigma}_t)$ gives the BOG-FC update

$$\boldsymbol{\mu}_t = \boldsymbol{\Sigma}_t \boldsymbol{\Sigma}_{t|t-1}^{-1} \boldsymbol{\mu}_{t|t-1} + \alpha \boldsymbol{\Sigma}_t \boldsymbol{\Sigma}_{t|t-1} \boldsymbol{g}_t \tag{141}$$

$$\boldsymbol{\Sigma}_t^{-1} = \boldsymbol{\Sigma}_{t|t-1}^{-1} - 4\alpha \boldsymbol{\Sigma}_{t|t-1} \boldsymbol{g}_t \boldsymbol{\mu}_{t|t-1}^{\mathsf{T}} - 2\alpha \boldsymbol{\Sigma}_{t|t-1} \mathbf{G}_t \boldsymbol{\Sigma}_{t|t-1} \tag{142}$$

This update takes $O(P^3)$ because of the matrix inversion. The greater cost of the BOG-FC update relative to BONG-FC can be traced to the difference between GD and NGD: the NLL gradients wrt $\boldsymbol{\psi}_{t|t-1}$ in Eqs. (114) and (115) are more complicated than the gradients wrt $\boldsymbol{\rho}_{t|t-1}$ in Eqs. (104) and (105).

### E.1.6  BOG-LIN-HESS FC

Applying Proposition 4.2 to Eqs. (141) and (142) gives the BOG-LIN-HESS-FC update

$$\boldsymbol{\mu}_t = \boldsymbol{\Sigma}_t \boldsymbol{\Sigma}_{t|t-1}^{-1} \boldsymbol{\mu}_{t|t-1} + \alpha \boldsymbol{\Sigma}_t \boldsymbol{\Sigma}_{t|t-1} \mathbf{H}_t^{\mathsf{T}} \mathbf{R}_t^{-1} \left( \boldsymbol{y}_t - \hat{\boldsymbol{y}}_t \right) \tag{143}$$

$$\boldsymbol{\Sigma}_t^{-1} = \boldsymbol{\Sigma}_{t|t-1}^{-1} - 4\alpha \boldsymbol{\Sigma}_{t|t-1} \mathbf{H}_t^{\mathsf{T}} \mathbf{R}_t^{-1} \left( \boldsymbol{y}_t - \hat{\boldsymbol{y}}_t \right) \boldsymbol{\mu}_{t|t-1}^{\mathsf{T}} + 2\alpha \boldsymbol{\Sigma}_{t|t-1} \mathbf{H}_t^{\mathsf{T}} \mathbf{R}_t^{-1} \mathbf{H}_t \boldsymbol{\Sigma}_{t|t-1} \tag{144}$$

This update takes $O(P^3)$ because of the matrix inversion.

### E.1.7 BBB FC

Substituting Eqs. (114) to (117) into Eq. (34) gives

$$\boldsymbol{\psi}_{t,i}^{(1)} = \boldsymbol{\psi}_{t,i-1}^{(1)} + \alpha\boldsymbol{\Sigma}_{t,i-1}\boldsymbol{g}_{t,i} - \alpha\boldsymbol{\Sigma}_{t,i-1}\boldsymbol{\Sigma}_{t|t-1}^{-1}\left(\boldsymbol{\mu}_{t,i-1} - \boldsymbol{\mu}_{t|t-1}\right) \tag{145}$$

$$\boldsymbol{\psi}_{t}^{(2)} = \boldsymbol{\psi}_{t,i-1}^{(2)} + 2\alpha\boldsymbol{\Sigma}_{t,i-1}\boldsymbol{g}_{t,i}\boldsymbol{\mu}_{t,i-1}^{\mathsf{T}} + \alpha\boldsymbol{\Sigma}_{t|t-1}\mathbf{G}_t\boldsymbol{\Sigma}_{t|t-1}$$
$$- 2\alpha\boldsymbol{\Sigma}_{t,i-1}\boldsymbol{\Sigma}_{t|t-1}^{-1}\left(\boldsymbol{\mu}_{t,i-1} - \boldsymbol{\mu}_{t|t-1}\right)\boldsymbol{\mu}_{t,i-1}^{\mathsf{T}} - \alpha\boldsymbol{\Sigma}_{t,i-1}\left(\boldsymbol{\Sigma}_{t|t-1}^{-1} - \boldsymbol{\Sigma}_{t,i-1}^{-1}\right)\boldsymbol{\Sigma}_{t,i-1} \tag{146}$$

Translating to $(\boldsymbol{\mu}_{t,i}, \boldsymbol{\Sigma}_{t,i})$ gives the BBB-FC update

$$\boldsymbol{\mu}_{t,i} = \boldsymbol{\Sigma}_{t,i}\boldsymbol{\Sigma}_{t,i-1}^{-1}\boldsymbol{\mu}_{t,i-1} + \alpha\boldsymbol{\Sigma}_{t,i}\boldsymbol{\Sigma}_{t,i-1}\left(\boldsymbol{g}_{t,i} + \boldsymbol{\Sigma}_{t|t-1}^{-1}\left(\boldsymbol{\mu}_{t|t-1} - \boldsymbol{\mu}_{t,i-1}\right)\right) \tag{147}$$

$$\boldsymbol{\Sigma}_{t,i}^{-1} = \boldsymbol{\Sigma}_{t,i-1}^{-1} - 2\alpha\boldsymbol{\Sigma}_{t,i-1}\begin{pmatrix} 2\boldsymbol{\Sigma}_{t|t-1}^{-1}\left(\boldsymbol{\mu}_{t|t-1} - \boldsymbol{\mu}_{t,i-1}\right)\boldsymbol{\mu}_{t,i-1}^{\mathsf{T}} + \mathbf{I}_P \\ +2\boldsymbol{g}_{t,i}\boldsymbol{\mu}_{t,i-1}^{\mathsf{T}} + \left(\mathbf{G}_{t,i} - \boldsymbol{\Sigma}_{t|t-1}^{-1}\right)\boldsymbol{\Sigma}_{t,i-1} \end{pmatrix} \tag{148}$$

This update takes $O(P^3)$ per iteration because of the matrix inversion.

### E.1.8 BBB-LIN-HESS FC

Applying Proposition 4.2 to Eqs. (147) and (148) gives the BBB-LIN-HESS-FC update

$$\boldsymbol{\mu}_{t,i} = \boldsymbol{\Sigma}_{t,i}\boldsymbol{\Sigma}_{t,i-1}^{-1}\boldsymbol{\mu}_{t,i-1}$$
$$+ \alpha\boldsymbol{\Sigma}_{t,i}\boldsymbol{\Sigma}_{t,i-1}\left(\mathbf{H}_{t,i}^{\mathsf{T}}\mathbf{R}_{t,i}^{-1}\left(\boldsymbol{y}_t - \hat{\boldsymbol{y}}_{t,i}\right) + \boldsymbol{\Sigma}_{t|t-1}^{-1}\left(\boldsymbol{\mu}_{t|t-1} - \boldsymbol{\mu}_{t,i-1}\right)\right) \tag{149}$$

$$\boldsymbol{\Sigma}_{t,i}^{-1} = \boldsymbol{\Sigma}_{t,i-1}^{-1} - 2\alpha\boldsymbol{\Sigma}_{t,i-1}\begin{pmatrix} 2\boldsymbol{\Sigma}_{t|t-1}^{-1}\left(\boldsymbol{\mu}_{t|t-1} - \boldsymbol{\mu}_{t,i-1}\right)\boldsymbol{\mu}_{t,i-1}^{\mathsf{T}} + \mathbf{I}_P \\ +2\mathbf{H}_{t,i}^{\mathsf{T}}\mathbf{R}_{t,i}^{-1}\left(\boldsymbol{y}_t - \hat{\boldsymbol{y}}_{t,i}\right)\boldsymbol{\mu}_{t,i-1}^{\mathsf{T}} \\ -\left(\mathbf{H}_{t,i}^{\mathsf{T}}\mathbf{R}_{t,i}^{-1}\mathbf{H}_{t,i} + \boldsymbol{\Sigma}_{t|t-1}^{-1}\right)\boldsymbol{\Sigma}_{t,i-1} \end{pmatrix} \tag{150}$$

This update takes $O(P^3)$ per iteration because of the matrix inversion.

## E.2 Full covariance Gaussian, Moment parameters

The Bonnet and Price theorems give

$$\nabla_{\boldsymbol{\mu}_{t,i-1}}\mathbb{E}_{\boldsymbol{\theta}_t \sim q_{\boldsymbol{\psi}_{t,i-1}}}\left[\log p\left(\boldsymbol{y}_t | f_t\left(\boldsymbol{\theta}_t\right)\right)\right] = \boldsymbol{g}_{t,i} \tag{151}$$

$$\nabla_{\boldsymbol{\Sigma}_{t,i-1}}\mathbb{E}_{\boldsymbol{\theta}_t \sim q_{\boldsymbol{\psi}_{t,i-1}}}\left[\log p\left(\boldsymbol{y}_t | f_t\left(\boldsymbol{\theta}_t\right)\right)\right] = \tfrac{1}{2}\mathbf{G}_{t,i} \tag{152}$$

From Appendix E.1 we have

$$\nabla_{\boldsymbol{\mu}_{t,i-1}}D_{\mathbb{KL}}\left(q_{\boldsymbol{\psi}_{t,i-1}} | q_{\boldsymbol{\psi}_{t|t-1}}\right) = \boldsymbol{\Sigma}_{t|t-1}^{-1}\left(\boldsymbol{\mu}_{t,i-1} - \boldsymbol{\mu}_{t|t-1}\right) \tag{153}$$

$$\nabla_{\boldsymbol{\Sigma}_{t,i-1}}D_{\mathbb{KL}}\left(q_{\boldsymbol{\psi}_{t,i-1}} | q_{\boldsymbol{\psi}_{t|t-1}}\right) = \tfrac{1}{2}\left(\boldsymbol{\Sigma}_{t|t-1}^{-1} - \boldsymbol{\Sigma}_{t,i-1}^{-1}\right) \tag{154}$$

We write the Fisher with respect to the moment parameters $\boldsymbol{\psi} = (\boldsymbol{\mu}, \mathrm{vec}(\boldsymbol{\Sigma}))$ as a block matrix:

$$\mathbf{F} = \begin{bmatrix} \mathbf{F}_{\boldsymbol{\mu},\boldsymbol{\mu}} & \mathbf{F}_{\boldsymbol{\mu},\boldsymbol{\Sigma}} \\ \mathbf{F}_{\boldsymbol{\Sigma},\boldsymbol{\mu}} & \mathbf{F}_{\boldsymbol{\Sigma},\boldsymbol{\Sigma}} \end{bmatrix} \tag{155}$$

The blocks can be calculated by the second-order Fisher formula

$$\mathbf{F}_{\boldsymbol{\mu},\boldsymbol{\mu}} = -\mathbb{E}_{q_\psi}[\nabla_{\boldsymbol{\mu},\boldsymbol{\mu}} \log q_\psi(\boldsymbol{\theta})] \tag{156}$$

$$= \boldsymbol{\Sigma}^{-1} \tag{157}$$

$$\mathbf{F}_{\boldsymbol{\mu},\boldsymbol{\Sigma}} = -\mathbb{E}_{q_\psi}[\nabla_{\boldsymbol{\mu},\boldsymbol{\Sigma}} \log q_\psi(\boldsymbol{\theta})] \tag{158}$$

$$= -\mathbb{E}_{q_\psi}[(\nabla_{\boldsymbol{\Sigma}}\boldsymbol{\Sigma}^{-1})(\boldsymbol{\theta}-\boldsymbol{\mu})] \tag{159}$$

$$= \mathbf{0} \tag{160}$$

$$\mathbf{F}_{\boldsymbol{\Sigma},\boldsymbol{\Sigma}} = -\mathbb{E}_{q_\psi}[\nabla_{\boldsymbol{\Sigma},\boldsymbol{\Sigma}} \log q_\psi(\boldsymbol{\theta})] \tag{161}$$

$$= -\mathbb{E}_{q_\psi}\left[\nabla_{\boldsymbol{\Sigma}}\left(\tfrac{1}{2}\boldsymbol{\Sigma}^{-1}(\boldsymbol{\theta}-\boldsymbol{\mu})(\boldsymbol{\theta}-\boldsymbol{\mu})^{\mathsf{T}}\boldsymbol{\Sigma}^{-1} - \tfrac{1}{2}\boldsymbol{\Sigma}^{-1}\right)\right] \tag{162}$$

$$= -\tfrac{1}{2}\mathbb{E}_{q_\psi}\left[\begin{array}{c}(\nabla_{\boldsymbol{\Sigma}}\boldsymbol{\Sigma}^{-1})(\boldsymbol{\theta}-\boldsymbol{\mu})(\boldsymbol{\theta}-\boldsymbol{\mu})^{\mathsf{T}}\boldsymbol{\Sigma}^{-1} \\ +\boldsymbol{\Sigma}^{-1}(\boldsymbol{\theta}-\boldsymbol{\mu})(\boldsymbol{\theta}-\boldsymbol{\mu})^{\mathsf{T}}(\nabla_{\boldsymbol{\Sigma}}\boldsymbol{\Sigma}^{-1}) - (\nabla_{\boldsymbol{\Sigma}}\boldsymbol{\Sigma}^{-1})\end{array}\right] \tag{163}$$

$$= -\tfrac{1}{2}\nabla_{\boldsymbol{\Sigma}}\boldsymbol{\Sigma}^{-1} \tag{164}$$

$$= \tfrac{1}{2}\boldsymbol{\Sigma}^{-1} \otimes \boldsymbol{\Sigma}^{-1} \tag{165}$$

In the final line we used

$$\nabla_{\Sigma_{k\ell}}\left(\boldsymbol{\Sigma}^{-1}\right)_{ij} = -\left(\boldsymbol{\Sigma}^{-1}\right)_{ik}\left(\boldsymbol{\Sigma}^{-1}\right)_{j\ell} \tag{166}$$

$$= -\left(\boldsymbol{\Sigma}^{-1} \otimes \boldsymbol{\Sigma}^{-1}\right)_{ij,k\ell} \tag{167}$$

with $ij$ and $k\ell$ treated as composite indices in $[P^2]$. Therefore the preconditioner in the NGD methods is

$$\mathbf{F}_{\psi_{t,i-1}}^{-1} = \left[\begin{array}{cc} \boldsymbol{\Sigma}_{t,i-1} & \mathbf{0} \\ \mathbf{0} & 2\boldsymbol{\Sigma}_{t,i-1} \otimes \boldsymbol{\Sigma}_{t,i-1} \end{array}\right] \tag{168}$$

### E.2.1  BONG FC, Moment

Substituting Eqs. (151), (152) and (168) into Eq. (3) gives the BONG-FC_MOM update

$$\boldsymbol{\mu}_t = \boldsymbol{\mu}_{t|t-1} + \boldsymbol{\Sigma}_{t|t-1}\boldsymbol{g}_t \tag{169}$$

$$\boldsymbol{\Sigma}_t = \boldsymbol{\Sigma}_{t|t-1} + \boldsymbol{\Sigma}_{t|t-1}\mathbf{G}_t\boldsymbol{\Sigma}_{t|t-1} \tag{170}$$

This update takes $O(P^3)$ using $\mathbf{G}_t^{\text{MC-HESS}}$, $O(MP^2)$ using $\mathbf{G}_t^{\text{MC-EF}}$, and $O(P^2)$ using $\mathbf{G}_t^{\text{LIN-EF}}$. The update is similar to the BONG-FC update except that Eq. (170) ignores the $\hat{\mathbf{G}}_t^{(1:M)\mathsf{T}}\boldsymbol{\Sigma}_{t|t-1}\hat{\mathbf{G}}_t^{(1:M)}$ term in Eq. (124) or the $(\boldsymbol{g}_t^{\text{LIN}})^{\mathsf{T}}\boldsymbol{\Sigma}_{t|t-1}\boldsymbol{g}_t^{\text{LIN}}$ term in Eq. (127) which estimate the epistemic part of predictive uncertainty.

### E.2.2  BONG-LIN-HESS FC, Moment

Applying Proposition 4.2 to Eqs. (169) and (170) gives the BONG-LIN-HESS-FC_MOM update

$$\boldsymbol{\mu}_t = \boldsymbol{\mu}_{t|t-1} + \boldsymbol{\Sigma}_{t|t-1}\mathbf{H}_t^{\mathsf{T}}\mathbf{R}_t^{-1}(\boldsymbol{y}_t - \hat{\boldsymbol{y}}_t) \tag{171}$$

$$\boldsymbol{\Sigma}_t = \boldsymbol{\Sigma}_{t|t-1} - \boldsymbol{\Sigma}_{t|t-1}\mathbf{H}_t^{\mathsf{T}}\mathbf{R}_t^{-1}\mathbf{H}_t\boldsymbol{\Sigma}_{t|t-1} \tag{172}$$

This update takes $O(CP^2)$.

### E.2.3  BLR FC, Moment

Substituting Eqs. (151) to (154) and (168) into Eq. (32) gives the BLR-FC_MOM update

$$\boldsymbol{\mu}_{t,i} = \boldsymbol{\mu}_{t,i-1} + \alpha\boldsymbol{\Sigma}_{t,i-1}\boldsymbol{\Sigma}_{t|t-1}^{-1}\left(\boldsymbol{\mu}_{t|t-1} - \boldsymbol{\mu}_{t,i-1}\right) + \alpha\boldsymbol{\Sigma}_{t,i-1}\boldsymbol{g}_{t,i} \tag{173}$$

$$\boldsymbol{\Sigma}_{t,i} = (1+\alpha)\boldsymbol{\Sigma}_{t,i-1} + \alpha\boldsymbol{\Sigma}_{t,i-1}\left(\mathbf{G}_{t,i} - \boldsymbol{\Sigma}_{t|t-1}^{-1}\right)\boldsymbol{\Sigma}_{t,i-1} \tag{174}$$

This update takes $O(P^3)$ per iteration because of the matrix inversion. Using $\mathbf{G}_t^{\text{MC-EF}} = -\frac{1}{M}\hat{\mathbf{G}}_t^{(1:M)}\hat{\mathbf{G}}_t^{(1:M)\mathsf{T}}$ or $\mathbf{G}^{\text{LIN-EF}} = -\boldsymbol{g}_t^{\text{LIN}}(\boldsymbol{g}_t^{\text{LIN}})^{\mathsf{T}}$ allows the BONG-FC_MOM covariance update in Eq. (170) to scale quadratically, but this does not help here. Instead the BLR-FC_MOM update scales cubically because of the additional $\boldsymbol{\Sigma}_{t|t-1}^{-1}$ term that comes from the KL divergence in the VI objective.

### E.2.4 BLR-LIN-HESS FC, Moment

Applying Proposition 4.2 to Eqs. (173) and (174) gives the BLR-LIN-HESS-FC_MOM update

$$\boldsymbol{\mu}_{t,i} = \boldsymbol{\mu}_{t,i-1} + \alpha\boldsymbol{\Sigma}_{t,i-1}\boldsymbol{\Sigma}_{t|t-1}^{-1}\left(\boldsymbol{\mu}_{t|t-1} - \boldsymbol{\mu}_{t,i-1}\right) + \alpha\boldsymbol{\Sigma}_{t,i-1}\mathbf{H}_{t,i}^{\mathsf{T}}\mathbf{R}_{t,i}^{-1}(\boldsymbol{y}_t - \hat{\boldsymbol{y}}_{t,i}) \tag{175}$$

$$\boldsymbol{\Sigma}_{t,i} = (1 + \alpha)\,\boldsymbol{\Sigma}_{t,i-1} - \alpha\boldsymbol{\Sigma}_{t,i-1}\left(\boldsymbol{\Sigma}_{t|t-1}^{-1} + \mathbf{H}_{t,i}^{\mathsf{T}}\mathbf{R}_{t,i}^{-1}\mathbf{H}_{t,i}\right)\boldsymbol{\Sigma}_{t,i-1} \tag{176}$$

This update takes $O(P^3)$ per iteration because of the matrix inversion in the $\boldsymbol{\Sigma}_{t|t-1}^{-1}$ term that comes from the KL divergence in the VI objective.

### E.2.5 BOG FC, Moment

Substituting Eqs. (151) and (152) into Eq. (31) gives the BOG-FC_MOM update

$$\boldsymbol{\mu}_t = \boldsymbol{\mu}_{t|t-1} + \alpha\boldsymbol{g}_t \tag{177}$$

$$\boldsymbol{\Sigma}_t = \boldsymbol{\Sigma}_{t|t-1} + \frac{\alpha}{2}\mathbf{G}_t \tag{178}$$

Note the mean update is vanilla online gradient descent (OGD) and does not depend on the covariance. This update takes $O(MP^2)$ using $\mathbf{G}_t^{\text{MC-HESS}}$ or $\mathbf{G}_t^{\text{MC-EF}}$ and $O(P^2)$ using $\mathbf{G}_t^{\text{LIN-EF}}$.

### E.2.6 BOG-LIN-HESS FC, Moment

Applying Proposition 4.2 to Eqs. (177) and (178) gives the BOG-LIN-HESS-FC_MOM update

$$\boldsymbol{\mu}_t = \boldsymbol{\mu}_{t|t-1} + \alpha\mathbf{H}_t^{\mathsf{T}}\mathbf{R}_t^{-1}(\boldsymbol{y}_t - \hat{\boldsymbol{y}}_t) \tag{179}$$

$$\boldsymbol{\Sigma}_t = \boldsymbol{\Sigma}_{t|t-1} - \frac{\alpha}{2}\mathbf{H}_t^{\mathsf{T}}\mathbf{R}_t^{-1}\mathbf{H}_t \tag{180}$$

This update takes $O(CP^2)$.

### E.2.7 BBB FC, Moment

Substituting Eqs. (151) to (154) into Eq. (34) gives the BBB-FC_MOM

$$\boldsymbol{\mu}_{t,i} = \boldsymbol{\mu}_{t,i-1} + \alpha\boldsymbol{\Sigma}_{t|t-1}^{-1}\left(\boldsymbol{\mu}_{t|t-1} - \boldsymbol{\mu}_{t,i-1}\right) + \alpha\boldsymbol{g}_{t,i} \tag{181}$$

$$\boldsymbol{\Sigma}_{t,i} = \boldsymbol{\Sigma}_{t,i-1} + \frac{\alpha}{2}\left(\boldsymbol{\Sigma}_{t,i-1}^{-1} - \boldsymbol{\Sigma}_{t|t-1}^{-1} + \mathbf{G}_{t,i}\right) \tag{182}$$

This update takes $O(P^3)$ per iteration because of the matrix inversion, which traces back to the VI objective. Comparing to the BOG-FC_MOM update in Eqs. (177) and (178) (which has quadratic complexity in $P$), the extra terms here come from the KL part of Eq. (34).

### E.2.8 BBB-LIN-HESS FC, Moment

Applying Proposition 4.2 to Eqs. (181) and (182) gives the BBB-LIN-HESS-FC_MOM update

$$\boldsymbol{\mu}_{t,i} = \boldsymbol{\mu}_{t,i-1} + \alpha\boldsymbol{\Sigma}_{t|t-1}^{-1}\left(\boldsymbol{\mu}_{t|t-1} - \boldsymbol{\mu}_{t,i-1}\right) + \alpha\mathbf{H}_{t,i}^{\mathsf{T}}\mathbf{R}_{t,i}^{-1}(\boldsymbol{y}_t - \hat{\boldsymbol{y}}_{t,i}) \tag{183}$$

$$\boldsymbol{\Sigma}_t = \boldsymbol{\Sigma}_{t,i-1} + \frac{\alpha}{2}\left(\boldsymbol{\Sigma}_{t,i-1}^{-1} - \boldsymbol{\Sigma}_{t|t-1}^{-1} - \mathbf{H}_{t,i}^{\mathsf{T}}\mathbf{R}_{t,i}^{-1}\mathbf{H}_{t,i}\right) \tag{184}$$

This update takes $O(P^3)$ per iteration because of the matrix inversion. Comparing to the BOG-LIN-HESS-FC_MOM update in Eqs. (179) and (180) (which has quadratic complexity in $P$), the extra terms here come from the KL part of Eq. (34).

### E.3 Diagonal Gaussian, Natural parameters

Throughout this subsection, vector multiplication and exponents are elementwise.

The natural and dual parameters for a diagonal Gaussian are given by

$$\boldsymbol{\psi}_{t,i-1}^{(1)} = \boldsymbol{\sigma}_{t,i-1}^{-2}\boldsymbol{\mu}_{t,i-1} \qquad \boldsymbol{\rho}_{t,i-1}^{(1)} = \boldsymbol{\mu}_{t,i-1} \tag{185}$$

$$\boldsymbol{\psi}_{t,i-1}^{(2)} = -\tfrac{1}{2}\boldsymbol{\sigma}_{t,i-1}^{-2} \qquad \boldsymbol{\rho}_{t,i-1}^{(2)} = \boldsymbol{\mu}_{t,i-1}\boldsymbol{\mu}_{t,i-1}^{\mathsf{T}} + \boldsymbol{\sigma}_{t,i-1}^2 \tag{186}$$

Inverting these relationships gives

$$\boldsymbol{\mu}_{t,i-1} = -\tfrac{1}{2}\left(\boldsymbol{\psi}_{t,i-1}^{(2)}\right)^{-1}\boldsymbol{\psi}_{t,i-1}^{(1)} = \boldsymbol{\rho}_{t,i-1}^{(1)} \tag{187}$$

$$\boldsymbol{\sigma}_{t,i-1}^2 = -\tfrac{1}{2}\left(\boldsymbol{\psi}_{t,i-1}^{(2)}\right)^{-1} \qquad = \boldsymbol{\rho}_{t,i-1}^{(2)} - \left(\boldsymbol{\rho}_{t,i-1}^{(1)}\right)^2 \tag{188}$$

The KL divergence in the VI loss is

$$D_{\mathbb{KL}}\big(q_{\boldsymbol{\psi}_{t,i-1}}|q_{\boldsymbol{\psi}_{t|t-1}}\big) = \tfrac{1}{2}\left(\boldsymbol{\mu}_{t,i-1}-\boldsymbol{\mu}_{t|t-1}\right)^2\boldsymbol{\sigma}_{t|t-1}^{-2}+\tfrac{1}{2}\sum\left(\boldsymbol{\sigma}_{t|t-1}^{-2}\boldsymbol{\sigma}_{t,i-1}^2-\log\boldsymbol{\sigma}_{t,i-1}^2\right)+\text{const} \tag{189}$$

with gradients

$$\nabla_{\boldsymbol{\mu}_{t,i-1}}D_{\mathbb{KL}}\big(q_{\boldsymbol{\psi}_{t,i-1}}|q_{\boldsymbol{\psi}_{t|t-1}}\big) = \boldsymbol{\sigma}_{t|t-1}^{-2}\left(\boldsymbol{\mu}_{t,i-1}-\boldsymbol{\mu}_{t|t-1}\right) \tag{190}$$

$$\nabla_{\boldsymbol{\sigma}_{t,i-1}^2}D_{\mathbb{KL}}\big(q_{\boldsymbol{\psi}_{t,i-1}}|q_{\boldsymbol{\psi}_{t|t-1}}\big) = \tfrac{1}{2}\left(\boldsymbol{\sigma}_{t|t-1}^{-2}-\boldsymbol{\sigma}_{t,i-1}^{-2}\right) \tag{191}$$

For any scalar function $\ell$ the chain rule gives

$$\nabla_{\boldsymbol{\rho}_{t,i-1}^{(1)}}\ell = \frac{\partial\boldsymbol{\mu}_{t,i-1}}{\partial\boldsymbol{\rho}_{t,i-1}^{(1)}}\nabla_{\boldsymbol{\mu}_{t,i-1}}\ell + \frac{\partial\boldsymbol{\sigma}_{t,i-1}^2}{\partial\boldsymbol{\rho}_{t,i-1}^{(1)}}\nabla_{\boldsymbol{\sigma}_{t,i-1}^2}\ell \tag{192}$$

$$= \nabla_{\boldsymbol{\mu}_{t,i-1}}\ell - 2\boldsymbol{\mu}_{t,i-1}\nabla_{\boldsymbol{\sigma}_{t,i-1}^2}\ell \tag{193}$$

$$\nabla_{\boldsymbol{\rho}_{t,i-1}^{(2)}}\ell = \frac{\partial\boldsymbol{\mu}_{t,i-1}}{\partial\boldsymbol{\rho}_{t,i-1}^{(2)}}\nabla_{\boldsymbol{\mu}_{t,i-1}}\ell + \frac{\partial\boldsymbol{\sigma}_{t,i-1}^2}{\partial\boldsymbol{\rho}_{t,i-1}^{(2)}}\nabla_{\boldsymbol{\sigma}_{t,i-1}^2}\ell \tag{194}$$

$$= \nabla_{\boldsymbol{\sigma}_{t,i-1}^2}\ell \tag{195}$$

Therefore

$$\nabla_{\boldsymbol{\rho}_{t,i-1}^{(1)}}\mathbb{E}_{\boldsymbol{\theta}_t\sim q_{\boldsymbol{\psi}_{t,i-1}}}\left[\log p\left(\boldsymbol{y}_t|f_t\left(\boldsymbol{\theta}_t\right)\right)\right] = \boldsymbol{g}_{t,i} - \text{diag}\left(\mathbf{G}_{t,i}\right)\boldsymbol{\mu}_{t,i-1} \tag{196}$$

$$\nabla_{\boldsymbol{\rho}_{t,i-1}^{(2)}}\mathbb{E}_{\boldsymbol{\theta}_t\sim q_{\boldsymbol{\psi}_{t,i-1}}}\left[\log p\left(\boldsymbol{y}_t|f_t\left(\boldsymbol{\theta}_t\right)\right)\right] = \tfrac{1}{2}\text{diag}\left(\mathbf{G}_{t,i}\right) \tag{197}$$

and

$$\nabla_{\boldsymbol{\rho}_{t,i-1}^{(1)}}D_{\mathbb{KL}}\big(q_{\boldsymbol{\psi}_{t,i-1}}|q_{\boldsymbol{\psi}_{t|t-1}}\big) = \boldsymbol{\sigma}_{t,i-1}^{-2}\boldsymbol{\mu}_{t,i-1} - \boldsymbol{\sigma}_{t|t-1}^{-2}\boldsymbol{\mu}_{t|t-1} \tag{198}$$

$$\nabla_{\boldsymbol{\rho}_{t,i-1}^{(2)}}D_{\mathbb{KL}}\big(q_{\boldsymbol{\psi}_{t,i-1}}|q_{\boldsymbol{\psi}_{t|t-1}}\big) = \tfrac{1}{2}\left(\boldsymbol{\sigma}_{t|t-1}^{-2} - \boldsymbol{\sigma}_{t,i-1}^{-2}\right) \tag{199}$$

Following the same approach for $\boldsymbol{\psi}$ gives

$$\nabla_{\boldsymbol{\psi}_{t,i-1}^{(1)}}\ell = \frac{\partial\boldsymbol{\mu}_{t,i-1}}{\partial\boldsymbol{\psi}_{t,i-1}^{(1)}}\nabla_{\boldsymbol{\mu}_{t,i-1}}\ell + \frac{\partial\boldsymbol{\sigma}_{t,i-1}^2}{\partial\boldsymbol{\psi}_{t,i-1}^{(1)}}\nabla_{\boldsymbol{\Sigma}_{t,i-1}}\ell \tag{200}$$

$$= -\tfrac{1}{2}\left(\boldsymbol{\psi}_{t,i-1}^{(2)}\right)^{-1}\nabla_{\boldsymbol{\mu}_{t,i-1}}\ell \tag{201}$$

$$= \boldsymbol{\sigma}_{t,i-1}^2\nabla_{\boldsymbol{\mu}_{t,i-1}}\ell \tag{202}$$

$$\nabla_{\boldsymbol{\psi}_{t,i-1}^{(2)}}\ell = \frac{\partial\boldsymbol{\mu}_{t,i-1}}{\partial\boldsymbol{\psi}_{t,i-1}^{(2)}}\nabla_{\boldsymbol{\mu}_{t,i-1}}\ell + \frac{\partial\boldsymbol{\sigma}_{t,i-1}^2}{\partial\boldsymbol{\psi}_{t,i-1}^{(2)}}\nabla_{\boldsymbol{\sigma}_{t,i-1}^2}\ell \tag{203}$$

$$= \tfrac{1}{2}\left(\boldsymbol{\psi}_{t,i-1}^{(2)}\right)^{-2}\boldsymbol{\psi}_{t,i-1}^{(1)}\nabla_{\boldsymbol{\mu}_{t,i-1}}\ell + \tfrac{1}{2}\left(\boldsymbol{\psi}_{t,i-1}^{(2)}\right)^{-2}\nabla_{\boldsymbol{\sigma}_{t,i-1}^2}\ell \tag{204}$$

$$= 2\boldsymbol{\sigma}_{t,i-1}^2\boldsymbol{\mu}_{t,i-1}\nabla_{\boldsymbol{\mu}_{t,i-1}}\ell + 2\boldsymbol{\sigma}_{t,i-1}^4\nabla_{\boldsymbol{\sigma}_{t,i-1}^2}\ell \tag{205}$$

Therefore

$$\nabla_{\boldsymbol{\psi}_{t,i-1}^{(1)}}\mathbb{E}_{\boldsymbol{\theta}_t\sim q_{\boldsymbol{\psi}_{t,i-1}}}\left[\log p\left(\boldsymbol{y}_t|f_t\left(\boldsymbol{\theta}_t\right)\right)\right] = \boldsymbol{\sigma}_{t,i-1}^2\boldsymbol{g}_{t,i} \tag{206}$$

$$\nabla_{\boldsymbol{\psi}_{t,i-1}^{(2)}}\mathbb{E}_{\boldsymbol{\theta}_t\sim q_{\boldsymbol{\psi}_{t,i-1}}}\left[\log p\left(\boldsymbol{y}_t|f_t\left(\boldsymbol{\theta}_t\right)\right)\right] = 2\boldsymbol{\sigma}_{t,i-1}^2\boldsymbol{\mu}_{t,i-1}\boldsymbol{g}_{t,i} + \boldsymbol{\sigma}_{t,i-1}^4\text{diag}\left(\mathbf{G}_{t,i}\right) \tag{207}$$

and

$$\nabla_{\boldsymbol{\psi}_{t,i-1}^{(1)}} D_{\mathbb{KL}}\big(q_{\boldsymbol{\psi}_{t,i-1}}|q_{\boldsymbol{\psi}_{t|t-1}}\big) = \boldsymbol{\sigma}_{t,i-1}^2 \boldsymbol{\sigma}_{t|t-1}^{-2}\big(\boldsymbol{\mu}_{t,i-1} - \boldsymbol{\mu}_{t|t-1}\big) \tag{208}$$

$$\nabla_{\boldsymbol{\psi}_{t,i-1}^{(2)}} D_{\mathbb{KL}}\big(q_{\boldsymbol{\psi}_{t,i-1}}|q_{\boldsymbol{\psi}_{t|t-1}}\big) = 2\boldsymbol{\sigma}_{t,i-1}^2 \boldsymbol{\sigma}_{t|t-1}^{-2}\boldsymbol{\mu}_{t,i-1}\big(\boldsymbol{\mu}_{t,i-1} - \boldsymbol{\mu}_{t|t-1}\big)$$
$$+ \boldsymbol{\sigma}_{t,i-1}^4\left(\boldsymbol{\sigma}_{t|t-1}^{-2} - \boldsymbol{\sigma}_{t,i-1}^{-2}\right) \tag{209}$$

Our implementations often make use of the following trick: Suppose $\mathbf{A} \in \mathbb{R}^{n \times m}$ and $\mathbf{B} \in \mathbb{R}^{m \times n}$. Then we can efficiently compute $\mathrm{diag}(\mathbf{AB})$ in $O(mn)$ time using $(\mathbf{AB})_{ii} = \sum_{j=1}^{M} A_{ij}B_{ji}$.

For MC-HESS methods, we approximate the diagonal of the Hessian for each MC sample $\hat{\boldsymbol{\theta}}_t^{(m)}$ using Hutchinson's trace estimation method [Hutchinson, 1989] which has been used in other DNN optimization papers such as adahessian Yao et al. [2021]. This involves an extra inner loop with size denoted $N$.

### E.3.1   BONG Diag

Substituting Eqs. (196) and (197) into Eq. (5) gives

$$\boldsymbol{\psi}_t^{(1)} = \boldsymbol{\psi}_{t|t-1}^{(1)} + \boldsymbol{g}_t - \mathrm{diag}\left(\mathbf{G}_t\right)\boldsymbol{\mu}_{t|t-1} \tag{210}$$

$$\boldsymbol{\psi}_t^{(2)} = \boldsymbol{\psi}_{t|t-1}^{(2)} + \tfrac{1}{2}\mathrm{diag}\left(\mathbf{G}_t\right) \tag{211}$$

Translating to $\left(\boldsymbol{\mu}_t, \boldsymbol{\sigma}_t^2\right)$ gives the BONG-DIAG update

$$\boldsymbol{\mu}_t = \boldsymbol{\mu}_{t|t-1} + \boldsymbol{\sigma}_t^2 \boldsymbol{g}_t \tag{212}$$

$$\boldsymbol{\sigma}_t^{-2} = \boldsymbol{\sigma}_{t|t-1}^{-2} - \mathrm{diag}\left(\mathbf{G}_t\right) \tag{213}$$

This update takes $O(MP)$ to estimate $\mathbf{G}_t$ using $\mathbf{G}_t^{\text{MC-EF}}$, $O(NMP)$ using $\mathbf{G}_t^{\text{MC-HESS}}$ and Hutchinson's method, and $O(P)$ using $\mathbf{G}_t^{\text{LIN-EF}}$.

### E.3.2   BONG-LIN-HESS Diag (VD-EKF)

Applying Proposition 4.2 to Eqs. (210) and (211) gives the BONG-LIN-HESS-DIAG update

$$\boldsymbol{\mu}_t = \boldsymbol{\mu}_{t|t-1} + \boldsymbol{\sigma}_t^2\left(\mathbf{H}_t^\intercal \mathbf{R}_t^{-1}\left(\boldsymbol{y}_t - \hat{\boldsymbol{y}}_t\right)\right) \tag{214}$$

$$\boldsymbol{\sigma}_t^{-2} = \boldsymbol{\sigma}_{t|t-1}^{-2} + \mathrm{diag}\left(\mathbf{H}_t^\intercal \mathbf{R}_t^{-1}\mathbf{H}_t\right) \tag{215}$$

This update is equivalent to VD-EKF [Chang et al., 2022] and takes $O(C^2 P)$.

### E.3.3   BLR Diag (VON)

Substituting Eqs. (196) to (199) into Eq. (33) gives

$$\boldsymbol{\psi}_{t,i}^{(1)} = \boldsymbol{\psi}_{t,i-1}^{(1)} + \alpha\left(\boldsymbol{g}_{t,i} - \mathrm{diag}\left(\mathbf{G}_{t,i}\right)\boldsymbol{\mu}_{t,i-1} - \boldsymbol{\sigma}_{t,i-1}^{-2}\boldsymbol{\mu}_{t,i-1} + \boldsymbol{\sigma}_{t|t-1}^{-2}\boldsymbol{\mu}_{t|t-1}\right) \tag{216}$$

$$\boldsymbol{\psi}_{t,i}^{(2)} = \boldsymbol{\psi}_{t,i-1}^{(2)} + \frac{\alpha}{2}\left(\mathrm{diag}\left(\mathbf{G}_{t,i}\right) + \boldsymbol{\sigma}_{t,i-1}^{-2} - \boldsymbol{\sigma}_{t|t-1}^{-2}\right) \tag{217}$$

Translating to $\left(\boldsymbol{\mu}_{t,i}, \boldsymbol{\sigma}_{t,i}^2\right)$ gives the BLR-DIAG update

$$\boldsymbol{\mu}_{t,i} = \boldsymbol{\mu}_{t,i-1} + \alpha\boldsymbol{\sigma}_{t,i}^2 \boldsymbol{\sigma}_{t|t-1}^{-2}\left(\boldsymbol{\mu}_{t|t-1} - \boldsymbol{\mu}_{t,i-1}\right) + \alpha\boldsymbol{\sigma}_{t,i}^2 \boldsymbol{g}_{t,i} \tag{218}$$

$$\boldsymbol{\sigma}_{t,i}^{-2} = (1-\alpha)\boldsymbol{\sigma}_{t,i-1}^{-2} + \alpha\boldsymbol{\sigma}_{t|t-1}^{-2} - \alpha\,\mathrm{diag}\left(\mathbf{G}_{t,i}\right) \tag{219}$$

This update takes $O(MP)$ per iteration to estimate $\mathbf{G}_t$ using $\mathbf{G}_t^{\text{MC-EF}}$, $O(NMP)$ per iteration using $\mathbf{G}_t^{\text{MC-HESS}}$ and Hutchinson's method, and $O(P)$ per iteration using $\mathbf{G}_t^{\text{LIN-EF}}$.

The MC-HESS and MC-EF versions of this update are respectively equivalent to VON and VOGN [Khan et al., 2018b] in the batch setting where we replace $q_{\boldsymbol{\psi}_{t|t-1}}$ with a spherical prior $\mathcal{N}(\mathbf{0}, \lambda^{-1}\mathbf{I}_P)$ (see Appendix E.6).

### E.3.4 BLR-LIN-HESS Diag

Applying Proposition 4.2 to Eqs. (218) and (219) gives the BLR-LIN-HESS-DIAG update

$$\boldsymbol{\mu}_{t,i} = \boldsymbol{\mu}_{t,i-1} + \alpha \boldsymbol{\sigma}_{t,i}^2 \boldsymbol{\sigma}_{t|t-1}^{-2} \left( \boldsymbol{\mu}_{t|t-1} - \boldsymbol{\mu}_{t,i-1} \right) + \alpha \boldsymbol{\sigma}_{t,i}^2 \left( \mathbf{H}_{t,i}^{\mathsf{T}} \mathbf{R}_{t,i}^{-1} \left( \boldsymbol{y}_t - \hat{\boldsymbol{y}}_{t,i} \right) \right) \tag{220}$$

$$\boldsymbol{\sigma}_{t,i}^{-2} = (1-\alpha)\,\boldsymbol{\sigma}_{t,i-1}^{-2} + \alpha \boldsymbol{\sigma}_{t|t-1}^{-2} + \alpha \operatorname{diag}\left( \mathbf{H}_{t,i}^{\mathsf{T}} \mathbf{R}_{t,i}^{-1} \mathbf{H}_{t,i} \right) \tag{221}$$

This update takes $O(C^2 P)$ per iteration.

### E.3.5 BOG Diag

Substituting Eqs. (206) and (207) into Eq. (31) gives

$$\boldsymbol{\psi}_t^{(1)} = \boldsymbol{\psi}_{t|t-1}^{(1)} + \alpha \boldsymbol{\sigma}_{t|t-1}^2 \boldsymbol{g}_t \tag{222}$$

$$\boldsymbol{\psi}_t^{(2)} = \boldsymbol{\psi}_{t|t-1}^{(2)} + 2\alpha \boldsymbol{\sigma}_{t|t-1}^2 \boldsymbol{\mu}_{t|t-1} \boldsymbol{g}_t + \alpha \boldsymbol{\sigma}_{t|t-1}^4 \operatorname{diag}\left( \mathbf{G}_t \right) \tag{223}$$

Translating to $\left( \boldsymbol{\mu}_t, \boldsymbol{\sigma}_t^2 \right)$ gives the BOG-DIAG update

$$\boldsymbol{\mu}_t = \boldsymbol{\sigma}_t^2 \boldsymbol{\sigma}_{t|t-1}^{-2} \boldsymbol{\mu}_{t|t-1} + \alpha \boldsymbol{\sigma}_t^2 \boldsymbol{\sigma}_{t|t-1}^2 \boldsymbol{g}_t \tag{224}$$

$$\boldsymbol{\sigma}_t^{-2} = \boldsymbol{\sigma}_{t|t-1}^{-2} - 4\alpha \boldsymbol{\sigma}_{t|t-1}^2 \boldsymbol{\mu}_{t|t-1} \boldsymbol{g}_t - 2\alpha \boldsymbol{\sigma}_{t|t-1}^4 \operatorname{diag}\left( \mathbf{G}_t \right) \tag{225}$$

This update takes $O(MP)$ to estimate $\mathbf{G}_t$ using $\mathbf{G}_t^{\text{MC-EF}}$, $O(NMP)$ using $\mathbf{G}_t^{\text{MC-HESS}}$ and Hutchinson's method, and $O(P)$ using $\mathbf{G}_t^{\text{LIN-EF}}$.

### E.3.6 BOG-LIN-HESS Diag

Applying Proposition 4.2 to Eqs. (224) and (225) gives the BOG-LIN-HESS-DIAG update

$$\boldsymbol{\mu}_t = \boldsymbol{\sigma}_t^2 \boldsymbol{\sigma}_{t|t-1}^{-2} \boldsymbol{\mu}_{t|t-1} + \alpha \boldsymbol{\sigma}_t^2 \boldsymbol{\sigma}_{t|t-1}^2 \left( \mathbf{H}_t^{\mathsf{T}} \mathbf{R}_t^{-1} \left( \boldsymbol{y}_t - \hat{\boldsymbol{y}}_t \right) \right) \tag{226}$$

$$\boldsymbol{\sigma}_t^{-2} = \boldsymbol{\sigma}_{t|t-1}^{-2} - 4\alpha \boldsymbol{\sigma}_{t|t-1}^2 \boldsymbol{\mu}_{t|t-1} \left( \mathbf{H}_t^{\mathsf{T}} \mathbf{R}_t^{-1} \left( \boldsymbol{y}_t - \hat{\boldsymbol{y}}_t \right) \right) + 2\alpha \boldsymbol{\sigma}_{t|t-1}^4 \operatorname{diag}\left( \mathbf{H}_t^{\mathsf{T}} \mathbf{R}_t^{-1} \mathbf{H}_t \right) \tag{227}$$

This update takes $O(C^2 P)$.

### E.3.7 BBB Diag

Substituting Eqs. (206) to (209) into Eq. (34) gives

$$\boldsymbol{\psi}_{t,i}^{(1)} = \boldsymbol{\psi}_{t,i-1}^{(1)} + \alpha \boldsymbol{\sigma}_{t,i-1}^2 \boldsymbol{g}_{t,i} - \alpha \boldsymbol{\sigma}_{t,i-1}^2 \boldsymbol{\sigma}_{t|t-1}^{-2} \left( \boldsymbol{\mu}_{t,i-1} - \boldsymbol{\mu}_{t|t-1} \right) \tag{228}$$

$$\begin{aligned} \boldsymbol{\psi}_t^{(2)} = {}& \boldsymbol{\psi}_{t,i-1}^{(2)} + 2\alpha \boldsymbol{\sigma}_{t,i-1}^2 \boldsymbol{\mu}_{t,i-1} \boldsymbol{g}_{t,i} + \alpha \boldsymbol{\sigma}_{t,i-1}^4 \operatorname{diag}\left( \mathbf{G}_{t,i} \right) \\ & - 2\alpha \boldsymbol{\sigma}_{t,i-1}^2 \boldsymbol{\sigma}_{t|t-1}^{-2} \boldsymbol{\mu}_{t,i-1} \left( \boldsymbol{\mu}_{t,i-1} - \boldsymbol{\mu}_{t|t-1} \right) - \alpha \boldsymbol{\sigma}_{t,i-1}^4 \left( \boldsymbol{\sigma}_{t|t-1}^{-2} - \boldsymbol{\sigma}_{t,i-1}^{-2} \right) \end{aligned} \tag{229}$$

Translating to $\left( \boldsymbol{\mu}_{t,i}, \boldsymbol{\Sigma}_{t,i} \right)$ gives the BBB-DIAG update

$$\boldsymbol{\mu}_{t,i} = \boldsymbol{\sigma}_{t,i}^2 \boldsymbol{\sigma}_{t,i-1}^{-2} \boldsymbol{\mu}_{t,i-1} + \alpha \boldsymbol{\sigma}_{t,i}^2 \boldsymbol{\sigma}_{t,i-1}^2 \boldsymbol{g}_{t,i} + \alpha \boldsymbol{\sigma}_{t,i}^2 \boldsymbol{\sigma}_{t,i-1}^2 \boldsymbol{\sigma}_{t|t-1}^{-2} \left( \boldsymbol{\mu}_{t|t-1} - \boldsymbol{\mu}_{t,i-1} \right) \tag{230}$$

$$\begin{aligned} \boldsymbol{\sigma}_{t,i}^{-2} = {}& \boldsymbol{\sigma}_{t,i-1}^{-2} - 4\alpha \boldsymbol{\sigma}_{t,i-1}^2 \boldsymbol{\mu}_{t,i-1} \boldsymbol{g}_{t,i} - 2\alpha \boldsymbol{\sigma}_{t,i-1}^4 \operatorname{diag}\left( \mathbf{G}_{t,i} \right) \\ & + 4\alpha \boldsymbol{\sigma}_{t,i-1}^2 \boldsymbol{\sigma}_{t|t-1}^{-2} \boldsymbol{\mu}_{t,i-1} \left( \boldsymbol{\mu}_{t,i-1} - \boldsymbol{\mu}_{t|t-1} \right) + 2\alpha \boldsymbol{\sigma}_{t,i-1}^4 \left( \boldsymbol{\sigma}_{t|t-1}^{-2} - \boldsymbol{\sigma}_{t,i-1}^{-2} \right) \end{aligned} \tag{231}$$

This update takes $O(MP)$ per iteration to estimate $\mathbf{G}_t$ using $\mathbf{G}_t^{\text{MC-EF}}$, $O(NMP)$ per iteration using $\mathbf{G}_t^{\text{MC-HESS}}$ and Hutchinson's method, and $O(P)$ per iteration using $\mathbf{G}_t^{\text{LIN-EF}}$.

### E.3.8 BBB-LIN-HESS Diag

Applying Proposition 4.2 to Eqs. (230) and (231) gives the BBB-LIN-HESS-DIAG update

$$\begin{aligned} \boldsymbol{\mu}_{t,i} = {}& \boldsymbol{\sigma}_{t,i}^2 \boldsymbol{\sigma}_{t,i-1}^{-2} \boldsymbol{\mu}_{t,i-1} + \alpha \boldsymbol{\sigma}_{t,i}^2 \boldsymbol{\sigma}_{t,i-1}^2 \left( \mathbf{H}_{t,i}^{\mathsf{T}} \mathbf{R}_{t,i}^{-1} \left( \boldsymbol{y}_t - \hat{\boldsymbol{y}}_{t,i} \right) \right) \\ & + \alpha \boldsymbol{\sigma}_{t,i}^2 \boldsymbol{\sigma}_{t,i-1}^2 \boldsymbol{\sigma}_{t|t-1}^{-2} \left( \boldsymbol{\mu}_{t|t-1} - \boldsymbol{\mu}_{t,i-1} \right) \end{aligned} \tag{232}$$

$$\begin{aligned} \boldsymbol{\sigma}_{t,i}^{-2} = {}& \boldsymbol{\sigma}_{t,i-1}^{-2} - 4\alpha \boldsymbol{\sigma}_{t,i-1}^2 \boldsymbol{\mu}_{t,i-1} \left( \mathbf{H}_{t,i}^{\mathsf{T}} \mathbf{R}_{t,i}^{-1} \left( \boldsymbol{y}_t - \hat{\boldsymbol{y}}_{t,i} \right) \right) + 2\alpha \boldsymbol{\sigma}_{t,i-1}^4 \operatorname{diag}\left( \mathbf{H}_{t,i}^{\mathsf{T}} \mathbf{R}_{t,i}^{-1} \mathbf{H}_{t,i} \right) \\ & + 4\alpha \boldsymbol{\sigma}_{t,i-1}^2 \boldsymbol{\sigma}_{t|t-1}^{-2} \boldsymbol{\mu}_{t,i-1} \left( \boldsymbol{\mu}_{t,i-1} - \boldsymbol{\mu}_{t|t-1} \right) + 2\alpha \boldsymbol{\sigma}_{t,i-1}^4 \boldsymbol{\sigma}_{t|t-1}^{-2} - 2\alpha \boldsymbol{\sigma}_{t,i-1}^2 \end{aligned} \tag{233}$$

This update takes $O(C^2 P)$ per iteration.

## E.4 Diagonal Gaussian, Moment parameters

Throughout this subsection, vector multiplication and exponents are elementwise.

The Bonnet and Price theorems give

$$\nabla_{\boldsymbol{\mu}_{t,i-1}} \mathbb{E}_{\boldsymbol{\theta}_t \sim q_{\boldsymbol{\psi}_{t,i-1}}} \left[\log p\left(\boldsymbol{y}_t | f_t\left(\boldsymbol{\theta}_t\right)\right)\right] = \boldsymbol{g}_{t,i} \tag{234}$$

$$\nabla_{\boldsymbol{\sigma}_{t,i-1}^2} \mathbb{E}_{\boldsymbol{\theta}_t \sim q_{\boldsymbol{\psi}_{t,i-1}}} \left[\log p\left(\boldsymbol{y}_t | f_t\left(\boldsymbol{\theta}_t\right)\right)\right] = \tfrac{1}{2}\mathrm{diag}\left(\mathbf{G}_{t,i}\right) \tag{235}$$

From Appendix E.3 we have

$$\nabla_{\boldsymbol{\mu}_{t,i-1}} D_{\mathbb{KL}}\left(q_{\boldsymbol{\psi}_{t,i-1}} | q_{\boldsymbol{\psi}_{t|t-1}}\right) = \boldsymbol{\sigma}_{t|t-1}^{-2}\left(\boldsymbol{\mu}_{t,i-1} - \boldsymbol{\mu}_{t|t-1}\right) \tag{236}$$

$$\nabla_{\boldsymbol{\sigma}_{t,i-1}^2} D_{\mathbb{KL}}\left(q_{\boldsymbol{\psi}_{t,i-1}} | q_{\boldsymbol{\psi}_{t|t-1}}\right) = \tfrac{1}{2}\left(\boldsymbol{\sigma}_{t|t-1}^{-2} - \boldsymbol{\sigma}_{t,i-1}^{-2}\right) \tag{237}$$

We write the Fisher with respect to the moment parameters $\boldsymbol{\psi} = (\boldsymbol{\mu}, \boldsymbol{\sigma}^2)$ as a block matrix:

$$\mathbf{F}_{\boldsymbol{\psi}} = \left[\begin{array}{cc} \mathbf{F}_{\boldsymbol{\mu},\boldsymbol{\mu}} & \mathbf{F}_{\boldsymbol{\mu},\boldsymbol{\sigma}^2} \\ \mathbf{F}_{\boldsymbol{\sigma}^2,\boldsymbol{\mu}} & \mathbf{F}_{\boldsymbol{\sigma}^2,\boldsymbol{\sigma}^2} \end{array}\right] \tag{238}$$

The blocks can be calculated by the second-order Fisher formula

$$\mathbf{F}_{\boldsymbol{\mu},\boldsymbol{\mu}} = -\mathbb{E}_{q_{\boldsymbol{\psi}}}[\nabla_{\boldsymbol{\mu},\boldsymbol{\mu}} \log q_{\boldsymbol{\psi}}(\boldsymbol{\theta})] \tag{239}$$

$$= \mathrm{Diag}\left(\boldsymbol{\sigma}^{-2}\right) \tag{240}$$

$$\mathbf{F}_{\boldsymbol{\mu},\boldsymbol{\sigma}^2} = -\mathbb{E}_{q_{\boldsymbol{\psi}}}\left[\nabla_{\boldsymbol{\mu},\boldsymbol{\sigma}^2} \log q_{\boldsymbol{\psi}}(\boldsymbol{\theta})\right] \tag{241}$$

$$= \mathbb{E}_{q_{\boldsymbol{\psi}}}\left[\mathrm{Diag}\left((\boldsymbol{\theta} - \boldsymbol{\mu})\boldsymbol{\sigma}^{-4}\right)\right] \tag{242}$$

$$= \mathbf{0} \tag{243}$$

$$\mathbf{F}_{\boldsymbol{\sigma}^2,\boldsymbol{\sigma}^2} = -\mathbb{E}_{q_{\boldsymbol{\psi}}}\left[\nabla_{\boldsymbol{\sigma}^2,\boldsymbol{\sigma}^2} \log q_{\boldsymbol{\psi}}(\boldsymbol{\theta})\right] \tag{244}$$

$$= -\mathbb{E}_{q_{\boldsymbol{\psi}}}\left[\mathrm{Diag}\left(-(\boldsymbol{\mu} - \boldsymbol{\theta})^2 \boldsymbol{\sigma}^{-6} + \tfrac{1}{2}\boldsymbol{\sigma}^{-4}\right)\right] \tag{245}$$

$$= \tfrac{1}{2}\mathrm{Diag}\left(\boldsymbol{\sigma}^{-4}\right) \tag{246}$$

Therefore the preconditioner for the NGD methods is

$$\mathbf{F}_{\boldsymbol{\psi}_{t,i-1}}^{-1} = \left[\begin{array}{cc} \mathrm{Diag}\left(\sigma_{t,i-1}^2\right) & \mathbf{0} \\ \mathbf{0} & 2\mathrm{Diag}\left(\sigma_{t,i-1}^4\right) \end{array}\right] \tag{247}$$

### E.4.1 BONG Diag, Moment

Substituting Eqs. (234), (235) and (247) into Eq. (3) gives the BONG-DIAG_MOM update

$$\boldsymbol{\mu}_t = \boldsymbol{\mu}_{t|t-1} + \boldsymbol{\sigma}_{t|t-1}^2 \boldsymbol{g}_t \tag{248}$$

$$\boldsymbol{\sigma}_t^2 = \boldsymbol{\sigma}_{t|t-1}^2 + \boldsymbol{\sigma}_{t|t-1}^4 \mathrm{diag}\left(\mathbf{G}_t\right) \tag{249}$$

This update takes $O(MP)$ to estimate $\mathbf{G}_t$ using $\mathbf{G}_t^{\text{MC-EF}}$, $O(NMP)$ using $\mathbf{G}_t^{\text{MC-HESS}}$ and Hutchinson's method, and $O(P)$ using $\mathbf{G}_t^{\text{LIN-EF}}$.

### E.4.2 BONG-LIN-HESS Diag, Momemt

Applying Proposition 4.2 to Eqs. (248) and (249) gives the BONG-LIN-HESS-DIAG_MOM update

$$\boldsymbol{\mu}_t = \boldsymbol{\mu}_{t|t-1} + \boldsymbol{\sigma}_{t|t-1}^2 \left(\mathbf{H}_t^{\mathsf{T}}\mathbf{R}_t^{-1}(\boldsymbol{y}_t - \hat{\boldsymbol{y}}_t)\right) \tag{250}$$

$$\boldsymbol{\sigma}_t^2 = \boldsymbol{\sigma}_{t|t-1}^2 - \boldsymbol{\sigma}_{t|t-1}^4 \mathrm{diag}\left(\mathbf{H}_t^{\mathsf{T}}\mathbf{R}_t^{-1}\mathbf{H}_t\right) \tag{251}$$

This update takes $O(C^2 P)$.

### E.4.3 BLR Diag, Moment

Substituting Eqs. (234) to (237) and (247) into Eq. (32) gives the BLR-DIAG_MOM update

$$\boldsymbol{\mu}_{t,i} = \boldsymbol{\mu}_{t,i-1} + \alpha\boldsymbol{\sigma}_{t,i-1}^2 \boldsymbol{\sigma}_{t|t-1}^{-2}\left(\boldsymbol{\mu}_{t|t-1} - \boldsymbol{\mu}_{t,i-1}\right) + \alpha\boldsymbol{\sigma}_{t,i-1}^2 \boldsymbol{g}_{t,i} \tag{252}$$

$$\boldsymbol{\sigma}_{t,i}^2 = \boldsymbol{\sigma}_{t,i-1}^2 + \alpha\boldsymbol{\sigma}_{t,i-1}^4\left(\boldsymbol{\sigma}_{t,i-1}^{-2} - \boldsymbol{\sigma}_{t|t-1}^{-2}\right) + \alpha\boldsymbol{\sigma}_{t,i-1}^4 \mathrm{diag}\left(\mathbf{G}_{t,i}\right) \tag{253}$$

This update takes $O(MP)$ per iteration to estimate $\mathbf{G}_t$ using $\mathbf{G}_t^{\text{MC-EF}}$, $O(NMP)$ per iteration using $\mathbf{G}_t^{\text{MC-HESS}}$ and Hutchinson's method, and $O(P)$ per iteration using $\mathbf{G}_t^{\text{LIN-EF}}$.

### E.4.4 BLR-LIN-HESS Diag, Moment

Applying Proposition 4.2 to Eqs. (252) and (253) gives the BLR-LIN-HESS-DIAG_MOM update

$$\boldsymbol{\mu}_{t,i} = \boldsymbol{\mu}_{t,i-1} + \alpha\boldsymbol{\sigma}_{t,i-1}^2\boldsymbol{\sigma}_{t|t-1}^{-2}\left(\boldsymbol{\mu}_{t|t-1} - \boldsymbol{\mu}_{t,i-1}\right) + \alpha\boldsymbol{\sigma}_{t,i-1}^2\left(\mathbf{H}_{t,i}^\intercal\mathbf{R}_{t,i}^{-1}(\boldsymbol{y}_t - \hat{\boldsymbol{y}}_{t,i})\right) \tag{254}$$

$$\boldsymbol{\sigma}_{t,i}^2 = \boldsymbol{\sigma}_{t,i-1}^2 + \alpha\boldsymbol{\sigma}_{t,i-1}^4\left(\boldsymbol{\sigma}_{t,i-1}^{-2} - \boldsymbol{\sigma}_{t|t-1}^{-2}\right) - \alpha\boldsymbol{\sigma}_{t,i-1}^4\mathrm{diag}\left(\mathbf{H}_{t,i}^\intercal\mathbf{R}_{t,i}^{-1}\mathbf{H}_{t,i}\right) \tag{255}$$

This update takes $O(C^2P)$ per iteration.

### E.4.5 BOG Diag, Moment

Substituting Eqs. (234) and (235) into Eq. (31) gives the BOG-DIAG_MOM update

$$\boldsymbol{\mu}_t = \boldsymbol{\mu}_{t|t-1} + \alpha\boldsymbol{g}_t \tag{256}$$

$$\boldsymbol{\sigma}_t^2 = \boldsymbol{\sigma}_{t|t-1}^2 + \frac{\alpha}{2}\mathrm{diag}\left(\mathbf{G}_t\right) \tag{257}$$

This update takes $O(MP)$ to estimate $\mathbf{G}_t$ using $\mathbf{G}_t^{\text{MC-EF}}$, $O(NMP)$ using $\mathbf{G}_t^{\text{MC-HESS}}$ and Hutchinson's method, and $O(P)$ using $\mathbf{G}_t^{\text{LIN-EF}}$.

### E.4.6 BOG-LIN-HESS Diag, Moment

Applying Proposition 4.2 to Eqs. (256) and (257) gives the BOG-LIN-HESS-DIAG_MOM update

$$\boldsymbol{\mu}_t = \boldsymbol{\mu}_{t|t-1} + \alpha\mathbf{H}_t^\intercal\mathbf{R}_t^{-1}(\boldsymbol{y}_t - \hat{\boldsymbol{y}}_t) \tag{258}$$

$$\boldsymbol{\sigma}_t^2 = \boldsymbol{\sigma}_{t|t-1}^2 - \frac{\alpha}{2}\mathrm{diag}\left(\mathbf{H}_t^\intercal\mathbf{R}_t^{-1}\mathbf{H}_t\right) \tag{259}$$

This update takes $O(C^2P)$.

### E.4.7 BBB Diag, Moment

Substituting Eqs. (234) to (237) into Eq. (34) gives the BBB-DIAG_MOM

$$\boldsymbol{\mu}_{t,i} = \boldsymbol{\mu}_{t,i-1} + \alpha\boldsymbol{\sigma}_{t|t-1}^{-2}\left(\boldsymbol{\mu}_{t|t-1} - \boldsymbol{\mu}_{t,i-1}\right) + \alpha\boldsymbol{g}_{t,i} \tag{260}$$

$$\boldsymbol{\sigma}_t^2 = \boldsymbol{\sigma}_{t,i-1}^2 + \frac{\alpha}{2}\left(\boldsymbol{\sigma}_{t,i-1}^{-2} - \boldsymbol{\sigma}_{t|t-1}^{-2}\right) + \frac{\alpha}{2}\mathrm{diag}\left(\mathbf{G}_{t,i}\right) \tag{261}$$

This update takes $O(MP)$ per iteration to estimate $\mathbf{G}_t$ using $\mathbf{G}_t^{\text{MC-EF}}$, $O(NMP)$ per iteration using $\mathbf{G}_t^{\text{MC-HESS}}$ and Hutchinson's method, and $O(P)$ per iteration using $\mathbf{G}_t^{\text{LIN-EF}}$.

This is similar to the original diagonal Gaussian method in [Blundell et al., 2015] except (1) they reparameterize $\boldsymbol{\sigma} = \log(1 + \exp(\boldsymbol{\rho}))$ (elementwise) and do GD on $(\boldsymbol{\mu}, \boldsymbol{\rho})$ instead of $(\boldsymbol{\mu}, \boldsymbol{\sigma}^2)$, and (2) they use the reparameterization trick instead of Price's theorem for calculating the gradient with respect to $\boldsymbol{\rho}$ (via $\boldsymbol{\sigma}$).

### E.4.8 BBB-LIN-HESS Diag, Moment

Applying Proposition 4.2 to Eqs. (260) and (261) gives the BBB-LIN-HESS-DIAG_MOM update

$$\boldsymbol{\mu}_{t,i} = \boldsymbol{\mu}_{t,i-1} + \alpha\boldsymbol{\sigma}_{t|t-1}^{-2}\left(\boldsymbol{\mu}_{t|t-1} - \boldsymbol{\mu}_{t,i-1}\right) + \alpha\mathbf{H}_{t,i}^\intercal\mathbf{R}_{t,i}^{-1}(\boldsymbol{y}_t - \hat{\boldsymbol{y}}_{t,i}) \tag{262}$$

$$\boldsymbol{\sigma}_t^2 = \boldsymbol{\sigma}_{t,i-1}^2 + \frac{\alpha}{2}\left(\boldsymbol{\sigma}_{t,i-1}^{-2} - \boldsymbol{\sigma}_{t|t-1}^{-2}\right) - \frac{\alpha}{2}\mathrm{diag}\left(\mathbf{H}_{t,i}^\intercal\mathbf{R}_{t,i}^{-1}\mathbf{H}_{t,i}\right) \tag{263}$$

This update takes $O(C^2P)$ per iteration.

### E.5 Diagonal plus low rank

Assume the prior is given by

$$q_{\boldsymbol{\psi}_{t|t-1}}(\boldsymbol{\theta}_t) = \mathcal{N}\left(\boldsymbol{\theta}_t | \boldsymbol{\mu}_{t|t-1}, \left(\boldsymbol{\Upsilon}_{t|t-1} + \mathbf{W}_{t|t-1}\mathbf{W}_{t|t-1}^\intercal\right)^{-1}\right) \tag{264}$$

with $\mathbf{W} \in \mathbb{R}^{P \times R}$ and diagonal $\mathbf{\Upsilon}_{t|t-1} \in \mathbb{R}^{P \times P}$. We sometimes abuse notation by writing $\mathbf{\Upsilon}_{t|t-1}$ for the vector $\mathrm{diag}\left(\mathbf{\Upsilon}_{t|t-1}\right)$ when the meaning is clear from context.

Substituting the DLR form in the gradients for the KL divergence derived in Appendix E.2 gives

$$\nabla_{\boldsymbol{\mu}_{t,i-1}} D_{\mathbb{KL}}\left(q_{\boldsymbol{\psi}_{t,i-1}} | q_{\boldsymbol{\psi}_{t|t-1}}\right) = \left(\mathbf{\Upsilon}_{t|t-1} + \mathbf{W}_{t|t-1}\mathbf{W}_{t|t-1}^{\mathsf{T}}\right)\left(\boldsymbol{\mu}_{t,i-1} - \boldsymbol{\mu}_{t|t-1}\right) \tag{265}$$

$$\nabla_{\boldsymbol{\Sigma}_{t,i-1}} D_{\mathbb{KL}}\left(q_{\boldsymbol{\psi}_{t,i-1}} | q_{\boldsymbol{\psi}_{t|t-1}}\right) = \tfrac{1}{2}\left(\mathbf{\Upsilon}_{t|t-1} - \mathbf{\Upsilon}_{t,i-1} + \mathbf{W}_{t|t-1}\mathbf{W}_{t|t-1}^{\mathsf{T}} - \mathbf{W}_{t,i-1}\mathbf{W}_{t,i-1}^{\mathsf{T}}\right) \tag{266}$$

For any function $\ell$ the chain rule gives

$$\nabla_{\mathbf{\Upsilon}_{t|t-1}} \ell = -\mathrm{diag}\left(\left(\mathbf{\Upsilon}_{t|t-1} + \mathbf{W}_{t|t-1}\mathbf{W}_{t|t-1}^{\mathsf{T}}\right)^{-1}\left(\nabla_{\boldsymbol{\Sigma}_{t|t-1}}\ell\right)\left(\mathbf{\Upsilon}_{t|t-1} + \mathbf{W}_{t|t-1}\mathbf{W}_{t|t-1}^{\mathsf{T}}\right)^{-1}\right) \tag{267}$$

$$\nabla_{\mathbf{W}_{t|t-1}} \ell = -2\left(\mathbf{\Upsilon}_{t|t-1} + \mathbf{W}_{t|t-1}\mathbf{W}_{t|t-1}^{\mathsf{T}}\right)^{-1}\left(\nabla_{\boldsymbol{\Sigma}_{t|t-1}}\ell\right)\left(\mathbf{\Upsilon}_{t|t-1} + \mathbf{W}_{t|t-1}\mathbf{W}_{t|t-1}^{\mathsf{T}}\right)^{-1}\mathbf{W}_{t|t-1} \tag{268}$$

Therefore the gradients we need are

$$\nabla_{\boldsymbol{\mu}_{t,i-1}} \mathbb{E}_{\boldsymbol{\theta}_t \sim q_{\boldsymbol{\psi}_{t,i-1}}}\left[\log p\left(\boldsymbol{y}_t | h_t\left(\boldsymbol{\theta}_t\right)\right)\right] = \boldsymbol{g}_{\boldsymbol{\psi}_{t,i}} \tag{269}$$

$$\nabla_{\mathbf{\Upsilon}_{t,i-1}} \mathbb{E}_{\boldsymbol{\theta}_t \sim q_{\boldsymbol{\psi}_{t,i-1}}}\left[\log p\left(\boldsymbol{y}_t | h_t\left(\boldsymbol{\theta}_t\right)\right)\right] = -\tfrac{1}{2}\mathrm{diag}\left(\begin{array}{c}\left(\mathbf{\Upsilon}_{t,i-1} + \mathbf{W}_{t,i-1}\mathbf{W}_{t,i-1}^{\mathsf{T}}\right)^{-1}\mathbf{G}_{t,i} \\ \times\left(\mathbf{\Upsilon}_{t,i-1} + \mathbf{W}_{t,i-1}\mathbf{W}_{t,i-1}^{\mathsf{T}}\right)^{-1}\end{array}\right) \tag{270}$$

$$\nabla_{\mathbf{W}_{t,i-1}} \mathbb{E}_{\boldsymbol{\theta}_t \sim q_{\boldsymbol{\psi}_{t,i-1}}}\left[\log p\left(\boldsymbol{y}_t | h_t\left(\boldsymbol{\theta}_t\right)\right)\right] = -\left(\mathbf{\Upsilon}_{t,i-1} + \mathbf{W}_{t,i-1}\mathbf{W}_{t,i-1}^{\mathsf{T}}\right)^{-1}\mathbf{G}_{t,i}$$
$$\times\left(\mathbf{\Upsilon}_{t,i-1} + \mathbf{W}_{t,i-1}\mathbf{W}_{t,i-1}^{\mathsf{T}}\right)^{-1}\mathbf{W}_{t,i-1} \tag{271}$$

and

$$\nabla_{\mathbf{\Upsilon}_{t,i-1}} D_{\mathbb{KL}}\left(q_{\boldsymbol{\psi}_{t,i-1}} | q_{\boldsymbol{\psi}_{t|t-1}}\right) =$$
$$-\tfrac{1}{2}\mathrm{diag}\left(\begin{array}{c}\left(\mathbf{\Upsilon}_{t,i-1} + \mathbf{W}_{t,i-1}\mathbf{W}_{t,i-1}^{\mathsf{T}}\right)^{-1} \\ \times\left(\mathbf{\Upsilon}_{t|t-1} - \mathbf{\Upsilon}_{t,i-1} + \mathbf{W}_{t|t-1}\mathbf{W}_{t|t-1}^{\mathsf{T}} - \mathbf{W}_{t,i-1}\mathbf{W}_{t,i-1}^{\mathsf{T}}\right) \\ \times\left(\mathbf{\Upsilon}_{t,i-1} + \mathbf{W}_{t,i-1}\mathbf{W}_{t,i-1}^{\mathsf{T}}\right)^{-1}\end{array}\right) \tag{272}$$

$$\nabla_{\mathbf{W}_{t,i-1}} D_{\mathbb{KL}}\left(q_{\boldsymbol{\psi}_{t,i-1}} | q_{\boldsymbol{\psi}_{t|t-1}}\right) = -\left(\mathbf{\Upsilon}_{t,i-1} + \mathbf{W}_{t,i-1}\mathbf{W}_{t,i-1}^{\mathsf{T}}\right)^{-1}$$
$$\times\left(\mathbf{\Upsilon}_{t|t-1} - \mathbf{\Upsilon}_{t,i-1} + \mathbf{W}_{t|t-1}\mathbf{W}_{t|t-1}^{\mathsf{T}} - \mathbf{W}_{t,i-1}\mathbf{W}_{t,i-1}^{\mathsf{T}}\right)$$
$$\times\left(\mathbf{\Upsilon}_{t,i-1} + \mathbf{W}_{t,i-1}\mathbf{W}_{t,i-1}^{\mathsf{T}}\right)^{-1}\mathbf{W}_{t,i-1} \tag{273}$$

The Fisher matrix can be decomposed as a block-diagonal with blocks for $\boldsymbol{\mu}_{t,i}$ and for $\left(\mathbf{\Upsilon}_{t,i}, \mathbf{W}_{t,i}\right)$, but (in contrast to the FC Gaussian case in Eq. (168)) we have not found an efficient way to analytically invert the latter block, which has size $P + RP$. To avoid the $O\left(R^3 P^3\right)$ cost of brute force inversion, we use a different strategy for BONG and BLR of performing the update on the natural parameters of the FC Gaussian as in Appendix E.1 and projecting the result back to rank $R$ using SVD. Specifically, assume the updated precision from applying Eq. (121) for BONG or Eq. (136) for BLR can be written as

$$\tilde{\boldsymbol{\Sigma}}_{t,i}^{-1} = \tilde{\mathbf{\Upsilon}}_{t,i} + \tilde{\mathbf{W}}_{t,i}\tilde{\mathbf{W}}_{t,i}^{\mathsf{T}} \tag{274}$$

and let the SVD of $\tilde{\mathbf{W}}_{t,i}$ be

$$\tilde{\mathbf{W}}_{t,i} = \mathbf{U}_{t,i}\mathbf{\Lambda}_{t,i}\mathbf{V}_{t,i}^{\mathsf{T}} \tag{275}$$

with $\mathbf{U}_{t,i}, \mathbf{V}_{t,i}$ orthogonal and $\mathbf{\Lambda}_{t,i}$ rectangular-diagonal. Following [Mishkin et al., 2018, Chang et al., 2023] we define the update

$$\mathbf{W}_{t,i} = \mathbf{U}_{t,i}\left[:, : R\right]\mathbf{\Lambda}_{t,i}\left[: R, : R\right] \tag{276}$$

$$\mathbf{\Upsilon}_{t,i} = \tilde{\mathbf{\Upsilon}}_{t,i} + \mathrm{diag}\left(\tilde{\mathbf{W}}_{t,i}\tilde{\mathbf{W}}_{t,i}^{\mathsf{T}} - \mathbf{W}_{t,i}\mathbf{W}_{t,i}^{\mathsf{T}}\right) \tag{277}$$

so that $\mathbf{W}_{t,i}$ contains the top $R$ singular vectors and values of the FC posterior and the diagonal is preserved: $\mathrm{diag}\left(\mathbf{\Upsilon}_{t,i} + \mathbf{W}_{t,i}\mathbf{W}_{t,i}^\intercal\right) = \mathrm{diag}\left(\tilde{\mathbf{\Sigma}}_{t,i}^{-1}\right)$. This approach works for MC-EF and LIN-EF methods but not MC-HESS, which we omit.

Finally, in the MC-EF methods we sample from the DLR prior using the routine in [Mishkin et al., 2018, Lambert et al., 2023] which takes $O(R(R + M)P)$.

### E.5.1 BONG DLR

Substituting the DLR prior from Eq. (264) into the FC precision update from Eq. (121) and using the MC-EF approximation yields the posterior precision

$$\tilde{\mathbf{\Sigma}}_t^{-1} = \mathbf{\Upsilon}_{t|t-1} + \mathbf{W}_{t|t-1}\mathbf{W}_{t|t-1}^\intercal - \mathbf{G}_t^{\text{MC-EF}} \tag{278}$$

$$= \tilde{\mathbf{\Upsilon}}_t + \tilde{\mathbf{W}}_t\tilde{\mathbf{W}}_t^\intercal \tag{279}$$

$$\tilde{\mathbf{\Upsilon}}_t = \mathbf{\Upsilon}_{t|t-1} \tag{280}$$

$$\tilde{\mathbf{W}}_t = \left[\mathbf{W}_{t|t-1}, \frac{1}{\sqrt{M}}\hat{\mathbf{G}}_t^{(1:M)}\right] \tag{281}$$

Note $\tilde{\mathbf{W}}_t \in \mathbb{R}^{P \times (R+M)}$. Using this posterior precision in the mean update from Eq. (120) yields

$$\boldsymbol{\mu}_t = \boldsymbol{\mu}_{t|t-1} + \left(\mathbf{\Upsilon}_{t|t-1} + \tilde{\mathbf{W}}_t\tilde{\mathbf{W}}_t^\intercal\right)^{-1}\boldsymbol{g}_t \tag{282}$$

$$= \boldsymbol{\mu}_{t|t-1} + \left(\mathbf{\Upsilon}_{t|t-1}^{-1} - \mathbf{\Upsilon}_{t|t-1}^{-1}\tilde{\mathbf{W}}_t\left(\mathbf{I}_{R+M} + \tilde{\mathbf{W}}_t^\intercal\mathbf{\Upsilon}_{t|t-1}^{-1}\tilde{\mathbf{W}}_t\right)^{-1}\tilde{\mathbf{W}}_t^\intercal\mathbf{\Upsilon}_{t|t-1}^{-1}\right)\boldsymbol{g}_t \tag{283}$$

where the second line comes from the Woodbury identity and can be computed in $O((R + M)^2P + (R + M)^3)$. Applying the SVD projection gives

$$\mathbf{W}_t = \mathbf{U}_t\left[:, : R\right]\mathbf{\Lambda}_t\left[: R, :R\right] \tag{284}$$

$$\mathbf{\Upsilon}_t = \mathbf{\Upsilon}_{t|t-1} + \mathrm{diag}\left(\tilde{\mathbf{W}}_t\tilde{\mathbf{W}}_t^\intercal - \mathbf{W}_t\mathbf{W}_t^\intercal\right) \tag{285}$$

$$(\mathbf{U}_t, \mathbf{\Lambda}_t, \_) = \mathrm{SVD}\left(\tilde{\mathbf{W}}_t\right) \tag{286}$$

which takes $O((R + M)^2P)$ for the SVD. Therefore the BONG-MC-EF-DLR update is defined by Eqs. (281) and (283) to (286) and takes $O((R + M)^2P + (R + M)^3)$.

The BONG-LIN-EF-DLR update comes from replacing $\frac{1}{\sqrt{M}}\hat{\mathbf{G}}_t^{(1:M)}$ with $\boldsymbol{g}_t^{\text{LIN}}$ in Eq. (281) and replacing $\mathbf{I}_{R+M}$ with $\mathbf{I}_{R+1}$ in Eq. (283). This update takes $O((R + 1)^2P + (R + 1)^3)$.

### E.5.2 BONG-LIN-HESS DLR (LO-FI)

Applying Proposition 4.2 to Eqs. (281) and (283) gives the BONG-LIN-HESS-DLR update:

$$\boldsymbol{\mu}_t = \boldsymbol{\mu}_{t|t-1} + \left(\mathbf{\Upsilon}_{t|t-1}^{-1} - \mathbf{\Upsilon}_{t|t-1}^{-1}\tilde{\mathbf{W}}_t\left(\mathbf{I}_{R+C} + \tilde{\mathbf{W}}_t^\intercal\mathbf{\Upsilon}_{t|t-1}^{-1}\tilde{\mathbf{W}}_t\right)^{-1}\tilde{\mathbf{W}}_t^\intercal\mathbf{\Upsilon}_{t|t-1}^{-1}\right)$$

$$\times \mathbf{H}_t^\intercal\mathbf{R}_t^{-1}\left(\boldsymbol{y}_t - \hat{\boldsymbol{y}}_t\right) \tag{287}$$

$$\mathbf{W}_t = \mathbf{U}_t\left[:, :R\right]\mathbf{\Lambda}_t\left[:R, :R\right] \tag{288}$$

$$\mathbf{\Upsilon}_t = \mathbf{\Upsilon}_{t|t-1} + \mathrm{diag}\left(\tilde{\mathbf{W}}_t\tilde{\mathbf{W}}_t^\intercal - \mathbf{W}_t\mathbf{W}_t^\intercal\right) \tag{289}$$

$$(\mathbf{U}_t, \mathbf{\Lambda}_t, \_) = \mathrm{SVD}\left(\tilde{\mathbf{W}}_t\right) \tag{290}$$

$$\tilde{\mathbf{W}}_t = \left[\mathbf{W}_{t|t-1}, \mathbf{H}_t^\intercal\mathbf{A}_t^\intercal\right] \tag{291}$$

$$\mathbf{A}_t = \mathrm{chol}\left(\mathbf{R}_t^{-1}\right) \tag{292}$$

This is equivalent to LO-FI [Chang et al., 2023] and takes $O((R + C)^2P + (R + C)^3)$.

### E.5.3 BLR DLR (SLANG)

Substituting DLR forms for $q_{\psi_{t|t-1}}$ and $q_{\psi_{t,i-1}}$ into the FC precision update from Eq. (136) and using the MC-EF approximation yields the posterior precision

$$\tilde{\boldsymbol{\Sigma}}_{t,i}^{-1} = (1-\alpha)\left(\boldsymbol{\Upsilon}_{t,i-1} + \mathbf{W}_{t,i-1}\mathbf{W}_{t,i-1}^{\mathsf{T}}\right) + \alpha\left(\boldsymbol{\Upsilon}_{t|t-1} + \mathbf{W}_{t|t-1}\mathbf{W}_{t|t-1}^{\mathsf{T}}\right) - \alpha\mathbf{G}_{t,i}^{\text{MC-EF}} \quad (293)$$

$$= \tilde{\boldsymbol{\Upsilon}}_{t,i} + \tilde{\mathbf{W}}_{t,i}\tilde{\mathbf{W}}_{t,i}^{\mathsf{T}} \quad (294)$$

$$\tilde{\boldsymbol{\Upsilon}}_{t,i} = (1-\alpha)\,\boldsymbol{\Upsilon}_{t,i-1} + \alpha\boldsymbol{\Upsilon}_{t|t-1} \quad (295)$$

$$\tilde{\mathbf{W}}_{t,i} = \left[\sqrt{1-\alpha}\mathbf{W}_{t,i-1}, \sqrt{\alpha}\mathbf{W}_{t|t-1}, \sqrt{\frac{\alpha}{M}}\hat{\mathbf{G}}_{t,i}^{(1:M)}\right] \quad (296)$$

Note $\tilde{\mathbf{W}}_t \in \mathbb{R}^{P \times (2R+M)}$. Using this posterior precision in the mean update from Eq. (135) yields

$$\boldsymbol{\mu}_{t,i} = \boldsymbol{\mu}_{t,i-1} + \alpha\left(\tilde{\boldsymbol{\Upsilon}}_{t,i} + \tilde{\mathbf{W}}_{t,i}\tilde{\mathbf{W}}_{t,i}^{\mathsf{T}}\right)^{-1}$$
$$\times \left(\left(\boldsymbol{\Upsilon}_{t|t-1} + \mathbf{W}_{t|t-1}\mathbf{W}_{t|t-1}^{\mathsf{T}}\right)\left(\boldsymbol{\mu}_{t|t-1} - \boldsymbol{\mu}_{t,i-1}\right) + \boldsymbol{g}_{t,i}\right) \quad (297)$$

$$= \boldsymbol{\mu}_{t,i-1} + \alpha\left(\tilde{\boldsymbol{\Upsilon}}_{t,i}^{-1} - \tilde{\boldsymbol{\Upsilon}}_{t,i}^{-1}\tilde{\mathbf{W}}_{t,i}\left(\mathbf{I}_{2R+M} + \tilde{\mathbf{W}}_{t,i}^{\mathsf{T}}\tilde{\boldsymbol{\Upsilon}}_{t,i}^{-1}\tilde{\mathbf{W}}_{t,i}\right)^{-1}\tilde{\mathbf{W}}_{t,i}^{\mathsf{T}}\tilde{\boldsymbol{\Upsilon}}_{t,i}^{-1}\right)$$
$$\times \left(\left(\boldsymbol{\Upsilon}_{t|t-1} + \mathbf{W}_{t|t-1}\mathbf{W}_{t|t-1}^{\mathsf{T}}\right)\left(\boldsymbol{\mu}_{t|t-1} - \boldsymbol{\mu}_{t,i-1}\right) + \boldsymbol{g}_{t,i}\right) \quad (298)$$

where the second line comes from the Woodbury identity and can be computed in $O((2R+M)^2 P + (2R+M)^3)$. Applying the SVD projection gives

$$\mathbf{W}_{t,i} = \mathbf{U}_{t,i}\left[:,:R\right]\boldsymbol{\Lambda}_{t,i}\left[:R,:R\right] \quad (299)$$

$$\boldsymbol{\Upsilon}_{t,i} = \tilde{\boldsymbol{\Upsilon}}_{t,i} + \text{diag}\left(\tilde{\mathbf{W}}_{t,i}\tilde{\mathbf{W}}_{t,i}^{\mathsf{T}} - \mathbf{W}_{t,i}\mathbf{W}_{t,i}^{\mathsf{T}}\right) \quad (300)$$

$$(\mathbf{U}_{t,i}, \boldsymbol{\Lambda}_{t,i}, \_) = \text{SVD}\left(\tilde{\mathbf{W}}_{t,i}\right) \quad (301)$$

which takes $O((2R+M)^2 P)$ for the SVD. Therefore the BLR-MC-EF-DLR update is defined by Eqs. (295), (296) and (298) to (301) and takes $O((2R+M)^2 P + (2R+M)^3)$ per iteration. Notice $\tilde{\mathbf{W}}_t$ has larger rank for BLR than for BONG ($2R+M$ vs. $R+M$) because of the extra $\sqrt{\alpha}\mathbf{W}_{t|t-1}$ term that originates in the KL part of the VI loss. This difference will slow BLR-MC-EF-DLR relative to BONG-MC-EF-DLR especially when $R$ is not small relative to $M$.

This method closely resembles SLANG Mishkin et al. [2018] in the batch setting where we replace $q_{\psi_{t|t-1}}$ with a spherical prior $\mathcal{N}(\mathbf{0}, \lambda^{-1}\mathbf{I}_P)$ (see Appendix E.6).

We can also define a BLR-LIN-EF-DLR method by replacing $\sqrt{\frac{\alpha}{M}}\hat{\mathbf{G}}_t^{(1:M)}$ with $\sqrt{\alpha}g_t^{\text{LIN}}$ in Eq. (296) and $\mathbf{I}_{2R+M}$ with $\mathbf{I}_{2R+1}$ in Eq. (298). This update takes $O((2R+1)^2 P + (2R+1)^3)$ per iteration.

### E.5.4 BLR-LIN-HESS DLR

Applying Proposition 4.2 to Eqs. (296) and (298) gives the BLR-LIN-HESS-DLR update:

$$\boldsymbol{\mu}_{t,i} = \boldsymbol{\mu}_{t,i-1} + \alpha\left(\tilde{\boldsymbol{\Upsilon}}_{t,i}^{-1} - \tilde{\boldsymbol{\Upsilon}}_{t,i}^{-1}\tilde{\mathbf{W}}_{t,i}\left(\mathbf{I}_{2R+C} + \tilde{\mathbf{W}}_{t,i}^{\mathsf{T}}\tilde{\boldsymbol{\Upsilon}}_{t,i}^{-1}\tilde{\mathbf{W}}_{t,i}\right)^{-1}\tilde{\mathbf{W}}_{t,i}^{\mathsf{T}}\tilde{\boldsymbol{\Upsilon}}_{t,i}^{-1}\right)$$
$$\times \left(\left(\boldsymbol{\Upsilon}_{t|t-1} + \mathbf{W}_{t|t-1}\mathbf{W}_{t|t-1}^{\mathsf{T}}\right)\left(\boldsymbol{\mu}_{t|t-1} - \boldsymbol{\mu}_{t,i-1}\right) + \mathbf{H}_{t,i}^{\mathsf{T}}\mathbf{R}_{t,i}^{-1}\left(\boldsymbol{y}_t - \hat{\boldsymbol{y}}_{t,i}\right)\right) \quad (302)$$

$$\mathbf{W}_{t,i} = \mathbf{U}_{t,i}\left[:,:R\right]\boldsymbol{\Lambda}_{t,i}\left[:R,:R\right] \quad (303)$$

$$\boldsymbol{\Upsilon}_{t,i} = \tilde{\boldsymbol{\Upsilon}}_{t,i} + \text{diag}\left(\tilde{\mathbf{W}}_{t,i}\tilde{\mathbf{W}}_{t,i}^{\mathsf{T}} - \mathbf{W}_{t,i}\mathbf{W}_{t,i}^{\mathsf{T}}\right) \quad (304)$$

$$(\mathbf{U}_{t,i}, \boldsymbol{\Lambda}_{t,i}, \_) = \text{SVD}\left(\tilde{\mathbf{W}}_{t,i}\right) \quad (305)$$

$$\tilde{\mathbf{W}}_{t,i} = \left[\sqrt{1-\alpha}\mathbf{W}_{t,i-1}, \sqrt{\alpha}\mathbf{W}_{t|t-1}, \sqrt{\alpha}\mathbf{H}_{t,i}^{\mathsf{T}}\mathbf{A}_{t,i}^{\mathsf{T}}\right] \quad (306)$$

$$\mathbf{A}_{t,i} = \text{chol}\left(\mathbf{R}_{t,i}^{-1}\right) \quad (307)$$

$$\tilde{\boldsymbol{\Upsilon}}_{t,i} = (1-\alpha)\,\boldsymbol{\Upsilon}_{t,i-1} + \alpha\boldsymbol{\Upsilon}_{t|t-1} \quad (308)$$

This update takes $O((2R + C)^2 P + (2R + C)^3)$ per iteration. As with the EF versions of BLR-DLR, $\tilde{\mathbf{W}}_t$ has larger rank for BLR-LIN-HESS-DLR than for BONG-LIN-HESS-DLR ($2R + C$ vs. $R + C$). This difference will slow BLR-LIN-HESS-DLR relative to BONG-LIN-HESS-DLR especially when $R$ is not small relative to $C$.

### E.5.5   BOG DLR

Substituting Eqs. (269) to (271) into Eq. (31) gives the BOG-DLR update

$$\boldsymbol{\mu}_t = \boldsymbol{\mu}_{t|t-1} + \alpha \boldsymbol{g}_t \tag{309}$$

$$\boldsymbol{\Upsilon}_t = \boldsymbol{\Upsilon}_{t|t-1} - \frac{\alpha}{2}\mathrm{diag}\left(\left(\boldsymbol{\Upsilon}_{t|t-1} + \mathbf{W}_{t|t-1}\mathbf{W}_{t|t-1}^\intercal\right)^{-1}\mathbf{G}_t\left(\boldsymbol{\Upsilon}_{t|t-1} + \mathbf{W}_{t|t-1}\mathbf{W}_{t|t-1}^\intercal\right)^{-1}\right) \tag{310}$$

$$\mathbf{W}_t = \mathbf{W}_{t|t-1} - \alpha\left(\boldsymbol{\Upsilon}_{t|t-1} + \mathbf{W}_{t|t-1}\mathbf{W}_{t|t-1}^\intercal\right)^{-1}\mathbf{G}_t\left(\boldsymbol{\Upsilon}_{t|t-1} + \mathbf{W}_{t|t-1}\mathbf{W}_{t|t-1}^\intercal\right)^{-1}\mathbf{W}_{t|t-1} \tag{311}$$

Using the EF approximation and Woodbury, the BOG-MC-EF-DLR update can be rewritten as

$$\boldsymbol{\Upsilon}_t = \boldsymbol{\Upsilon}_{t|t-1} + \frac{\alpha}{2M}\mathrm{diag}\left(\mathbf{B}_t\mathbf{B}_t^\intercal\right) \tag{312}$$

$$\mathbf{W}_t = \mathbf{W}_{t|t-1} + \frac{\alpha}{M}\mathbf{B}_t\mathbf{B}_t^\intercal\mathbf{W}_{t|t-1} \tag{313}$$

$$\mathbf{B}_t = \left(\boldsymbol{\Upsilon}_{t|t-1}^{-1} - \boldsymbol{\Upsilon}_{t|t-1}^{-1}\mathbf{W}_{t|t-1}\left(\mathbf{I}_R + \mathbf{W}_{t|t-1}^\intercal\boldsymbol{\Upsilon}_{t|t-1}^{-1}\mathbf{W}_{t|t-1}\right)^{-1}\mathbf{W}_{t|t-1}^\intercal\boldsymbol{\Upsilon}_{t|t-1}^{-1}\right)\hat{\mathbf{G}}_t^{(1:M)} \tag{314}$$

which takes $O(RMP + MR^2 + R^3)$.

The BOG-LIN-EF-DLR update comes from replacing $\hat{\mathbf{G}}_t^{(1:M)}$ with $\boldsymbol{g}_t^{\mathrm{LIN}}$ in Eq. (314) and dropping the $M^{-1}$ factors in Eqs. (312) and (313). This update takes $O(RP + R^3)$.

### E.5.6   BOG-LIN-HESS DLR

Applying Proposition 4.2 to Eqs. (309) and (312) to (314) gives the BOG-LIN-HESS-DLR update

$$\boldsymbol{\mu}_t = \boldsymbol{\mu}_{t|t-1} + \alpha\mathbf{H}_t^\intercal\mathbf{R}_t^{-1}(\boldsymbol{y}_t - \hat{\boldsymbol{y}}_t) \tag{315}$$

$$\boldsymbol{\Upsilon}_t = \boldsymbol{\Upsilon}_{t|t-1} + \frac{\alpha}{2}\mathrm{diag}\left(\mathbf{B}_t\mathbf{B}_t^\intercal\right) \tag{316}$$

$$\mathbf{W}_t = \mathbf{W}_{t|t-1} + \alpha\mathbf{B}_t\mathbf{B}_t^\intercal\mathbf{W}_{t|t-1} \tag{317}$$

$$\mathbf{B}_t = \left(\boldsymbol{\Upsilon}_{t|t-1}^{-1} - \boldsymbol{\Upsilon}_{t|t-1}^{-1}\mathbf{W}_{t|t-1}\left(\mathbf{I}_R + \mathbf{W}_{t|t-1}^\intercal\boldsymbol{\Upsilon}_{t|t-1}^{-1}\mathbf{W}_{t|t-1}\right)^{-1}\mathbf{W}_{t|t-1}^\intercal\boldsymbol{\Upsilon}_{t|t-1}^{-1}\right)\mathbf{H}_t^\intercal\mathbf{A}_t^\intercal \tag{318}$$

This update takes $O(C(C + R)P + CR^2 + R^3)$.

### E.5.7   BBB DLR

Substituting Eqs. (265) and (269) to (273) into Eq. (34) gives the BBB-DLR update

$$\boldsymbol{\mu}_{t,i} = \boldsymbol{\mu}_{t,i-1} + \alpha\left(\boldsymbol{\Upsilon}_{t|t-1} + \mathbf{W}_{t|t-1}\mathbf{W}_{t|t-1}^\intercal\right)\left(\boldsymbol{\mu}_{t|t-1} - \boldsymbol{\mu}_{t,i-1}\right) + \alpha\boldsymbol{g}_t \tag{319}$$

$$\begin{aligned}\boldsymbol{\Upsilon}_{t,i} &= \boldsymbol{\Upsilon}_{t,i-1} \\ &\quad + \frac{\alpha}{2}\mathrm{diag}\left(\boldsymbol{\Sigma}_{t,i-1}\left(\boldsymbol{\Upsilon}_{t|t-1} - \boldsymbol{\Upsilon}_{t,i-1} + \mathbf{W}_{t|t-1}\mathbf{W}_{t|t-1}^\intercal - \mathbf{W}_{t,i-1}\mathbf{W}_{t,i-1}^\intercal - \mathbf{G}_{t,i}\right)\boldsymbol{\Sigma}_{t,i-1}\right)\end{aligned} \tag{320}$$

$$\begin{aligned}\mathbf{W}_{t,i} &= \mathbf{W}_{t,i-1} \\ &\quad + \alpha\boldsymbol{\Sigma}_{t,i-1}\left(\boldsymbol{\Upsilon}_{t|t-1} - \boldsymbol{\Upsilon}_{t,i-1} + \mathbf{W}_{t|t-1}\mathbf{W}_{t|t-1}^\intercal - \mathbf{W}_{t,i-1}\mathbf{W}_{t,i-1}^\intercal - \mathbf{G}_{t,i}\right)\boldsymbol{\Sigma}_{t,i-1}\mathbf{W}_{t,i-1}\end{aligned} \tag{321}$$

The previous covariance can be written using Woodbury as

$$\Sigma_{t,i-1} = \Upsilon_{t,i-1}^{-1} - \Upsilon_{t,i-1}^{-1}\mathbf{W}_{t,i-1}\left(\mathbf{I}_R + \mathbf{W}_{t,i-1}^{\mathsf{T}}\Upsilon_{t,i-1}^{-1}\mathbf{W}_{t,i-1}\right)^{-1}\mathbf{W}_{t,i-1}^{\mathsf{T}}\Upsilon_{t,i-1}^{-1} \qquad (322)$$

The BBB-MC-EF-DLR update can be computed efficiently by expanding terms in Eqs. (320) to (322). For example the terms involving $\mathbf{G}_{t,i}$ can be calculated as

$$\mathrm{diag}\left(-\Sigma_{t,i-1}\mathbf{G}_{t,i}^{\text{MC-EF}}\Sigma_{t,i-1}\right) = \frac{1}{M}\mathrm{diag}\left(\begin{array}{c} \Upsilon_{t,i-1}^{-1}\hat{\mathbf{G}}_{t,i}^{(1:M)}\hat{\mathbf{G}}_{t,i}^{(1:M)^{\mathsf{T}}}\Upsilon_{t,i-1}^{-1} \\ -\Upsilon_{t,i-1}^{-1}\hat{\mathbf{G}}_{t,i}^{(1:M)}\mathbf{B}_{t,i}^{\mathsf{T}} \\ -\mathbf{B}_{t,i}\hat{\mathbf{G}}_{t,i}^{(1:M)^{\mathsf{T}}}\Upsilon_{t,i-1}^{-1} + \mathbf{B}_{t,i}\mathbf{B}_{t,i}^{\mathsf{T}} \end{array}\right) \qquad (323)$$

$$-\Sigma_{t,i-1}\mathbf{G}_{t,i}^{\text{MC-EF}}\Sigma_{t,i-1}\mathbf{W}_{t,i-1} = \frac{1}{M}\Upsilon_{t,i-1}^{-1}\hat{\mathbf{G}}_{t,i}^{(1:M)}\hat{\mathbf{G}}_{t,i}^{(1:M)^{\mathsf{T}}}\Upsilon_{t,i-1}^{-1}\mathbf{W}_{t,i-1}$$

$$-\frac{1}{M}\Upsilon_{t,i-1}^{-1}\hat{\mathbf{G}}_{t,i}^{(1:M)}\mathbf{B}_{t,i}^{\mathsf{T}}\mathbf{W}_{t,i-1}$$

$$-\frac{1}{M}\mathbf{B}_{t,i}\hat{\mathbf{G}}_{t,i}^{(1:M)^{\mathsf{T}}}\Upsilon_{t,i-1}^{-1}\mathbf{W}_{t,i-1} + \frac{1}{M}\mathbf{B}_{t,i}\mathbf{B}_{t,i}^{\mathsf{T}}\mathbf{W}_{t,i-1} \qquad (324)$$

$$\mathbf{B}_{t,i} = \Upsilon_{t,i-1}^{-1}\mathbf{W}_{t,i-1}\left(\mathbf{I}_R + \mathbf{W}_{t,i-1}^{\mathsf{T}}\Upsilon_{t,i-1}^{-1}\mathbf{W}_{t,i-1}\right)^{-1}$$

$$\times \mathbf{W}_{t,i-1}^{\mathsf{T}}\Upsilon_{t,i-1}^{-1}\hat{\mathbf{G}}_{t,i}^{(1:M)} \qquad (325)$$

Using this strategy the update takes $O((R+M)RP + MR^2 + R^3)$.

The BBB-LIN-EF-DLR update comes from replacing $\hat{\mathbf{G}}_t^{(1:M)}$ with $\mathbf{g}_t^{\text{LIN}}$ and dropping the $M^{-1}$ factors in Eqs. (323) to (325). This update takes $O(R^2P + R^3)$.

### E.5.8  BBB-LIN-HESS DLR

Applying Proposition 4.2 to Eqs. (319) to (321) gives the BBB-LIN-HESS-DLR update

$$\boldsymbol{\mu}_{t,i} = \boldsymbol{\mu}_{t,i-1} + \alpha\left(\Upsilon_{t|t-1} + \mathbf{W}_{t|t-1}\mathbf{W}_{t|t-1}^{\mathsf{T}}\right)\left(\boldsymbol{\mu}_{t|t-1} - \boldsymbol{\mu}_{t,i-1}\right) + \alpha\mathbf{H}_{t,i}^{\mathsf{T}}\mathbf{R}_{t,i}^{-1}(\boldsymbol{y}_t - \hat{\boldsymbol{y}}_{t,i}) \qquad (326)$$

$$\Upsilon_{t,i} = \Upsilon_{t,i-1} + \frac{\alpha}{2}\mathrm{diag}\left(\Sigma_{t,i-1}\left(\begin{array}{c}\Upsilon_{t|t-1} - \Upsilon_{t,i-1} + \mathbf{W}_{t|t-1}\mathbf{W}_{t|t-1}^{\mathsf{T}} \\ -\mathbf{W}_{t,i-1}\mathbf{W}_{t,i-1}^{\mathsf{T}} + \mathbf{H}_{t,i}^{\mathsf{T}}\mathbf{R}_{t,i}^{-1}\mathbf{H}_{t,i}\end{array}\right)\Sigma_{t,i-1}\right) \qquad (327)$$

$$\mathbf{W}_{t,i} = \mathbf{W}_{t,i-1} + \alpha\Sigma_{t,i-1}\left(\begin{array}{c}\Upsilon_{t|t-1} - \Upsilon_{t,i-1} + \mathbf{W}_{t|t-1}\mathbf{W}_{t|t-1}^{\mathsf{T}} \\ -\mathbf{W}_{t,i-1}\mathbf{W}_{t,i-1}^{\mathsf{T}} + \mathbf{H}_{t,i}^{\mathsf{T}}\mathbf{R}_{t,i}^{-1}\mathbf{H}_{t,i}\end{array}\right)\Sigma_{t,i-1}\mathbf{W}_{t,i-1} \qquad (328)$$

This can be computed in $O((C+R)^2P + CR^2 + R^3)$ using Eq. (322) and following a computational approach similar to Eqs. (323) to (325).

### E.6  Batch BLR

It is interesting to translate the BLR updates derived here back to the batch setting where BLR was developed [Khan and Rue, 2023], by replacing $\mathcal{N}\left(\boldsymbol{\mu}_{t|t-1}, \Sigma_{t|t-1}\right)$ with a centered spherical prior $\mathcal{N}\left(\mathbf{0}, \lambda^{-1}\mathbf{I}_P\right)$.

The batch BLR-FC update becomes

$$\boldsymbol{\mu}_i = \boldsymbol{\mu}_{i-1} + \alpha\Sigma_i\left(\boldsymbol{g}_i - \lambda\boldsymbol{\mu}_{i-1}\right) \qquad (329)$$

$$\Sigma_i^{-1} = (1-\alpha)\Sigma_{i-1}^{-1} + \alpha\left(\lambda\mathbf{I}_P - \mathbf{G}_i\right) \qquad (330)$$

The batch BLR-LIN-HESS-FC update becomes

$$\boldsymbol{\mu}_i = \boldsymbol{\mu}_{i-1} + \alpha\Sigma_i\left(\mathbf{H}_i^{\mathsf{T}}\mathbf{R}_i^{-1}(\boldsymbol{y}_t - \hat{\boldsymbol{y}}_i) - \lambda\boldsymbol{\mu}_{i-1}\right) \qquad (331)$$

$$\Sigma_i^{-1} = (1-\alpha)\Sigma_{i-1}^{-1} + \alpha\left(\lambda\mathbf{I}_P + \mathbf{H}_i^{\mathsf{T}}\mathbf{R}_i^{-1}\mathbf{H}_i\right) \qquad (332)$$

The batch BLR-FC_MOM update becomes

$$\boldsymbol{\mu}_i = \boldsymbol{\mu}_{i-1} + \alpha\Sigma_{i-1}\left(\boldsymbol{g}_i - \lambda\boldsymbol{\mu}_{i-1}\right) \qquad (333)$$

$$\Sigma_i = (1+\alpha)\Sigma_{i-1} + \alpha\Sigma_{i-1}\left(\mathbf{G}_i - \lambda\mathbf{I}_P\right)\Sigma_{i-1} \qquad (334)$$

The batch BLR-LIN-HESS-FC_MOM update becomes

$$\boldsymbol{\mu}_i = \boldsymbol{\mu}_{i-1} + \alpha \boldsymbol{\Sigma}_{i-1} \left( \mathbf{H}_i^\mathsf{T} \mathbf{R}_i^{-1} (\boldsymbol{y}_t - \hat{\boldsymbol{y}}_i) - \lambda \boldsymbol{\mu}_{i-1} \right) \tag{335}$$

$$\boldsymbol{\Sigma}_i = (1 + \alpha) \boldsymbol{\Sigma}_{i-1} - \alpha \boldsymbol{\Sigma}_{i-1} \left( \lambda \mathbf{I}_P + \mathbf{H}_i^\mathsf{T} \mathbf{R}_i^{-1} \mathbf{H}_i \right) \boldsymbol{\Sigma}_{i-1} \tag{336}$$

The batch BLR-DIAG update becomes

$$\boldsymbol{\mu}_i = \boldsymbol{\mu}_{i-1} + \alpha \boldsymbol{\sigma}_i^2 \left( \boldsymbol{g}_i - \lambda \boldsymbol{\mu}_{i-1} \right) \tag{337}$$

$$\boldsymbol{\sigma}_i^{-2} = (1 - \alpha) \boldsymbol{\sigma}_{i-1}^{-2} + \alpha \operatorname{diag} \left( \lambda \mathbf{I}_P - \mathbf{G}_i \right) \tag{338}$$

The MC-HESS version of this update is equivalent to VON [Khan et al., 2018b] if we use MC approximation with $M = 1$. The MC-EF version with $M = 1$ is equivalent to VOGN [Khan et al., 2018b].

The batch BLR-LIN-HESS-DIAG update becomes

$$\boldsymbol{\mu}_i = \boldsymbol{\mu}_{i-1} + \alpha \boldsymbol{\sigma}_i^2 \left( \mathbf{H}_i^\mathsf{T} \mathbf{R}_i^{-1} (\boldsymbol{y}_t - \hat{\boldsymbol{y}}_i) - \lambda \boldsymbol{\mu}_{i-1} \right) \tag{339}$$

$$\boldsymbol{\sigma}_i^{-2} = (1 - \alpha) \boldsymbol{\sigma}_{i-1}^{-2} + \alpha \operatorname{diag} \left( \lambda \mathbf{I}_P + \mathbf{H}_i^\mathsf{T} \mathbf{R}_i^{-1} \mathbf{H}_i \right) \tag{340}$$

The batch BLR-DIAG_MOM update becomes

$$\boldsymbol{\mu}_i = \boldsymbol{\mu}_{i-1} + \alpha \boldsymbol{\sigma}_{i-1}^2 \left( \boldsymbol{g}_i - \lambda \boldsymbol{\mu}_{i-1} \right) \tag{341}$$

$$\boldsymbol{\sigma}_i^2 = (1 + \alpha) \boldsymbol{\sigma}_{i-1}^2 + \alpha \boldsymbol{\sigma}_{i-1}^4 \operatorname{diag} \left( \mathbf{G}_i - \lambda \mathbf{I}_P \right) \tag{342}$$

The batch BLR-LIN-HESS-DIAG_MOM update becomes

$$\boldsymbol{\mu}_i = \boldsymbol{\mu}_{i-1} + \alpha \boldsymbol{\sigma}_{i-1}^2 \left( \mathbf{H}_i^\mathsf{T} \mathbf{R}_i^{-1} (\boldsymbol{y}_t - \hat{\boldsymbol{y}}_i) - \lambda \boldsymbol{\mu}_{i-1} \right) \tag{343}$$

$$\boldsymbol{\sigma}_i^2 = (1 + \alpha) \boldsymbol{\sigma}_{i-1}^2 - \alpha \boldsymbol{\sigma}_{i-1}^4 \operatorname{diag} \left( \lambda \mathbf{I}_P + \mathbf{H}_i^\mathsf{T} \mathbf{R}_i^{-1} \mathbf{H}_i \right) \tag{344}$$

The batch BLR-DLR update becomes

$$\boldsymbol{\mu}_i = \boldsymbol{\mu}_{i-1} + \alpha \left( \tilde{\boldsymbol{\Upsilon}}_i^{-1} - \tilde{\boldsymbol{\Upsilon}}_i^{-1} \tilde{\mathbf{W}}_i \left( \mathbf{I}_{R+M} + \tilde{\mathbf{W}}_i^\mathsf{T} \tilde{\boldsymbol{\Upsilon}}_i^{-1} \tilde{\mathbf{W}}_i \right)^{-1} \tilde{\mathbf{W}}_i^\mathsf{T} \tilde{\boldsymbol{\Upsilon}}_i^{-1} \right)$$
$$\times (\boldsymbol{g}_t - \lambda \boldsymbol{\mu}_{i-1}) \tag{345}$$

$$\mathbf{W}_i = \mathbf{U}_i \left[ :, : R \right] \boldsymbol{\Lambda}_i \left[ : R, : R \right] \tag{346}$$

$$\boldsymbol{\Upsilon}_i = \tilde{\boldsymbol{\Upsilon}}_i + \operatorname{diag} \left( \tilde{\mathbf{W}}_i \tilde{\mathbf{W}}_i^\mathsf{T} - \mathbf{W}_i \mathbf{W}_i^\mathsf{T} \right) \tag{347}$$

$$(\mathbf{U}_i, \boldsymbol{\Lambda}_i, \_) = \operatorname{SVD} \left( \tilde{\mathbf{W}}_i \right) \tag{348}$$

$$\tilde{\mathbf{W}}_i = \left[ \sqrt{1 - \alpha} \mathbf{W}_{i-1}, \sqrt{\frac{\alpha}{M}} \hat{\mathbf{G}}_i^{(1:M)} \right] \tag{349}$$

$$\mathbf{A}_i = \operatorname{chol} \left( \mathbf{R}_i^{-1} \right) \tag{350}$$

$$\tilde{\boldsymbol{\Upsilon}}_i = (1 - \alpha) \boldsymbol{\Upsilon}_{i-1} + \alpha \lambda \mathbf{I}_P \tag{351}$$

This is equivalent to SLANG except for the following differences. SLANG processes a minibatch of $M$ examples at each iteration, using a single sample $\hat{\boldsymbol{\theta}} \sim q_{\boldsymbol{\psi}_{i-1}}$ for each minibatch. It uses a different SVD routine which is slightly faster but stochastic, taken from [Halko et al., 2011]. Most significantly, SLANG applies the SVD before the mean update, meaning $\boldsymbol{\mu}_i$ is calculated using the rank-$R$ $\mathbf{W}_i$ and $\boldsymbol{\Upsilon}_i$ instead of the rank-$(R + M)$ $\tilde{\mathbf{W}}_i$ and $\hat{\boldsymbol{\Upsilon}}_i$, thus ignoring the non-diagonal information in the $M$ discarded singular vectors from $\tilde{\mathbf{W}}_i$.

The batch BLR-LIN-HESS-DLR update becomes

$$\boldsymbol{\mu}_i = \boldsymbol{\mu}_{i-1} + \alpha \left( \tilde{\boldsymbol{\Upsilon}}_i^{-1} - \tilde{\boldsymbol{\Upsilon}}_i^{-1} \tilde{\mathbf{W}}_i \left( \mathbf{I}_{R+C} + \tilde{\mathbf{W}}_i^\intercal \tilde{\boldsymbol{\Upsilon}}_i^{-1} \tilde{\mathbf{W}}_i \right)^{-1} \tilde{\mathbf{W}}_i^\intercal \tilde{\boldsymbol{\Upsilon}}_i^{-1} \right)$$

$$\times \left( \mathbf{H}_i^\intercal \mathbf{R}_i^{-1} (\boldsymbol{y}_t - \hat{\boldsymbol{y}}_i) - \lambda \boldsymbol{\mu}_{i-1} \right) \tag{352}$$

$$\mathbf{W}_i = \mathbf{U}_i \left[ :, : R \right] \boldsymbol{\Lambda}_i \left[ : R, : R \right] \tag{353}$$

$$\boldsymbol{\Upsilon}_i = \tilde{\boldsymbol{\Upsilon}}_i + \mathrm{diag} \left( \tilde{\mathbf{W}}_i \tilde{\mathbf{W}}_i^\intercal - \mathbf{W}_i \mathbf{W}_i^\intercal \right) \tag{354}$$

$$(\mathbf{U}_i, \boldsymbol{\Lambda}_i, \_) = \mathrm{SVD} \left( \tilde{\mathbf{W}}_i \right) \tag{355}$$

$$\tilde{\mathbf{W}}_i = \left[ \sqrt{1-\alpha} \mathbf{W}_{i-1}, \sqrt{\alpha} \mathbf{H}_i^\intercal \mathbf{A}_i^\intercal \right] \tag{356}$$

$$\mathbf{A}_i = \mathrm{chol} \left( \mathbf{R}_i^{-1} \right) \tag{357}$$

$$\tilde{\boldsymbol{\Upsilon}}_i = (1-\alpha) \boldsymbol{\Upsilon}_{i-1} + \alpha \lambda \mathbf{I}_P \tag{358}$$

This algorithm could be called SLANG-LIN-HESS and would be deterministic and faster than SLANG since it does not need MC sampling. We can also define SLANG-LIN-EF which would be even faster, by replacing $\sqrt{\frac{\alpha}{M}} \hat{\mathbf{G}}_i^{(1:M)}$ with $\sqrt{\alpha} \boldsymbol{g}_i^{\mathrm{LIN}}$ in Eq. (349) and $\mathbf{I}_{R+M}$ with $\mathbf{I}_R$ in Eq. (345).

