# OpenReview forum: "Bayesian Online Natural Gradient (BONG)"
_NeurIPS.cc/2024/Conference — NeurIPS 2024 poster_

### Official Review · Reviewer_zVLu · 2024-07-11

**Soundness:** 3
**Presentation:** 3
**Contribution:** 3
**Rating:** 7
**Confidence:** 3

**Summary:**

The paper introduces a framework for variational online learning, in which a method is defined by three ingredients: (1) implicit or explicit regularisation (2) type of approximation of the expected gradient and hessian (3) geometry in which the gradient is taken (ie employing natural gradients or standard gradients). In implicit regularisation, one learning step is taken for each datapoint, with no regularisation; explicit method is the more standard approach of optimising with multiple steps and regularisation to the posterior at the previous step. A case study on different combinations of the ingredients is conducted; some of the methods appeared in previous papers. The study finds the best performing method is the one with implicit regularisation, linearisation of the hessian, and natural gradient.

**Strengths:**

The paper is a valuable contribution to the field, offering a framework for disjointed methods. It reads like a review paper in variational online learning, with a novel contribution based on the review. The list of methods of approximation is thorough and the comparison exhaustive.

**Weaknesses:**

Given that the main strength of the paper is the unifying recipe, and the comparison of using different ingredients, a more exhaustive case study would strengthen the work (and the less essential comments, like the equivalence to the full Bayesian update in the conjugate case, in my opinion can be placed in the appendix and referenced). Nevertheless, I find the contribution of the paper warrants it being accepted.

**Questions:**

* Given the well documented efficiency of natural gradient steps, is there a danger of the method with no regularisation being more prone to "forgetting"?

---

> ### Author Rebuttal · Authors · 2024-08-05
>
> Thank you for the positive feedback.
>
> We appreciate the suggestion to report a more exhaustive case study and give less space to the motivation from the conjugate case. The reason we prefer the present structure is that it puts more weight on the new algorithms. We’re glad you find the organizing framework and review-like aspects of the paper to be a good contribution (we agree), but we also want to highlight the BONG method and its empirical success.
>
> _Can NGD promote forgetting because of its efficiency?_
>
> This is an interesting question. We think the right way to think about it is by comparison to the exact Bayesian update, which optimally maintains past information (i.e., forgets just the right amount). Our methods are approximations of the exact update. So even if the NGD step went all the way to the minimizer of the variational loss (as it does in the conjugate case) it should not overlearn or excessively forget past observations.

---

### Official Review · Reviewer_uSab · 2024-07-12

**Soundness:** 4
**Presentation:** 3
**Contribution:** 3
**Rating:** 6
**Confidence:** 3

**Summary:**

The authors explore sequential Bayesian inference using variational inference (VI). They propose to remove the regularizing KL term, performing a single natural gradient step using the expected log-likelihood only, and provide relevant ways to approximate the hard-to-compute expectations. They support their approach with detailed experiments.

**Strengths:**

The paper is sound, and the detailed experiments provide strong support for the method proposed in the paper. I also appreciated the nice motivation involving conjugacy, as well as the algorithmic considerations.

**Weaknesses:**

My main concern is about the claimed novelty of the approach ("removing the KL" and "implicit regularization towards the prior"). The idea of using a one-step natural gradient descent on the expected log-loss directly (without the KL term) has already been proposed in the literature and analyzed; for example, [Lyu & Tsang, 2021] studied this algorithm in a model-free setting where a general function f is considered instead of the log-loss. This one-step natural gradient update is in fact simply the solution of Equation (2) of the present paper, but with a linearized expected log-loss. This dual interpretation resonates with the dual interpretation of standard Euclidean gradient descent (single update or minimization of a first-order development of the loss to be minimized).

Y. Lyu and I. W. Tsang: Black-box optimizer with stochastic implicit natural gradient. In Joint European Conference on Machine Learning and Knowledge Discovery in Databases, pages 217–232, 2021.

**Questions:**

NA

---

> ### Author Rebuttal · Authors · 2024-08-05
>
> Thank you for your feedback.
>
> We were not aware of the Lyu & Tsang paper. We will add discussion of it to Section 2 but also want to point out some important differences: Their goal is optimization, not inference or Bayesian updating. Thus the KL regularizer is not part of their objective whereas it is for us. Instead their regularizer is an auxiliary device to obtain an incremental algorithm. (Also note that iteration number in their optimization setting is not the same as time step in our filtering setting.) Then they use the following two well-known facts (e.g., Raskutti & Mukherjee 2015) to obtain an NGD update:
>
> A) Mirror descent (MD) can be derived by optimizing linearized loss plus Bregman divergence (we believe this is the dual interpretation you refer to). That is, given parameters $\\boldsymbol{w}$, loss $\\mathcal{L}$, and Bregman function $\\Phi$, the MD update is equivalent to $\\boldsymbol{w}\_{i+1} = \\arg\\min\_{\\boldsymbol{w}} \\langle\\boldsymbol{w}, \\nabla\\mathcal{L}(\\boldsymbol{w}\_i)\\rangle + D\_{\\Phi}(\\boldsymbol{w}\_i,\\boldsymbol{w})$.
>
> B) NGD on the mean parameter of an exponential family is equivalent to MD where the Bregman function is the log-partition.
>
> Nevertheless their approach is closely related to an alternative MD-based derivation of BONG that we plan to add (this has already been written for an arXiv version after our NeurIPS submission). Using facts A and B above we show BONG can be obtained by linearizing the expected NLL wrt the variational mean parameter ($\\rho$). This is the same observation you make about our Eq (2) and linearized expected log-loss. But to be clear, connecting all these ideas to the ELBO is new in our paper and leads to a novel approach to variational inference.
>
> We believe our paper has multiple contributions, including a unified framework, a novel update, and extensive experimental comparison of different methods. Even if the BONG update equation is similar to algorithms used in other domains (such as Lyu & Tsang’s) its derivation and application to variational inference are novel, and moreover we believe it is just one piece of our overall contribution. We therefore respectfully ask you to consider raising your score.
>
> Raskutti, G., Mukherjee, S.: The information geometry of mirror descent. IEEE Transactions on Information Theory 61(3), 1451–1457 (2015)

---

> ### Comment · Reviewer_uSab · 2024-08-08
>
> Thank you very much for your reply and the detailed explanations. However, I remain unconvinced about the novelty of "connecting all these ideas to the ELBO", as well as the claim that the BONG update's "derivation and application to variational inference are novel".
>
> While it is true that Lyu & Tsang's paper "goal is optimization, not inference or Bayesian updating", it was merely an example of a paper that provides a general analysis of this specific algorithm. The idea has been explored in various papers with different approaches and motivations, such as [van der Hoeven et al., 2018; Cherief-Abdellatif et al., 2019; Khan & Rue, 2023]. In particular, the authors of [Khan & Rue, 2023] discuss the online VI case at the end of Section 5 and mention the linearization of the expected loss in the ELBO, referring to [van der Hoeven et al., 2018], which delves into these questions in detail (though not through the lens of VI). From a different perspective, [Cherief-Abdellatif et al., 2019] establish a connection between VI and online learning by proposing to perform a single gradient descent step for the expected loss without the KL, which they show to be equivalent to minimizing the ELBO with a linearized expected loss.
>
> Therefore, while I do believe this is a nice piece of work, I am still not fully convinced of its novelty. And although I recognize that the paper offers more than just this point, I feel that it is the aspect most emphasized (particularly in the abstract).
>
>
> [van der Hoeven et al., 2018]: D. van der Hoeven, T. van Erven, and W. Kotłowski. The many faces of exponential weights in online learning. In Conference On Learning Theory, pages 2067–2092. PMLR, 2018.
>
> [Cherief-Abdellatif et al., 2019]: B.E. Cherief-Abdellatif, P. Alquier, and M.E. Khan. A generalization bound for online variational inference. In Asian Conference on Machine Learning, pages 662–677. PMLR, 2019.
>
> [Khan & Rue, 2023]: M.E. Khan and H. Rue. The Bayesian Learning Rule. In Journal of Machine Learning Research, 2023.

---

> > ### Author Response · Authors · 2024-08-13
> >
> > Thank you for the additional comments.
> >
> > To clarify our previous reply, linearizing the loss$\^*$ in the full ELBO is an alternative means of deriving our update which we explain in our new appendix (to be added in the revision). The papers you reference are all good examples of this approach and we will cite them there. Our primary proposal of dropping the KL term and doing one NGD step still appears to be new.
> >
> > The algorithms in Chérief-Abdellatif, Alquier & Khan (2019) all use linearization of the loss, and we don't see any suggestions there to perform a single gradient descent step for the expected loss without the KL. Their SVB algorithm (eq 7) removes the KL divergence from the time 0 prior but replaces it with KL from the previous time step, which is the same KL we start with (before dropping it). Please let us know if we're missing something.
> >
> > We have discussed our work with Khan since our submission, and the closest his papers have come to the BONG update is a passing comment in section 5.1 of Khan and Rue (2023) that conjugate updating is equivalent to running the BLR for one step with learning rate 1. This is close to our proposition 4.1 except that BLR includes the KL term that BONG drops. The reason BLR and BONG agree in this case is that the gradient of the KL is zero on the first iteration: $\\nabla\_{\\boldsymbol{\\psi}=\\boldsymbol{\\psi}\_{t-1}} KL(q\_{\\boldsymbol{\\psi}} | q_{\\boldsymbol{\\psi}\_{t-1}}) = \\boldsymbol{0}$. When BLR is run for multiple iterations (as it normally is) the gradient of the KL contributes to the update. In our revision we will note that BONG can be obtained as a special case of BLR with $\\alpha=I=1$. Nevertheless we believe that omitting the KL entirely because its regularizing role can be replaced by truncated (one step) updating is a substantially new insight relative to noting that mathematically its gradient drops out on the first iteration.
> >
> > We appreciate your careful thought about our work. Hopefully this discussion increases your estimation of the novelty of our primary proposal, in addition to the other contributions of our paper.
> >
> > —
> >
> > $\^*$Just to avoid any possible confusion: Linearizing the loss should not be confused with linearizing the model predictions $f\_t(\\theta)$ or $h\_t(\\theta)$ as in our proposition 4.2. The latter is an additional trick for approximating the expected Hessian.

---

> > > ### Comment · Reviewer_uSab · 2024-08-14
> > >
> > > I think that your statement "our primary proposal of dropping the KL term and doing one NGD step still appears to be new" in the first paragraph of your last reply is only partly true: the motivation of dropping the KL is new, but the one NGD step is not. It's exactly what I meant in my first review by "the idea of using a one-step natural gradient descent on the expected log-loss directly (without the KL term) has already been proposed in the literature and analyzed". You're right saying that linearizing the NLL in the full ELBO is an alternative means of deriving your update, but to be fair it exactly results in the same update. This is either your Equation (5) or Equation (4) of Lyu & Tsang (2021). In a sense, the KL is dropped in both cases, even though the initial motivation may be different.
> > >
> > > I'm not sure I fully understand why Khan told you there is no close connection to his works, because unless I'm mistaken, the SVB Equation (7) in Cherief-Abdellatif, Alquier & Khan (2019) is exactly the update rule (2) of Lyu & Tsang (i.e. linearization of the NLL in the full ELBO), which is then reformulated as a single-step update in Equation (10) of Cherief-Abdellatif, Alquier & Khan. This is not the same rule as your Equation (5) or Equation (4) of Lyu & Tsang because of a different choice of the variational parameter, but this still provides a one-step GD-like update based on the NLL only?
> > >
> > > Once again, I want to insist on the fact that I enjoyed the paper. I am convinced by the comments of the authors and will raise my score to 6 accordingly. However, I strongly advise the authors to work on the presentation of their paper, and recommend to put emphasis on the idea rather than on the rule (5) that has already been proposed and studied in the literature (in the exact same form by Lyu & Yang, and under a slightly different variant by Cherief-Abdellatif, Alquier & Khan?). A particular point worth highlighting is that their approach is actually framed in its most general form in Equation (3), which is broader than the specific one-step NGD rule (5) that has already been derived from a different perspective for exponential families.

---

> > > > ### Author Response · Authors · 2024-08-14
> > > >
> > > > We fully agree with your suggestions about presentation. We will put more emphasis on the idea of implicit regularization while also acknowledging that the resulting update rule matches what others have proposed based on different motivations. Thanks again for your thoughtful feedback!

---

### Official Review · Reviewer_ARk4 · 2024-07-12

**Soundness:** 2
**Presentation:** 3
**Contribution:** 1
**Rating:** 4
**Confidence:** 3

**Summary:**

The paper proposed a novel parameter update rule. The rule is natural gradient descent on the variational inference loss, where the prior term is dropped. Such rule is motivated in theory under the assumption of likelihood and variational distribution being conjugate, and in practice though experiments on MNIST.
Several variants are considered: various loss, various hessian approximations and various way of dealing with the intractable expectation.

**Strengths:**

The paper is well written and clear.

The setting is explained properly and the related work are exhaustive.

A lot of variants of the algorithm are considered. These are well explained and the previously known methods are highlighted. This schematic approach is very valuable for two reasons: \
-it helps in better understanding the proposed algorithms \
-it creates a clear picture of the field of approximated bayesian inference, putting in perspective every method with each other

In my opinion this is a contribution that may be worth acceptance, not the proposed update rule.

**Weaknesses:**

The theoretical motivation is quite weak:\
-Proposition 4.1 is nice, but the assumptions are overly restrictive. It feels like this is such a special case that is has no implication on real case scenarios \
-The claimed contribution (in Line 44) is a very weak contribution. The closed from expression with linearization+gaussian likelihood was already observed by Immer, and I don't think it is fair to claim it as a novel contribution of the paper. Specifically because there is no motivation to such approximations.

On the other hand, as the authors also correctly point out in Line 307, the empirical evaluation is based on MNIST only. A very small dataset which doesn't really motivate the superiority of the proposed algorithm.

Overall, I don't think the authors provided enough evidence (either theoretical or empirical) of the good performance of the proposed update rule.

TYPOS: \
-Line 126. I think the $\theta_{-1}$ was supposed to be a $\theta{t-1}$ \
-There is a duplicate reference: Line 336 and Line 340 are the same

**Questions:**

The notation $f_t$ that appears in Equations (9) and (10) was never introduced. I guess it refers to the evaluation of $f$ on $x_t$, is it correct?

**Limitations:**

Limitations are well discussed. Good job on this point

---

> ### Author Rebuttal · Authors · 2024-08-05
>
> Thank you for your feedback. We have a few responses below.
>
> “Proposition 4.1 is nice, but the assumptions are overly restrictive. It feels like this is such a special case that is has no implication on real case scenarios.”
>
> Most efficient VI methods exploit tricks from conjugate Bayesian inference. Thus we believe that Proposition 4.1 provides a useful motivating foundation for our approach, and ensures that our method is exact in certain simple cases. The experiments complement the theory by showing our method also works well in more general settings.
>
> “The closed form expression with linearization+gaussian likelihood was already observed by Immer”
>
> We agree line 44 should be revised to better explain our second contribution. (Please note our primary theoretical contribution is replacing the KL in the ELBO with implicit regularization from one-step updating, which is definitely new.)  We cite Immer for the linear(f) approximation when we first introduce it (line 51). Our contribution here is that the update rule for CM-EKF (which we extand to other BONG variants) follows from two different approximations. One is known (and we credit Immer, Ollivier, and Tronarp et al.) while the other is new. Informally, showing the same algorithm arises in different ways from different approximations adds motivation for the approach. Further motivations for the linear(f)-Gaussian approximation are in Immer and Tronarp et al., which we could explain briefly in our paper.
>
> “the empirical evaluation is based on MNIST only”
>
> We also report experiments using the SARCOS dataset (appendix B4-B10). We agree it will be important to scale up as we continue this line of research, but we believe the current experiments are sufficient for a primarily theoretical paper.
>
> “I don't think the authors provided enough evidence (either theoretical or empirical) of the good performance of the proposed update rule.”
>
> We believe the experiments show a clear advantage of the proposed update rule, especially when taking compute costs into account. We agree that MNIST is a simple dataset, but we did not have time for more experiments, since our main contribution is the theoretical framework.
>
> “$f\_t$ in Equations (9) and (10) refers to the evaluation of $f$ on $x\_t$?”
>
> Yes we will add this.

---

> > ### Comment · Reviewer_ARk4 · 2024-08-08
> >
> > "...follows from two different approximations. One is known (and we credit Immer, Ollivier, and Tronarp et al.) while the other is new"
> >
> > What do you refer to with "the other"?

---

> > > ### Author Response · Authors · 2024-08-09
> > >
> > > The other is the linear($f$)-delta approximation.
> > >
> > > The linear($h$)-Gaussian approximation linearizes the mean parameters of the observation distribution (e.g., the predicted class probabilities) and replaces the likelihood with a moment-matched Gaussian. This is the method of Ollivier and Tronarp et al. The linear($f$)-delta approximation linearizes the natural parameters of the observation distribution (e.g., the predicted class logits) and replaces the expectations in (9) and (10) with plugin values at the prior mean. This is new.
> > >
> > > Proposition 4.2 shows the linear($f$)-delta and linear($h$)-Gaussian approximations yield the same expressions for the expected gradient $\\boldsymbol{g}_t$ and expected Hessian $\\boldsymbol{G}_t$ and consequently the same updates.
> > >
> > > We see that our initial rebuttal wrote linear($f$) in two places where we meant linear($h$). Apologies for the confusion. There is a similar error at lines 51-52 in the paper which we will correct.

---

### Official Review · Reviewer_6NTx · 2024-07-12

**Soundness:** 3
**Presentation:** 4
**Contribution:** 4
**Rating:** 7
**Confidence:** 4

**Summary:**

This paper proposes a new approximate Bayesian technique specifically for online learning whereby posterior distributions over modeling parameters at time $t$ can be achieved with a single natural gradient step evaluated on a model using the previous posterior distribution at time $t-1$ as the prior distribution. This method is rigorously derived in a conjugate setting with an exponential family and extended approximately with various ablations to more expressive black-box settings. Extensive empirical studies are conducted validating the performance of the method.

**Strengths:**

An interesting connection is made for online learning in the conjugate setting allowing for dropping the KL regularization term and producing posterior updates with a single natural gradient step. As mentioned in the summary, various ablations are explored which trade-off precision and computation speed. Finally, many extensive experiments are conducted comparing this method to other Bayesian online learning approaches.

**Weaknesses:**

The main weakness of the method is the lack of experiments evaluating the resulting model uncertainty achieved after training. From what I read, it seems to only be predictive performance via misclassification and negative log predictive density. Granted, the latter does have some ablations where it is integrated over posterior samples; however, the presentation of these results are a bit difficult to interpret other than just methods with lower values being better. It would be interesting to investigate the predictive uncertainty via expected calibration error, or some other calibration metric.

**Questions:**

No other questions, please address the main weaknesses. Please let me know if I missed or misunderstood anything; I am more than happy to be convinced otherwise.

**Limitations:**

The authors adequately addressed the limitations of the proposed method.

---

> ### Author Rebuttal · Authors · 2024-08-06
>
> Thank you for your thoughtful comments.
> As requested, we have attached some figures that plot the ECE (expected calibration error) vs time for the various methods, when evaluated on MNIST using a CNN. Fig 1 shows that the BONG method is (slightly) better calibrated than the other
> methods, for small sample sizes, but not surprisingly, all methods converge to similar results once they have seen enough data.
> We will add these to the appendix in our revision.

---

> > ### Comment · Reviewer_6NTx · 2024-08-10
> >
> > Thank you for the additional experiments conducted. These do indeed resolve the concerns I had regarding the method.
> >
> > I will maintain my initial score of a 7 with the hope of this paper being accepted.

---

### Official Review · Reviewer_ch8o · 2024-07-13

**Soundness:** 4
**Presentation:** 3
**Contribution:** 4
**Rating:** 7
**Confidence:** 4

**Summary:**

This paper proposes a generalized framework for variational sequential inference of Bayesian neural networks with Gaussian prior distributions by further approximating the KL-divergence and expectations. It presents a very elegant unification of a wide range of (approximate) variational inference algorithms for Bayesian neural networks developed over the past 10 years and contains an extensive empirical evaluation on subsets of the MNIST dataset.

**Strengths:**

The strongest point of the paper is the unification of many previously presented approximations as well as the methodological evaluation and comparison among them.

**Weaknesses:**

The space of 9 pages is way to short to do this full justice and I often had to read into the 30 pages (!) appendix. I recommend the authors to further considering a journal submission.

**Questions:**

Page 4: The abbreviation PSD as "positive-semi-definite" has never been introduced.
Page 5: jac has not been introduced
Page 7: In line 239, the component-wise product of \sigma^{-2} and \mu should be indicated somewhere

**Limitations:**

This is not applicable.

---

> ### Author Rebuttal · Authors · 2024-08-05
>
> Thank you for your feedback and positive evaluation. We will consider submitting to a journal in the future, but we feel that NeurIPS has higher visibility. We agree the unifying framework makes a good contribution on its own (and we have more theoretical work in progress on these lines) but for NeurIPS we want to highlight the proposed BONG methods and their empirical success.
>
> We will explain PSD, jac (Jacobian), and the componentwise products in the revision. Thanks for catching those.

---

### Author Rebuttal · Authors · 2024-08-06

We thank all the reviewers for their useful feedback. We give individual responses  below.
As requested by reviewer 6NTx, we are also attaching some figures that plot the ECE (expected calibration error) vs time for the various methods, when evaluated on MNIST using a CNN. Fig 1  shows that the BONG method is (slightly) better calibrated than the other
methods, for small sample sizes, but not surprisingly, all methods converge to similar results once they have seen enough data.
We will add these to the appendix in our revision.

---

### Decision · Program_Chairs · 2024-09-25

**Decision:**

Accept (poster)

**Comment:**

This paper studies the problem of online Bayesian inference in Bayesian neural networks using variational inference. The paper considers an online natural gradient update rule given by only considering the expected log-likelihood term and using a unit learning rate. This Bayesian online natural gradient update rule is studied theoretically under a conjugate prior setting. Next, the paper proposes a deterministic way of efficiently computing the expectations in the update rule for Gaussian variational families. The method is then evaluated on a number of data sets, including the MNIST data set.

Overall, the reviewers found the contribution to be interesting to the NeurIPS community. Reviewer 6NTx commented on a lack of experiments evaluating the resulting model uncertainty; these concerns were addressed during the rebuttal and discussion phase. Several reviewers brought up issues of novelty and wanted to see additional discussion that much more explicitly discusses the contributions of this paper in the context of related update rules in the literature. Please revise the paper to include these discussions in the final version.

Based on the reviewer comments, I recommend accepting the paper.